# Explainable Multimodal Regression via Information Decomposition

## Abstract

Multimodal regression aims to predict a continuous target from heterogeneous input sources and typically relies on fusion strategies such as early or late fusion. However, existing methods lack principled tools to disentangle and quantify the individual contributions of each modality and their interactions, limiting the interpretability of multimodal fusion. We propose a novel multimodal regression framework grounded in Partial Information Decomposition (PID), which decomposes modality-specific representations into unique, redundant, and synergistic components. The basic PID framework is inherently underdetermined. To resolve this, we introduce inductive bias by enforcing Gaussianity in the joint distribution of latent representations and the transformed response variable (after inverse normal transformation), thereby enabling analytical computation of the PID terms. Additionally, we derive a closed-form conditional independence regularizer to promote the isolation of unique information within each modality. Experiments on six real-world datasets, including a case study on large-scale brain age prediction from multimodal neuroimaging data, demonstrate that our framework outperforms state-of-the-art methods in both predictive accuracy and interpretability, while also enabling informed modality selection for efficient inference.

## 1 Introduction

Multimodal regression has become increasingly important due to its ability to effectively integrate heterogeneous data sources to predict a continuous target, and has found applications across a wide range of domains. In healthcare diagnostics, for example, it leverages medical imaging and clinical text data to predict patient outcomes such as survival time and disease severity scores (Soenksen et al., 2022). In sentiment analysis, models combine audio, visual, and textual information to assess human emotions and opinions more accurately (Soleymani et al., 2017). To support such tasks, various multimodal fusion paradigms have been proposed, ranging from widely used attention-based mechanisms (Hori et al., 2017; Tsai et al., 2019) to more recent methods grounded in the information bottleneck (IB) principle (Tishby et al., 2000; Mai et al., 2022). Despite its performance gains, multimodal regression often faces substantial interpretability challenges, particularly at the modality level. For example, in sentiment prediction, several critical questions arise: Does the audio modality contribute more predictive power than text? Do audio and video modalities create a synergistic effect that enhances the final decision, or are they largely redundant, such that a single modality is sufficient for reliable predictions? The lack of clarity regarding the specific contributions and interactions of modalities undermines model trustworthiness, transparency, and practical applicability in real-world settings (Das & Rad, 2020; Tsankova et al., 2015).

The partial information decomposition (PID) framework (Kraskov et al., 2004; Kolchinsky, 2022; Williams & Beer, 2010), originally developed in neuroscience, offers a formal approach to quantify how two random variables $x_1$ and $x_2$ interact with a third variable $y$ by decomposing the mutual information $I(x_1, x_2; y)$ between $(x_1, x_2)$ and $y$ into four non-negative components: two unique information terms, $U_1$ and $U_2$, which capture the individual contributions of $x_1$ and $x_2$; a synergy term $S$, representing information that emerges only from the joint knowledge of both variables; and a redundancy term $R$, which reflects information about $y$ that is attainable by either $x_1$ or $x_2$. This elegant decomposition makes PID a promising tool for analyzing how multimodal interactions contribute to predictive outcomes. However, its application in multimodal learning remains limited and underexplored (Liang et al., 2023; Xin et al., 2025), largely due to the underdetermined nature

of the decomposition, which leads to intractable optimization when dealing with continuous and high-dimensional variables.

This work presents **PIDReg**, a novel multimodal **reg**ression framework that enables the computation of **PID** and seamlessly integrates it into an end-to-end learning process. The key idea is to enforce the joint distribution of the learned modality-specific representations and the target response variable (after the inverse normal transformation (Conover, 1999)) to follow a multivariate Gaussian, thereby enabling an analytical PID solution even in high-dimensional settings. To support this, we introduce two regularization terms: one that promotes Gaussianity and another that encourages the uniqueness of information captured by each modality-specific encoder. Both are formulated using the recently re-emerged Cauchy–Schwarz (CS) divergence (Yu et al., 2025; 2024b). To summarize:

1. We propose a generic PID-based multimodal regression framework that ensures interpretability by revealing the contributions of individual modalities and their high-order interactions to the output.

2. We develop an analytically tractable optimization scheme for PIDReg for continuous and high-dimensional variables, incorporating Gaussianity enforcement via the Shapiro–Wilk test (Shapiro & Wilk, 1965) and a CS divergence-based regularization.

3. Extensive experiments on six real-world applications from diverse domains, including healthcare, physics, affective computing, and robotics, and covering both univariate and multivariate prediction tasks, demonstrate that PIDReg outperforms six state-of-the-art methods in terms of both predictive accuracy and interpretability.

## 2 RELATED WORK

### 2.1 FUSION STRATEGIES IN MULTIMODAL LEARNING

Various fusion paradigms have been proposed for multimodal learning (Li & Tang, 2024). Early fusion, also known as feature-level fusion, combines modalities either by concatenating raw features (Ortega et al., 2019) or integrating modality-specific embeddings (Mai et al., 2022; Tsai et al., 2019; Zadeh et al., 2017). Late fusion, or decision-level fusion, trains separate models per modality and aggregates their predictions (Huang et al., 2020). In addition, hybrid fusion strategies combine the merits of early and late fusion to exploit their complementary advantages (Hemker et al., 2024).

Among these, feature-level fusion is widely used. Beyond simple concatenation, advanced techniques compute tensor products of modality-specific representations to model higher-order interactions (Fukui et al., 2016; Zadeh et al., 2017), or apply gating and attention mechanisms (Hori et al., 2017; Kiela et al., 2018; Tsai et al., 2019). However, feature-level fusion methods can be vulnerable to noisy or corrupted modalities, which may degrade overall performance (Ma et al., 2021). Our framework constructs the joint representation as a linear combination of modality-specific embeddings, complemented by a pseudo representation explicitly capturing synergistic interactions. The use of linear fusion weights offers full interpretability, enabling dynamic modality selection: when a modality is unreliable, its contribution to the final prediction is naturally suppressed.

### 2.2 INTERPRETABILITY IN MULTIMODAL LEARNING

The heterogeneity of multimodal data, combined with their complex interdependencies, makes it challenging to interpret the prediction process and disentangle the contribution of each modality to the final decision (Liang et al., 2024; Binte Rashid et al., 2024). Several conventional explainable artificial intelligence (XAI) approaches can be straightforwardly extended to the multimodal setting (Rodis et al., 2024). For example, DIME (Lyu et al., 2022) applies LIME (Ribeiro et al., 2016) separately to each unimodal contribution and their interactions, assuming that the multimodal model is formed as an aggregation of these components. In another study (Wang et al., 2021), image data and metadata are jointly used for skin lesion diagnosis, where Grad-CAM (Selvaraju et al., 2017) is employed to interpret the image features, while kernel SHAP (Lundberg & Lee, 2017) is applied to explain the contribution of metadata. Recently, Zhu et al. (2025) and Wang et al. (2023) apply the information bottleneck (IB) principle (Tishby et al., 2000) to cross-modal feature attribution, improving interpretability in vision-language models (Radford et al., 2021). However, these methods are fundamentally limited by their reliance on post-hoc explanations, applied only after

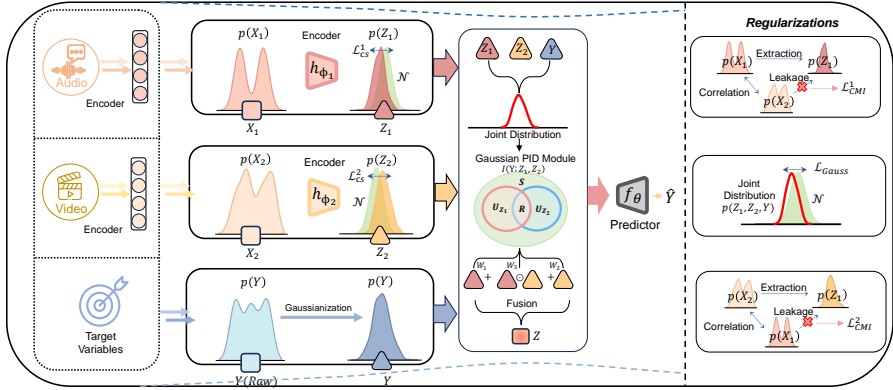

Figure 1: Framework of Partial Information Decomposition for Multimodal Regression (PIDReg), illustrated with video and audio modalities, where $P(X_1)$, $P(X_2)$, and $P(Y)$ denote empirical data distributions that may deviate from Gaussianity (e.g., skewed or heavy-tailed).

model training. This creates a risk of inconsistency between the explanations and the model's actual decision-making process, raising concerns about their faithfulness (Das & Rad, 2020).

In contrast to existing methods that primarily offer *instance-level* interpretability in a *post-hoc* fashion, our approach emphasizes *intrinsic* interpretability by embedding explanatory mechanisms directly into the model design. This enables more transparent decision logic and direct explanations. Specifically, our method focuses on *modality-level* interpretability, identifying which modalities or cross-modal interactions are most critical to the decision-making process.

# 3 PIDREG: PARTIAL INFORMATION DECOMPOSITION FOR MULTIMODAL REGRESSION

## 3.1 OVERALL FRAMEWORK

Our PIDReg framework shown in Fig. 1 comprises two stochastic, modality-specific encoders, $h_{\varphi_1}$ and $h_{\varphi_2}$; an interpretable PID-guided feature fusion module; and a predictor $f_\theta$ operating on the fused features. Additional regularizers, including uniqueness information regularization and joint Gaussian regularization, are applied to ensure a rigorous implementation.

Given two modality–specific embeddings $R_1 = h_{\phi_1}(X_1)$, $R_2 = h_{\phi_2}(X_2)$ in $\mathbb{R}^d$, we introduce an adaptive linear–noise information bottleneck (IB) (Schulz et al., 2020) to regulate the information flow in each modality and enhance generalization. For each modality $m \in \{1, 2\}$, we compute empirical batch statistics mean vector $\mu_{R_m}$ and covariance matrix $\Sigma_{R_m}$ and sample Gaussian noise $\epsilon_m \sim \mathcal{N}(\mu_{R_m}, \Sigma_{R_m})$ matched to $R_m$. The bottleneck output is defined by a convex interpolation:

$$Z_m = \lambda_m R_m + (1 - \lambda_m)\epsilon_m, \tag{1}$$

where $\lambda_m \in (0, 1)$ is a trainable scalar. When $\lambda_m \approx 1$, $R_m$ is preserved; when $\lambda_m \approx 0$, it is replaced by homoscedastic noise with identical first and second moments, effectively pushing $I(Z_m; X_m)$ toward zero. This formulation eliminates the need for reparameterization (Kingma et al., 2014) and enables an end-to-end learning of bottleneck strength. The rationality of our IB mechanism, including comparison with the variational IB (Alemi et al., 2016), is provided in Appendix D.3.

After extracting modality-specific embeddings $Z_1$ and $Z_2$, we aim to construct a fused representation $Z$ as a linear combination of $Z_1$, $Z_2$, and $\tilde{Z}$:

$$Z = w_1 Z_1 + w_2 Z_2 + w_3 \tilde{Z}, \quad \text{s.t.} \quad \tilde{Z} = Z_1 \odot Z_2, \tag{2}$$

where $\tilde{Z}$ captures the *synergistic* effect that emerges only from the joint interaction between the two modalities. We expect the weights $w_1$, $w_2$, and $w_3$ to accurately reflect the contributions of $Z_1$, $Z_2$, and $\tilde{Z}$, respectively, thereby enhancing interpretability. The estimation of these weights is detailed in Section 3.1.1. The fused representation $Z$ is passed through a predictor $f_\theta$ to generate final output.

In this work, we model the synergistic effect using the Hadamard (i.e., element-wise) product $\tilde{Z} = Z_1 \odot Z_2$. When synergy arises from feature-specific dependencies, and $Z_1$ and $Z_2$ are regularized to encode primarily unique information (see Section 3.1.3), their element-wise product highlights cross-dimensional couplings between these unique components, which aligns with the concept of synergy. In practice, the Hadamard product is widely used to model interactions between two drug representations for drug synergy prediction (Al-Rabeah & Lakizadeh, 2022; Yang et al., 2023). It has also been employed to fuse image and text representations in visual question answering (Fukui et al., 2016; Kim et al., 2016) and multimodal sentiment analysis (Zadeh et al., 2017). Advantages of Hadamard product over other synergy modeling schemes are provided in Appendix G.6.

### 3.1.1 EXPLAINABLE FEATURE FUSION WITH GAUSSIAN PID

Given two random variables $Z_1$ and $Z_2$, and a target variable $Y$ with joint distribution $P_{Z_1 Z_2 Y}$, the total information that $(Z_1, Z_2)$ carries about $Y$ is the mutual information $I(Y; Z_1, Z_2)$. The PID framework decomposes this quantity into four non-negative terms (Williams & Beer, 2010):

$$\begin{cases} I(Y; Z_1, Z_2) = U_{Z_1} + U_{Z_2} + R + S \\ I(Y; Z_1) = U_{Z_1} + R \\ I(Y; Z_2) = U_{Z_2} + R, \end{cases} \tag{3}$$

where $U_{Z_1}$ and $U_{Z_2}$ denote the information *uniquely* provided by $Z_1$ and $Z_2$, respectively. The term $S$ captures the *synergistic* information that arises only through the joint interaction of $Z_1$ and $Z_2$, and cannot be obtained from either alone. The term $R$ represents the *redundant* information that is available in both $Z_1$ and $Z_2$ and can be extracted from either. Note that the redundancy in PID means that the same informative content about $Y$ is accessible through either $Z_1$ or $Z_2$, so once one source provides it, the other does not need to do so.

Despite the elegant expression of Eq. (3), it defines four terms with only three equations, resulting in an underdetermined system that admits infinitely many non-negative solutions. To resolve this ambiguity, one of the partial information components must be formally specified. In this work, we adopt the concept of union information $I^{\cup}(Y : Z_1; Z_2) := U_{Z_1} + U_{Z_2} + R$ (Bertschinger et al., 2014), which introduces an additional equation that restricts the sum of unique information and redundancy, and enjoy appealing properties such as additivity and continuity (Lyu et al., 2025). This additional constraint renders the system fully determined and enables a unique decomposition.

**Definition 1** (Union Information (Bertschinger et al., 2014)). *The union information about $Y$ present in both $Z_1$ and $Z_2$ is given by:*

$$\widetilde{I^{\cup}}(Y : Z_1; Z_2) := \min_{Q \in \Delta_P} I_Q(Y; Z_1, Z_2), \tag{4}$$

*where* $\Delta_P := \{Q_{Y Z_1 Z_2} : Q_{Y Z_1} = P_{Y Z_1}, Q_{Y Z_2} = P_{Y Z_2}\}$, *and* $I_Q$ *is the mutual information under the joint distribution* $Q_{Y Z_1 Z_2}$. *The remaining PID terms follow Eq. (3).*

Intuitively, union information quantifies the minimal amount of "shared evidence" about $Y$ that remains no matter how one couples $Z_1$ and $Z_2$ while preserving their marginals. A more detailed intuition and discussion are provided in Appendix A.1. However, optimizing Eq. (4) for high-dimensional variables is computationally infeasible. To simplify the problem, we restrict the search space of $\Delta_P$ in Eq. (4) and assume that the joint distribution $P_{Z_1 Z_2 Y} \sim \mathcal{N}(\mu, \Sigma^P)$ is multivariate Gaussian, which enables closed-form expressions for mutual information terms in Eq. (3). For instance, $I(Y; Z_1, Z_2) = \frac{1}{2} \log \left( \frac{\det(\Sigma_{Z_1 Z_2})}{\det(\Sigma_{Z_1 Z_2 | Y})} \right)$. Note that this Gaussian assumption is not imposed on the original input data $X_1, X_2$, or $Y$, but rather on the latent representations. Consequently, PIDReg fully accommodates real-world phenomena such as heavy tails, skewness, and even highly multimodal distributions, as demonstrated in our experiments. Further clarification of the Gaussian assumption rationale is provided in the Appendix A.2. This Gaussian assumption greatly simplifies the optimization and allows the problem in Eq. (4) to be reformulated as (Venkatesh et al., 2023):

$$\widetilde{I_G^{\cup}}(Y : Z_1; Z_2) := \min_{\Sigma_{Z_1 Z_2 | Y}^Q} \frac{1}{2} \log \det \left( I + \sigma_Y^{-2} \begin{bmatrix} \Sigma_{Y Z_1}^P \\ \Sigma_{Y Z_2}^P \end{bmatrix}^T \left( \Sigma_{Z_1 Z_2 | Y}^Q \right)^{-1} \begin{bmatrix} \Sigma_{Y Z_1}^P \\ \Sigma_{Y Z_2}^P \end{bmatrix} \right) \quad \text{s.t.} \quad \Sigma_{Z_1 Z_2 | Y}^Q \succeq 0, \tag{5}$$

which is amenable to projected gradient descent (Riedmiller & Braun, 1993) and admits an analytical gradient. We refer interested readers to (Venkatesh et al., 2023) for more details.

After solving Eqs. (3) and (5), the fusion weights in Eq. (2) are computed based on the PID components. Since the redundancy $R$ can be attributed to either modality-specific representation, we introduce a binary variable $\xi \sim \text{Bernoulli}(0.5)$ during training to stochastically control the assignment. The weights are computed as:

$$w_1 = \frac{U_{Z_1} + \xi R}{T}, \quad w_2 = \frac{U_{Z_2} + (1-\xi)R}{T}, \quad w_3 = \frac{S}{T}, \quad \text{where } T = U_{Z_1} + U_{Z_2} + S + R. \quad (6)$$

This stochastic formulation ensures symmetric sharing of redundancy, reduces bias toward either modality. Note that, the above mechanism can be naturally extended to more than two modalities, as further discussed in the Appendix F.

### 3.1.2 GAUSSIAN REGULARIZATION OF THE JOINT DISTRIBUTION $P_{Z_1 Z_2 Y}$

To encourage the marginal distribution $p(Z_m)$ to resemble a Gaussian, we adopt the CS divergence (Jenssen et al., 2006; Yu et al., 2025), which has recently gained renewed attention in representation learning (Tran et al., 2022; Yu et al., 2024b). It is defined as:

$$D_{\text{CS}}(p(z); q(z)) = \log \left( \int q(z)^2 \, dz \right) + \log \left( \int p(z)^2 \, dz \right) - 2 \log \left( \int p(z)q(z) \, dz \right). \quad (7)$$

The motivation for using CS divergence, rather than the popular Kullback-Leibler (KL) divergence and maximum mean discrepancy (MMD) (Gretton et al., 2012), is discussed in the Appendix A.3.

In our setting, $p(z)$ denotes the probability density function of $Z_m$, $q(z) \sim \mathcal{N}(0, I)$ represents an isotropic Gaussian. Empirically, given $M$ samples $\{z_i^p\}_{i=1}^M$ drawn from $p(z_m)$ and $N$ samples $\{z_j^q\}_{j=1}^N$ drawn from $\mathcal{N}(0, I)$, the CS divergence $D_{\text{CS}}(p(z); \mathcal{N}(0, I))$ can be estimated as (Jenssen et al., 2006):

$$\widehat{D}_{\text{CS}}(p(z); \mathcal{N}(0, I)) = \log \left( \frac{\sum_{i,j=1}^M \kappa(z_i^p, z_j^p)}{M^2} \right) + \log \left( \frac{\sum_{i,j=1}^N \kappa(z_i^q, z_j^q)}{N^2} \right) - 2 \log \left( \frac{\sum_{i=1}^M \sum_{j=1}^N \kappa(z_i^p, z_j^q)}{MN} \right), \quad (8)$$

where $\kappa$ is a kernel function with width $\sigma$ such as Gaussian $\kappa(z_i, z_j) = \exp \left( -\frac{\|z_i - z_j\|^2}{2\sigma^2} \right)$. Owing to the symmetry of the CS divergence, the regularization on $p(Z_1)$ and $p(Z_2)$ is formulated as:

$$\mathcal{L}_{\text{CS}} = \widehat{D}_{\text{CS}}(p(z_1); \mathcal{N}(0, I)) + \widehat{D}_{\text{CS}}(p(z_2); \mathcal{N}(0, I)). \quad (9)$$

Additionally, we apply a rank-based, outlier-aware inverse normal transformation (Conover, 1999) to the target variable $Y$ to approximate a normal distribution. However, regularizing the marginal distributions $P_{Z_m}$ and $P_Y$ does not ensure that the joint distribution $P_{Z_1 Z_2 Y}$ of $Z_1$, $Z_2$, and $Y$ is multivariate Gaussian. To address this, we further introduce a regularization term $\mathcal{L}_{\text{Gauss}}$ to promote joint normality. Following Palmer et al. (2018), we apply whitening and vectorization to convert the multivariate Gaussianity test into a univariate Shapiro-Wilk (SW) test (Shapiro & Wilk, 1965).

We construct a feature matrix $F = \{f_i\}_{i=1}^n$ with $f_i = [Y_i, Z_{1,i}, Z_{2,i}]^\top \in \mathbb{R}^{2d+1}$, compute its sample mean $\bar{f}$ and covariance matrix $S$, and apply the whitening transformation:

$$f_i^w = S^{-\frac{1}{2}}(f_i - \bar{f}). \quad (10)$$

Under the Gaussian assumption, the whitened samples $f_i^w$ should follow $f_i^w \sim \mathcal{N}_{2d+1}(0, I)$.

We then vectorize the whitened matrix $F^w = [f_1^w, \ldots, f_n^w]^\top \in \mathbb{R}^{n \times (2d+1)}$ as:

$$f_{\text{vec}} = \text{vec}(F^w) = \left( f_{11}^w, f_{12}^w, \ldots, f_{1(2d+1)}^w, f_{21}^w, \ldots, f_{n(2d+1)}^w \right)^\top \in \mathbb{R}^{n(2d+1) \times 1}. \quad (11)$$

The final SW test statistic is given by:

$$W = \left( \sum_{i=1}^{n(2d+1)} a_i f_{(i)}^w \right)^2 \Big/ \sum_{i=1}^{n(2d+1)} (f_i^w - \bar{f^w})^2 \quad (12)$$

where $f_{(i)}^w$ is the $i$-th order statistic and $a_i$ are coefficients under the standard normal distribution. If $W < W_\alpha(n(2d+1))$ at significance level $\alpha$, the null hypothesis $H_0$ (normality) is rejected.

The regularization term is defined as:

$$\mathcal{L}_{\text{Gauss}} = -\log(W), \tag{13}$$

which approaches zero as $W \to 1$ (ideal Gaussianity), and increases otherwise.

### 3.1.3 REGULARIZING FOR UNIQUE INFORMATION EXTRACTION

To ensure that $Z_1$ and $Z_2$ primarily capture unique information from their respective modalities, we introduce an additional regularization term that explicitly minimizes the conditional mutual information (CMI) $I(Z_1; X_2|X_1)$ and $I(Z_2; X_1|X_2)$. Minimizing $I(Z_1; X_2|X_1)$ encourages $Z_1$ to encode only information specific to $X_1$ that is independent of $X_2$.

From a probabilistic perspective, the conditional independence between $Z_1$ and $X_2$ given $X_1$ implies that $p(Z_1|X_1, X_2) = p(Z_1|X_1)$, which can be reformulated as $p(X_1, X_2, Z_1)p(X_1) = p(X_1, X_2)p(X_1, Z_1)$. This observation again motivates the use of the CS divergence in Eq. (7) to measure the closeness between these two joint distributions as a proxy for conditional independence:

$$I_{\text{CS}}(Z_1; X_2|X_1) = D_{\text{CS}}(p(X_1, X_2, Z_1)p(X_1); p(X_1, X_2)p(X_1, Z_1)). \tag{14}$$

Proposition 1 provides a closed-form estimation of our defined CMI in Eq. (14).

**Proposition 1.** *Given $N$ observations $\{\mathbf{x}_{1,i}, \mathbf{x}_{2,i}, \mathbf{z}_{1,i}\}_{i=1}^N$ drawing from an unknown and fixed joint distribution $p(X_1, X_2, Z_1)$ in which $\mathbf{x}_{1,i} \in \mathbb{R}^{d_1}$, $\mathbf{x}_{2,i} \in \mathbb{R}^{d_2}$, and $\mathbf{z}_{1,i} \in \mathbb{R}^d$. Let $M \in \mathbb{R}^{N \times N}$ be the Gram (a.k.a., kernel) matrix for variable $X_1$, that is, $M_{ji} = \exp\left(-\frac{\|x_{1,j} - x_{1,i}\|_2^2}{2\sigma^2}\right)$, in which $\sigma$ is the kernel width. Likewise, let $K \in \mathbb{R}^{N \times N}$ and $L \in \mathbb{R}^{N \times N}$ be the Gram matrices for variables $X_2$ and $Z_1$, respectively. The empirical estimator of Eq. (14) is given by:*

$$\widehat{I_{CS}}(Z_1; X_2|X_1) = -2\log\left(\sum_{j=1}^N \left(\left(\sum_{i=1}^N M_{ji}\right)\left(\sum_{i=1}^N K_{ji}M_{ji}\right)\left(\sum_{i=1}^N L_{ji}M_{ji}\right)\right)\right) +$$

$$\log\left(\sum_{j=1}^N \left(\left(\sum_{i=1}^N K_{ji}L_{ji}M_{ji}\right)\left(\sum_{i=1}^N L_{ji}\right)^2\right)\right) + \log\left(\sum_{j=1}^N \left(\frac{\left(\sum_{i=1}^N K_{ji}L_{ji}\right)^2 \left(\sum_{i=1}^N L_{ji}M_{ji}\right)^2}{\left(\sum_{i=1}^N K_{ji}L_{ji}M_{ji}\right)}\right)\right). \tag{15}$$

Estimator to $I(Z_2; X_1|X_2)$ can be derived similarly. Our final regularization is expressed as:

$$\mathcal{L}_{\text{CMI}} = \widehat{I}(Z_1; X_2|X_1) + \widehat{I}(Z_2; X_1|X_2). \tag{16}$$

### 3.2 OPTIMIZATION AND ALGORITHM

The overall loss function of our PIDReg framework is expressed as follows:

$$\mathcal{L} = \mathcal{L}_{\text{pred}} + \lambda_1 \mathcal{L}_{\text{CS}} + \lambda_2 \mathcal{L}_{\text{CMI}} + \lambda_3 \mathcal{L}_{\text{Gauss}}, \tag{17}$$

where $\lambda_1$, $\lambda_2$, and $\lambda_3$ are regularization weights. Here, $\mathcal{L}_{\text{pred}}$ denotes the mean squared error (MSE) between the ground-truth $y$ and the prediction $\hat{y}$, while $\mathcal{L}_{\text{CS}}$, $\mathcal{L}_{\text{CMI}}$, and $\mathcal{L}_{\text{Gauss}}$ are defined earlier.

Formally, the optimization problem can be expressed as:

$$\min_{\theta, \varphi_1, \varphi_2, \mathbf{w}} \mathcal{L}(f_\theta, h_{\varphi_1}, h_{\varphi_2}, \mathbf{w}), \tag{18}$$

where $\mathbf{w} = [w_1, w_2, w_3]^T$ denotes fusion parameters (see Eq. (6)).

We design a two-stage optimization strategy. In Stage I, all parameters are updated in an end-to-end manner. Once $\mathbf{w}$ stabilizes or exhibits temporal consistency, we move to Stage II, where we optimize the following objective:

$$\min_{\theta, \varphi_1, \varphi_2} \mathcal{L}(f_\theta, h_{\varphi_1}, h_{\varphi_2}, \mathbf{w}^*), \tag{19}$$

where $\mathbf{w}^*$ denotes the optimal fusion weight obtained at the end of Stage I. In Stage II, the predictor $f_\theta$ and modality-specific encoders $h_{\varphi_m}$ are updated according to Eq. (20) and Eq. (21), respectively:

$$f_\theta^{(t+1)} = f_\theta^{(t)} - \eta_{\text{pred}} \nabla_{f_\theta} \mathcal{L}_{\text{pred}}, \tag{20}$$

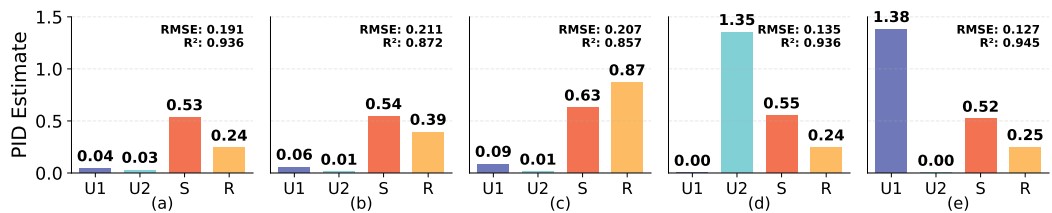

Figure 2: Estimated PID values when (a) $w_{u1} = 0$, $w_{u2} = 0$, $w_s = 0.75$, $w_r = 0.25$; (b) $w_{u1} = 0$, $w_{u2} = 0$, $w_s = 0.50$, $w_r = 0.50$; (c) $w_{u1} = 0$, $w_{u2} = 0$, $w_s = 0.25$, $w_r = 0.75$; (d) $w_{u1} = 0$, $w_{u2} = 0.80$, $w_s = 0.10$, $w_r = 0.10$; (e) $w_{u1} = 0.80$, $w_{u2} = 0$, $w_s = 0.10$, $w_r = 0.10$.

and,

$$h_{\varphi_m}^{(t+1)} = h_{\varphi_m}^{(t)} - \eta_{\text{encoder}} \nabla_{h_{\varphi_m}} (\lambda_1 \mathcal{L}_{\text{CS}}^m + \lambda_2 \mathcal{L}_{\text{CMI}}^m + \lambda_3 \mathcal{L}_{\text{Gauss}}). \quad (21)$$

We refer interested readers to Appendix C for a detailed description of the full algorithm, and to Appendix D.1 for an ablation study of the regularization components.

## 4 EXPERIMENTS

### 4.1 EXPERIMENTS ON SYNTHETIC DATA

We first demonstrate the properties of our model on synthetic data, where the trade-offs between redundancy, synergy, and unique information are controllable. First, latent variables representing redundancy and unique information are independently sampled from a standard normal distribution: $R$, $U_1$, $U_2 \sim \mathcal{N}(0, 1)$. The latent pair $[R, U_1]$ is then nonlinearly projected into a higher-dimensional observation space via a multi-layer perceptron (MLP):

$$X_1 = \tanh\left([R, U_1]W^{(1)} + b^{(1)}\right)W^{(2)} + b^{(2)} + \varepsilon_1, \quad \varepsilon_1 \sim \mathcal{N}(0, \sigma_1^2 I_{d_1}), \quad (22)$$

where the weights and biases are defined as $W^{(1)} \in \mathbb{R}^{2 \times h_1}$, $b^{(1)} \in \mathbb{R}^{h_1}$, $W^{(2)} \in \mathbb{R}^{h_1 \times d_1}$, and $b^{(2)} \in \mathbb{R}^{d_1}$, with all parameters initialized from $\mathcal{N}(0, \alpha^2)$. The second modality $X_2$ is generated analogously by applying an independent MLP to the latent pair $[R, U_2]$, producing $X_2 \in \mathbb{R}^{d_2}$.

The target variable $Y$ is then constructed from these informational components through:

$$Y = w_r \tanh(R) + w_{u1} \sin(U_1) + w_{u2} \sin(U_2) + w_s U_1 U_2 + \varepsilon, \quad \varepsilon \sim \mathcal{N}(0, \sigma_\varepsilon^2), \quad (23)$$

where the product $U_1 U_2$ synergistically influence $Y$. By adjusting the weights $w_r$, $w_{u1}$, $w_{u2}$, and $w_s$, the relative contributions of redundancy, uniqueness, and synergy can be explicitly controlled. Experimental results in Fig. 2 show that PIDReg accurately estimates the relative strengths of the underlying generative factors, as reflected by the positive correlation with the true weights (i.e., the monotonic trend of $S/R$ with respect to $w_s/w_r$) and the near-zero value of $U$ when $w_u \approx 0$. Experimental results for the non-Gaussian latents $R$, $U_1$ and $U_2$, along with detailed descriptions, are provided in Appendix G.4.

### 4.2 REAL-WORLD MULTIMODAL REGRESSION

To rigorously evaluate the effectiveness and interpretability of the proposed PIDReg framework, we conduct comprehensive experiments on six real-world datasets spanning diverse domains, including healthcare, physics, affective computing, and robotics, covering both univariate and multivariate prediction tasks. For empirical comparison, PIDReg is evaluated against state-of-the-art multimodal learning methods, including MIB (Mai et al., 2022), MoNIG (Ma et al., 2021), MEIB (Zhang et al., 2022), and DER (Amini et al., 2020). For large-scale datasets, we additionally compare with CoMM (Dufumier et al., 2025), a recent method that incorporates PID into multimodal contrastive learning. The performance of different approaches is evaluated on the test set using Root Mean Square Error (RMSE) and Pearson Correlation Coefficient (Corr). Experimental details are provided in the Appendix E.

***CT Slices*** (Graf & Cavallaro, 2011) is a medical imaging dataset that integrates two modalities derived from 53,500 CT slices across 74 patients: bone structure histograms (240 features) and air inclusion histograms (144 features). The regression target is the axial position along the cephalo-caudal axis, ranging from 0 (cranial vertex) to 180 (plantar surface). The dataset is split into 70% training, 10% validation, and 20% test sets. As shown in Table 1, our PIDReg achieves the lowest RMSE and highest correlation.

| *Metric* | MIB | MoNIG | MEIB | DER | PIDReg |
|---|---|---|---|---|---|
| RMSE↓ | 1.801 | 1.490 | 1.258 | 0.847 | **0.626** |
| Corr↑ | 0.997 | 0.996 | 0.999 | **1.000** | **1.000** |

| *Metric* | MIB | MoNIG | MEIB | DER | PIDReg |
|---|---|---|---|---|---|
| RMSE↓ | 15.18 | 14.59 | 14.04 | 12.37 | **10.37** |
| Corr↑ | 0.907 | 0.913 | 0.917 | 0.936 | **0.952** |

Table 1: CT Slice (left) and Superconductivity (right) regression (best results are shown in **bold**; second-best results are underlined. The same convention applies in all subsequent tables).

***Superconductivity*** (Hamidieh, 2018) is a superconductivity dataset from physics, containing 21,263 material samples, each represented by two modalities: an 81-dimensional vector of chemical properties and an 86-dimensional vector derived from chemical formulas. The regression target is the superconducting critical temperature, ranging from 0 to 185. As shown in Table 1, our PIDReg achieves the lowest RMSE and highest correlation, further demonstrating its effectiveness.

***CMU-MOSI*** (Zadeh et al., 2016) and ***CMU-MOSEI*** (Zadeh et al., 2018) contain 2,199 and 23,454 human-annotated sentiment labels, respectively, derived from short monologues and movie review video clips collected from YouTube. Both datasets provide three pre-extracted modalities: audio (A), text (T), and vision (V). The target variable is sentiment, represented as a continuous value in the range $[-3, 3]$. In each experiment, we select two modalities as input. Following the evaluation protocols in Pham et al. (2019); Liang et al. (2018), model performance is measured using 7-class accuracy (Acc7), binary accuracy (Acc2), F1-score, mean absolute error (MAE), and Corr. As shown in Table 2, our PIDReg consistently outperforms all baselines with Audio&Text and Visual&Text modalities, and achieves the second-best performance with the Audio&Visual modalities, where all competing methods exhibit a performance drop.

| *Method* | *CMU-MOSI* | | | | | *CMU-MOSEI* | | | | |
|---|---|---|---|---|---|---|---|---|---|---|
| | $A_7$ ↑ | $A_2$ ↑ | F1 ↑ | MAE ↓ | Corr ↑ | $A_7$ ↑ | $A_2$ ↑ | F1 ↑ | MAE ↓ | Corr ↑ |
| MIB$^\heartsuit$ | 28.9 | 70.7 | 70.8 | 1.088 | 0.578 | 45.8 | 78.9 | 77.9 | 0.736 | 0.653 |
| MoNIG$^\heartsuit$ | 31.9 | 79.1 | 79.1 | 0.976 | **0.671** | 43.1 | 79.2 | 79.0 | 0.687 | 0.603 |
| MEIB$^\heartsuit$ | 23.9 | 60.6 | 60.5 | 1.246 | 0.415 | 40.3 | 62.4 | 64.0 | 0.789 | 0.414 |
| DER$^\heartsuit$ | 30.9 | 78.5 | 78.4 | 1.086 | 0.637 | **48.5** | 79.9 | **80.0** | 0.637 | 0.655 |
| PIDReg$^\heartsuit$ | **32.0** | **80.0** | 79.7 | **0.938** | 0.662 | 47.4 | **80.2** | **80.0** | **0.634** | **0.662** |
| MIB$^\spadesuit$ | 27.4 | 73.2 | 73.3 | 1.092 | 0.601 | 46.7 | 79.4 | 78.8 | 0.733 | 0.656 |
| MoNIG$^\spadesuit$ | 29.9 | 78.1 | 78.0 | 1.046 | 0.627 | 44.2 | 80.2 | **80.2** | 0.680 | 0.606 |
| MEIB$^\spadesuit$ | 24.2 | 60.1 | 60.8 | 1.301 | 0.374 | 41.8 | 63.2 | 64.8 | 0.777 | 0.424 |
| DER$^\spadesuit$ | 33.2 | 80.5 | 80.7 | 0.969 | **0.666** | 46.6 | 79.8 | **80.2** | 0.651 | 0.630 |
| PIDReg$^\spadesuit$ | **37.2** | **80.8** | **80.9** | **0.947** | 0.664 | **47.0** | **80.6** | **80.2** | **0.642** | **0.661** |
| MIB$^\diamondsuit$ | 14.4 | 47.7 | 40.7 | 1.511 | 0.146 | 41.4 | 62.2 | 63.2 | 1.004 | 0.149 |
| MoNIG$^\diamondsuit$ | 15.9 | **52.7** | 55.6 | 1.428 | **0.224** | **42.5** | **65.6** | **67.0** | **0.808** | **0.262** |
| MEIB$^\diamondsuit$ | 15.2 | 45.0 | 52.2 | 1.478 | 0.152 | 41.7 | 53.9 | 58.1 | 0.825 | 0.178 |
| DER$^\diamondsuit$ | 15.7 | 44.5 | **58.5** | 1.476 | 0.212 | 41.4 | 63.2 | 65.6 | 0.827 | 0.185 |
| PIDReg$^\diamondsuit$ | **16.4** | 52.3 | 51.8 | **1.400** | 0.149 | 41.7 | 63.4 | 63.7 | 0.828 | 0.228 |

Table 2: Human sentiment analysis on CMU-MOSI and CMU-MOSEI (modality combinations are represented by symbols: $\heartsuit$ (Audio-Text), $\spadesuit$ (Visual-Text), and $\diamondsuit$ (Audio-Visual).

***Vision&Touch*** (Lee et al., 2020) is a large-scale raw multimodal robotics dataset consisting of 150 trajectories of a triangular-peg-insertion task using a 7-DoF Franka Emika Panda robot. Each trajectory includes 1,000 synchronized time steps of visual, haptic, and proprioceptive signals. Following Liang et al. (2021), we formulate a regression task where selected proprioceptive dimensions at time $t + 1$ are predicted from other modalities at time $t$. Specifically, we use a four-dimensional

| Metric | MIB | MoNIG | MEIB | DER | CoMM | PIDReg |
|---|---|---|---|---|---|---|
| MSE↓ (×10⁻⁴) | 3.00 | 3408 | 6.19 | 498 | **1.34** | 1.53 |
| Corr*↑ | 0.97 | 0.82 | 0.96 | 0.85 | **0.98** | **0.98** |

| Metric | MIB | MoNIG | MEIB | DER | CoMM | PIDReg |
|---|---|---|---|---|---|---|
| MAE↓ | 6.75 | 8.70 | 7.83 | 9.96 | 9.46 | **6.29** |
| Corr↑ | 0.64 | 0.59 | 0.65 | 0.54 | 0.27 | **0.75** |

Table 3: Vision&Touch (left) and Brain-Age regression (right).

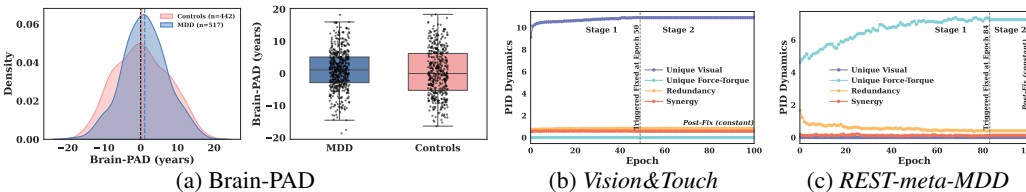

(a) Brain-PAD  (b) *Vision&Touch*  (c) *REST-meta-MDD*

Figure 3: (a) Bias-corrected predicted age difference; (b, c) convergence curves of PID components.

subset $(x, y, z, \text{yaw})$ that represents the end-effector's spatial position and yaw angle, thereby forming a *multivariate* prediction problem. To obtain modality-specific embeddings, we follow Dufumier et al. (2025): visual inputs are processed using a ResNet-18 backbone pretrained on ImageNet, while force/torque sequences are encoded using a five-layer causal convolutional network (Bai et al., 2018) applied directly to the raw sensor readings. The results are shown in Table 3, evaluated via MSE and RV coefficient (Corr*) (Robert & Escoufier, 1976).

***REST-meta-MDD*** (Yan et al., 2019) is the largest multimodal neuroimaging dataset for major depressive disorder (MDD), comprising 848 MDD patients and 794 healthy controls from 17 hospitals across China, along with metadata such as age and gender. We use $T_1$-weighted sMRI and resting-state fMRI (rs-fMRI) as two input modalities to predict brain age. For 3D sMRI volumes, we use a 3D CNN encoder with channel and spatial attention. For rs-fMRI, we apply a Graph Isomorphism Network (GIN) (Xu et al., 2019) enhanced with graph attention and hierarchical pooling. We conduct nine groups of experiments, where in each group, data from 15 hospitals are randomly selected for training and the remaining 2 hospitals are used for testing, ensuring that each test site contains more than 100 samples to avoid biased evaluation. The results in Table 3 are averaged across all groups.

Fig. 3(a) shows the histogram of predicted age difference (PAD), defined as the deviation between the predicted and true chronological age, for patients and healthy controls, following standard linear bias correction (Smith et al., 2019). PAD is commonly regarded as a robust biomarker for psychiatric diagnosis, as patients with conditions such as Alzheimer's disease (Cole et al., 2017; Ly et al., 2020) and MDD (Han et al., 2021) often exhibit accelerated brain aging, resulting in a larger PAD. Our findings are consistent with existing medical evidence (Han et al., 2021; Luo et al., 2022). The joint use of structural and functional brain connectivity offers a new way for brain age estimation.

## 4.3 INTERPRETABILITY ANALYSIS

| Comp. | CT | | SC | | MOSI♡ | | MOSI♠ | | MOSI◇ | | MOSEI♡ | | MOSEI♠ | | MOSEI◇ | | V&T | | MDD | |
|---|---|---|---|---|---|---|---|---|---|---|---|---|---|---|---|---|---|---|---|---|
| | Bone† | Air‡ | prop.† | form.‡ | A† | T‡ | V† | T‡ | A† | V‡ | A† | T‡ | V† | T‡ | A† | V‡ | Visual† | Touch‡ | fMRI† | sMRI‡ |
| $U_{Z_1}$ | 0.045 | | 0.375 | | 0.103 | | 0.053 | | 0.023 | | 0.021 | | 0.023 | | 0.012 | | 10.90 | | 0.000 | |
| $U_{Z_2}$ | | 0.067 | | 0.001 | | 0.209 | | 0.058 | | 0.152 | | 0.025 | | 0.030 | | 0.042 | | 0.000 | | 7.240 |
| $R$ | 1.675 | | 0.878 | | 9.147 | | 1.375 | | 9.148 | | 0.192 | | 0.318 | | 0.206 | | 0.824 | | 0.412 | |
| $S$ | 1.147 | | 0.394 | | 0.312 | | 4.228 | | 0.331 | | 0.298 | | 0.690 | | 0.290 | | 0.575 | | 0.140 | |

Table 4: Gaussian PID convergence values († and ‡ indicate $X_1$ and $X_2$, respectively).

Figs. 3(b) and 3(c) depict the learning dynamics of each PID term over the entire training process for the V&T and MDD datasets. Table 4 summarizes the final converged values of the PID components. In **CT Slice** (CT), bone-density and air-content histograms encode nearly identical information about axial position, and thus redundancy dominates. In **Superconductivity** (SC), elemental physicochemical vectors provide the primary unique signal, while formula strings add little. This aligns with (Stanev et al., 2018), which highlights composition-aware features as key to $T_c$ prediction.

In both **MOSI** and **MOSEI**, the presence of both high redundancy and high synergy across modality combinations suggests that sentiment analysis should not rely on a single modality alone. Moreover, Vision+Text is consistently dominated by synergy, as facial cues help disambiguate linguistic content. For example, (Castro et al., 2019) found that visual cues, such as neutral facial expressions or eye rolls, are critical for detecting sarcasm that cannot be captured by text alone. In contrast, Audio+Text typically exhibits notable redundancy. This aligns with the strong coupling between language and audio via word intonation (Zadeh et al., 2018); e.g., prosody often reflects the emotional valence of lexical content.

In **Vision&Touch** (V&T), vision serves as the primary predictive engine, while tactile signals contribute little unique or synergistic information. This finding aligns with the visuo-tactile ablation study (Lee et al., 2019), suggesting that the visual modality alone may be sufficient to achieve reliable predictive performance. In **REST-meta-MDD** (MDD), sMRI is the dominant modality for brain age prediction, while rs-fMRI contributes minimally. This aligns with clinical evidence (Sun et al., 2024; Jónsson et al., 2019), which shows that sMRI is more informative than fMRI for brain age estimation. It is also consistent with (Cole et al., 2017; Liem et al., 2017), where morphometry alone achieves sub-5-year MAE and multimodal improvements are marginal. We refer interested readers to the Appendix D for additional ablation study results.

## 5 CONCLUSION

We propose PIDReg, a framework that seamlessly integrates PID into multimodal regression to improve both prediction accuracy and interpretability. PIDReg identifies the individual contributions of each modality and determines whether their interaction is dominated by redundancy or synergy. It is applicable to a wide range of data types, including vector data, 3D volume images, and graph-structured data, spanning diverse application domains. In particular, our results in brain age prediction align with current clinical evidence, highlighting the strong potential of PIDReg in biomedical science. PIDReg can be extended to three or more modalities, as detailed in the Appendix F. Limitations and future work are discussed in Appendix H.

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

The appendix is organized into the following topics and sections:

## Outline

## A  DESIGN MOTIVATIONS OF PIDREG

### A.1  INTUITIVE EXPLAINATION OF THE UNION INFORMATION

In Definition 1, the union information

$$\widetilde{U}(Y : Z_1; Z_2) = \min_{Q \in \Delta_P} I_Q(Y; Z_1, Z_2) \tag{24}$$

is the smallest possible mutual information between $Y$ and the pair $(Z_1, Z_2)$ among all joint distributions $Q$ that keep the two marginals $P_{YZ_1}$ and $P_{YZ_2}$ fixed.

Union information asks: What is the amount of information about $Y$ that both modalities must share, no matter how we "glue together" $Z_1$ and $Z_2$, as long as each preserves its own relationship to $Y$?

- If both modalities contain the same evidence about $Y$, then every coupling of $Z_1$ and $Z_2$ consistent with the marginals must preserve that shared evidence. Consequently, the minimization still yields a large value. This quantity is what PID identifies as *redundancy*.

- Conversely, if the modalities provide different, complementary evidence about $Y$, then one can construct a coupling $Q \in \Delta_P$ in which the two modalities behave independently given $Y$, causing the mutual information to drop. In this case, the union information (and thus redundancy) becomes small, leaving the remaining information to be attributed to the *unique* components.

In short, union information provides an *operational lower bound on "shared evidence"* that cannot be removed by any permissible coupling, which is why it serves as the redundancy term in PID.

## A.2 GAUSSIAN ASSUMPTION IN THE LATENT SPACE

PIDReg does not impose any distributional assumptions on the original, complex input data $X_1$, $X_2$, or $Y$, fully accommodating real world phenomena such as heavy-tailedness and skewness. Instead, the Gaussian assumption is imposed in the joint latent space formed by the transformed representations $Z_1$, $Z_2$ (obtained via the deep nonlinear encoders $h_{\phi_m}$) together with the transformed target variable $Y$. Note that, this $Y$ refers to the rank-based, outlier-aware inverse normal transformation of the target (rather than the raw $Y$), a procedure commonly adopted in neuroscience applications.

The role of the encoder $h_{\phi_m}$ is precisely to extract task-relevant, more structured, and compact information from raw inputs that may follow arbitrarily complex distributions. This design is conceptually aligned with modern generative models such as variational autoencoders (VAEs) (Kingma et al., 2019), which enforce a simple prior distribution (e.g., Gaussian) in the latent space to achieve regularization, disentanglement, and generative capability. In PIDReg, the Gaussian assumption serves as an inductive bias to guide the learning process, rather than as a rigid constraint.

We further introduce two differentiable regularization terms, designed based on the CS divergence, which actively guide the system toward Gaussianity. Notably, this active regularization mechanism is independent of any distributional assumption on the raw data itself.

In particular, our regularization based on the CS divergence is inspired by the well-known MMD-VAE (Zhao et al., 2019), where we replace the MMD with CS divergence to match the aggregated posterior $p(z)$ with a Gaussian prior $q(z)$. Compared to the traditional VAE regularization term $\mathbb{E}_{p(x)}[\mathrm{KL}(p_\phi(z|x)\|q(z))]$, the CS-based (or MMD-based) penalty $\mathrm{CS}(p_\phi(z), q(z))$ is operated on marginals and less restrictive, and notably helps mitigate the problem of uninformative latent codes, as discussed in Zhao et al. (2019).

## A.3 THE MOTIVATION OF CS DIVERGENCE

A central challenge in comparing probability distributions lies in selecting a divergence measure that is both theoretically sound and practically robust. Classical approaches such as the MMD and KL divergence divergence represent two dominant paradigms: the integral probability metric (IPM) family and the $f$-divergence family, respectively. However, both exhibit critical limitations when applied to empirical distributions with limited support overlap or when robustly measuring distance between complex, high-dimensional densities.

### A.3.1 CS DIVERGENCE AGAINST KL DIVERGENCE: STABILITY AND SYMMETRY

First, although both the KL divergence and the CS divergence can be employed to measure the difference or similarity between two entities (such as probability distributions or vectors), the CS divergence is considerably more stable than the KL divergence in that it relaxes the constraints on the supports of the distributions (Yu et al., 2024a). For any two densities $p$ and $q$, $D_{\mathrm{KL}}(p; q)$ has finite values only if $\mathrm{supp}(p) \subseteq \mathrm{supp}(q)$ (otherwise, $p(x) \log\left(\frac{p(x)}{0}\right) \to \infty$); whereas $D_{\mathrm{KL}}(q; p)$ has finite values only if $\mathrm{supp}(q) \subseteq \mathrm{supp}(p)$. In contrast, $D_{\mathrm{CS}}(p; q)$ is symmetric and always yields finite values unless the supports of $p$ and $q$ have no overlap, i.e., $\mathrm{supp}(p) \cap \mathrm{supp}(q) = \emptyset$ (see Fig. 4 for an illustration).

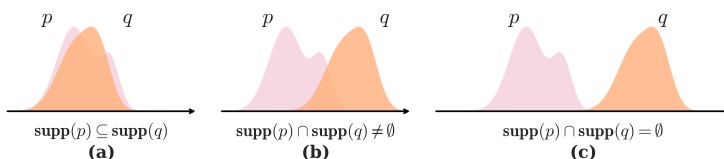

Figure 4: (a) $D_{KL}(p;q) \to \infty$, $D_{KL}(q;p) \to \infty$, $D_{CS}(p;q) \to \infty$; (b) $D_{KL}(p;q) \to \infty$, $D_{KL}(q;p) \to \infty$, $D_{CS}(p;q)$ finite; (c) $D_{KL}(p;q) \to \infty$, $D_{KL}(q;p) \to \infty$, $D_{CS}(p;q)$ finite.

Second, CS divergence is symmetric, eliminating the need to choose between $D_{\text{CS}}(p(z); \mathcal{N}(0, I))$ and $D_{\text{CS}}(\mathcal{N}(0, I); p(z))$.

### A.3.2 CS DIVERGENCE AGAINST MMD: EFFICIENCY IN ESTIMATOR

MMD measures the difference in kernel mean embeddings:

$$\text{MMD}^2(P, Q) = \left\| \mu_P - \mu_Q \right\|_{\mathcal{H}}^2, \tag{25}$$

where $\mu_P$ and $\mu_Q$ are the kernel mean embeddings of distributions $P$ and $Q$ in the reproducing kernel Hilbert space $\mathcal{H}$.

By contrast, the empirical CS divergence estimator reduces to the cosine similarity between kernel-mean embeddings of two sample sets (Yu et al., 2024b):

$$\widehat{D}_{\text{CS}}(p; q) = -2 \log \left( \frac{\langle \mu_p, \mu_q \rangle_{\mathcal{H}}}{\|\mu_p\|_{\mathcal{H}} \|\mu_q\|_{\mathcal{H}}} \right) = -2 \log \cos(\mu_p, \mu_q). \tag{26}$$

While MMD admits closed-form estimators with clear physical interpretations when comparing marginal distributions $P$ and $Q$, its extension to conditional distributions $p(Y|X)$ remains both theoretically unsettled and practically cumbersome: RKHS conditional-embedding (and related operator-based) approaches hinge on the conditional covariance operator:

$$C_{Y|X} = C_{YX} \left( C_{XX} + \lambda I \right)^{-1}, \tag{27}$$

which presupposes invertibility of the (uncentered) covariance $C_{XX}$ and idealized stationarity—assumptions that routinely fail in high-dimensional or finite-sample regimes—and, more broadly, no universally accepted estimator for conditional MMD has emerged, forcing costly kernel-matrix inversions or heavy regularization (Song et al., 2009; 2013; Park & Muandet, 2020; Ren et al., 2016).

In contrast, the CS divergence yields equally concise sample-based expressions for both marginal- and conditional-distributions comparison under far weaker hypotheses and with straightforward computation, motivating our choice to adopt CS divergence for simultaneous measurement of marginal and conditional distribution discrepancies.

## B    PROOF OF PROPOSITION ON CONDITIONAL MUTUAL INFORMATION ESTIMATOR

### B.1    DEFINITION

Consider an unknown but fixed joint distribution

$$p(X_1, X_2, Z_1), \tag{28}$$

from which we draw $N$ observations $\{(x_{1i}, x_{2i}, z_{1i})\}_{i=1}^N$, where $x_{1i} \in \mathbb{R}^{d_1}$, $x_{2i} \in \mathbb{R}^{d_2}$, and $z_{1i} \in \mathbb{R}^d$. We are interested in the following CS divergence between two product distributions constructed from $p(X_1, X_2, Z_1)$:

$$D_{\text{CS}}\big(p(X_1, X_2, Z_1)\, p(X_1);\ p(X_1, X_2)\, p(X_1, Z_1)\big). \tag{29}$$

A more explicit expression of this divergence is:

$$
\begin{aligned}
D_{\mathrm{CS}}&\big(p(X_1, X_2, Z_1)\, p(X_1) \,;\; p(X_1, X_2)\, p(X_1, Z_1)\big) \\
&= -2 \log\!\Big(\int p(X_1, X_2, Z_1)\, p(X_1)\, p(X_1, X_2)\, p(X_1, Z_1)\Big) \\
&\quad + \log\!\Big[\big(\int p^2(X_1, X_2, Z_1)\, p^2(X_1)\big)\big(\int p^2(X_1, X_2)\, p^2(X_1, Z_1)\big)\Big].
\end{aligned}
\tag{30}
$$

In section B.2, we provide a kernel-based empirical estimator for Eq.( 30).

## B.2   ESTIMATOR

**Approximation of the first integral.**   We start with the integral:

$$
\int (p(X_1, X_2, Z_1)\, p(X_1);\; p(X_1, X_2)\, p(X_1, Z_1),
\tag{31}
$$

which can be written as:

$$
\mathbb{E}_{p(X_1, X_2, Z_1)}\big[\, p(X_1)\, p(X_1, X_2)\, p(X_1, Z_1)\,\big].
\tag{32}
$$

Given the $N$ i.i.d. samples $\{(x_{1i},\ x_{2i},\ z_{1i})\}_{i=1}^N$, a simple Monte Carlo approximation yields:

$$
\frac{1}{N} \sum_{j=1}^N p\big(x_{1j}\big)\, p\big(x_{1j},\ x_{2j}\big)\, p\big(x_{1j},\ z_{1j}\big).
\tag{33}
$$

Next, we approximate each density term by a Gaussian kernel estimator. For instance,

$$
p\big(x_{1j}, z_{1j}\big) \approx \frac{1}{N\,(\sqrt{2\pi}\,\sigma)^{d_1+d}} \sum_{i=1}^N \exp\!\Big(-\frac{\|x_{1j}-x_{1i}\|^2}{2\,\sigma^2}\Big)\, \exp\!\Big(-\frac{\|z_{1j}-z_{1i}\|^2}{2\,\sigma^2}\Big),
\tag{34}
$$

with analogous forms for $p(x_{2j}, z_{1j})$ and $p(z_{1j})$. Substituting these estimates into Eq.( 33) and expanding the sums leads to a triple sum.

To simplify notation, define the following Gram (kernel) matrices:

- $M \in \mathbb{R}^{N\times N}$, the Gram matrix for $X_1$, with
$$
M_{ji} = \exp\!\Big(-\tfrac{\|x_{1j}-x_{1i}\|^2}{2\,\sigma^2}\Big).
$$

- $K \in \mathbb{R}^{N\times N}$, the Gram matrix for $X_2$, with
$$
K_{ji} = \exp\!\Big(-\tfrac{\|x_{2j}-x_{2i}\|^2}{2\,\sigma^2}\Big).
$$

- $L \in \mathbb{R}^{N\times N}$, the Gram matrix for $Z_1$, with
$$
L_{ji} = \exp\!\Big(-\tfrac{\|z_{1j}-z_{1i}\|^2}{2\,\sigma^2}\Big).
$$

In these terms, the integral Eq.( 31) can be approximated as (up to a constant factor involving $N$ and $\sigma$):

$$
\begin{aligned}
\int &p(X_1, X_2, Z_1)\, p(X_1)\, p(X_1, X_2)\, p(X_1, Z_1) \\
&\approx \frac{1}{N^4(\sqrt{2\pi}\,\sigma)^{d_1+d_2+3d}} \ \times \sum_{j=1}^N \Big(\sum_{i=1}^N M_{ji}\Big) \Big(\sum_{i=1}^N K_{ji} M_{ji}\Big) \Big(\sum_{i=1}^N L_{ji} M_{ji}\Big).
\end{aligned}
\tag{35}
$$

**Approximation of the remaining integrals.** Similarly, we evaluate the two terms inside the large bracket of the CSn divergence:

- For $\int p^2(X_1, X_2, Z_1)\, p^2(Z_1)$ the same procedure yields:

$$\int p^2(X_1, X_2, Z_1)\, p^2(Z_1) \approx \frac{1}{N^4\, (\sqrt{2\pi}\,\sigma)^{d_1+d_2+3d}} \sum_{j=1}^{N} \Big(\sum_{i=1}^{N} K_{ji}\, L_{ji}\, M_{ji}\Big) \Big(\sum_{i=1}^{N} L_{ji}\Big)^2. \quad (36)$$

- For $\int p^2(X_1, Z_1)\, p^2(X_2, Z_1)$ we obtain:

$$\int p^2(X_1, Z_1)\, p^2(X_2, Z_1) \approx \frac{1}{N^4\, (\sqrt{2\pi}\,\sigma)^{d_1+d_2+3d}} \sum_{j=1}^{N} \left[ \frac{\left(\sum_{i=1}^{N} K_{ji}\, L_{ji}\right)^2 \left(\sum_{i=1}^{N} L_{ji}\, M_{ji}\right)^2}{\sum_{i=1}^{N} K_{ji}\, L_{ji}\, M_{ji}} \right]. \quad (37)$$

**Final Format.** By combining these three approximations Eq.( 35), Eq.( 36) and Eq.( 37), and omitting the common normalization factor, we obtain the following empirical estimator for Eq.( 30):

$$\widehat{D}_{\mathrm{CS}}\big(p(X_1, X_2, Z_1)\, p(X_1)\; ;\; p(X_1, X_2)\, p(X_1, Z_1)\big) =$$

$$-2\, \log\Big(\sum_{j=1}^{N} \big(\sum_{i=1}^{N} M_{ji}\big) \big(\sum_{i=1}^{N} K_{ji} M_{ji}\big) \big(\sum_{i=1}^{N} L_{ji} M_{ji}\big)\Big)$$

$$+ \log\Big(\sum_{j=1}^{N} \big(\sum_{i=1}^{N} K_{ji} L_{ji} M_{ji}\big) \big(\sum_{i=1}^{N} L_{ji}\big)^2\Big) + \log\Big(\sum_{j=1}^{N} \frac{\big(\sum_{i=1}^{N} K_{ji} L_{ji}\big)^2 \big(\sum_{i=1}^{N} L_{ji} M_{ji}\big)^2}{\sum_{i=1}^{N} K_{ji} L_{ji} M_{ji}}\Big). \quad (38)$$

$\square$

# C  ALGORITHM

The pipeline of PIDReg is illustrated in Algorithm 1. In Stage 1, we initialize modality-specific encoders and employ a Gaussian PID-guided procedure to iteratively refine the weighting parameters, ensuring stable contributions from each modality. Once convergence is achieved, we fix these weights and move to Stage 2, where the network is fine-tuned under the current-static weighting scheme. Throughout the process, a parallel optimization step evaluates and updates the fused representation, ultimately yielding a robust, interpretable prediction.

Regarding the PID-converged check in Algorithm 1, it is defined as follows: let $G^t = \{U_{Z_1}^t, U_{Z_2}^t, R^t, S^t\}$ denote the average of the four parameters computed for each batch in the $i$-th iteration. Subsequently, we apply:

$$\delta^t = \left\| G^t - G^{t-1} \right\|_\infty < \epsilon, \quad (39)$$

when $\delta^t < \epsilon$ for $K$ consecutive epochs, we establish $\mathbf{w}^* = \mathbf{w}^t$ as the optimal fusion weights. In practice, we set $K = 5$ and $\epsilon = 0.01$ to ensure stable convergence across stochastic minibatches while avoiding unnecessary training overhead.

# D  ABLATION STUDY

## D.1  REGULARIZATION COMPONENT ABLATION

In our work, we introduce three regularization terms, $\mathcal{L}_{\mathrm{CS}}$, $\mathcal{L}_{\mathrm{Gauss}}$, and $\mathcal{L}_{\mathrm{CMI}}$, which serve two distinct purposes. The $\mathcal{L}_{\mathrm{CS}}$ and $\mathcal{L}_{\mathrm{Gauss}}$ terms impose constraints on the marginal and joint distributions, respectively, to ensure that the resulting latent representation $P(Z_1, Z_2, Y)$, closely approximates a

---

**Algorithm 1** PIDReg Algorithm

---

**Require:** Multimodal inputs $\{X_1, X_2\}$, target $Y$
**Ensure:** Prediction $\hat{y}$, information decomposition $\{U_{Z_1}, U_{Z_2}, R, S\}$
1: Initialize encoders $h_{\phi_m}$, predictor $f_\theta$, learnable IB parameters $\lambda_m^b$, fusion weights $\mathbf{w} = [w_1, w_2, w_3]$ and PID convergence flag $Converged \leftarrow false$
2: **while** not converged **do**
3:     **for** each batch **do**
4:         **for** $m \in \{1, 2\}$ **do**
5:             $R_m = h_{\phi_m}(X_m)$                             ▷ Raw Representations
6:             $Z_m = \lambda_m^b \cdot R_m + (1 - \lambda_m^b) \cdot \mathcal{N}(\mu_{R_m}, \sigma_{R_m}^2)$       ▷ IB Mechanism
7:             $\mathcal{L}_{CS} = \sum_{i=1}^2 \widehat{D}_{CS}\big(p(z_i); \mathcal{N}(0, I)\big)$         ▷ Marginal Gaussianity
8:         **end for**
9:         $\mathcal{L}_{cmi} = \widehat{I}(Z_1; X_2 \mid X_1) + \widehat{I}(Z_2; X_1 \mid X_2)$     ▷ Unique Information Guarantee
10:         **if** not PID-converged **then**
11:             $\Sigma^P \leftarrow$ Estimate covariance matrix for $(Z_1, Z_2, Y)$
12:             $\{U_{Z_1}, U_{Z_2}, R, S\} \leftarrow$ Gaussian PID decomposition of $\Sigma^P$
13:             $\mathbf{w} \leftarrow [\frac{U_{Z_1}+R}{T}, \frac{U_{Z_2}}{T}, \frac{S}{T}]^T$ where $T = U_{Z_1} + U_{Z_2} + R + S$
14:             $\mathcal{L}_{Gauss} =$ Gaussian normality deviation of $(Z_1, Z_2, Y)$
15:             Check PID convergence criteria, update convergence flag if stable
16:         **end if**
17:         $Z = w_1 Z_1 + w_2 Z_2 + w_3 (Z_1 \odot Z_2)$         ▷ Information-weighted Fusion
18:         $\hat{y} = f_\theta(Z)$                                    ▷ Prediction
19:         $\mathcal{L}_{pred} = MSE(\hat{y}, y)$                     ▷ Prediction Loss
20:         Update network parameters and $\lambda_m^b$ using respective optimizers
21:     **end for**
22:     Validate and adjust learning rates
23: **end while**
24: **return** Trained model with fixed optimal fusion weights $\mathbf{w}^*$

---

Gaussian distribution. In contrast, $\mathcal{L}_{\mathrm{CMI}}$ penalizes information leakage between $Z_1$ and $X_2$ as well as between $Z_2$ and $X_1$, thereby guaranteeing that the feature fusion guided by the PID module's learned weights remains interpretable and faithful. In section D.1, we perform two ablations (section D.1.1, section D.1.2) to evaluate.

### D.1.1   JOINT GAUSSIAN GUARANTEE

Taking the **_Superconductivity_** as an example, the empirical distributions of the raw inputs $X_1$ and $X_2$ are shown in Fig. 5, where one can observe severe skewness and non-Gaussianity. It is worth emphasizing that PIDReg does not rely on any distributional assumptions about the input data itself, rather, it aims to learn a Gaussian representation in the latent space. For visualization, the latent features of $X_1$ and $X_2$ are individually projected into two dimensions using Principal Component Analysis (PCA) with whitening, while the target variable $Y$, being one-dimensional, is directly shown without projection.

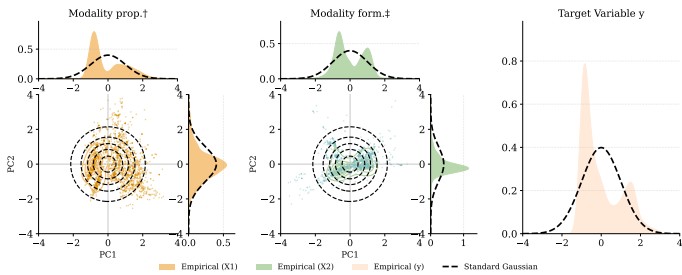

Figure 5: Distributions of $P(X_1)$, $P(X_2)$ and raw $P(Y)$.

To further illustrate how PIDReg learns such a Gaussian system from highly non-Gaussian real world data, and to highlight the role of each Gaussianity regularizer, we ablated one or both Gaussianity regularizers ($\mathcal{L}_{\text{CS}}$, $\mathcal{L}_{\text{Gauss}}$), recorded the joint latent representations $(Z_1, Z_2, Y)$ at the epoch when the PID module converged. To enable consistent visual comparison across different ablation settings and the full-loss baseline, we apply PCA with whitening on the joint representations to obtain two-dimensional projections. The resulting 2D distributions are overlaid with reference circles at $1\sigma$, $2\sigma$, and $3\sigma$ radii of a standard normal distribution. This enables an intuitive assessment of distributional concentration, isotropy, and deviation from Gaussianity, as illustrated in Fig. 6.

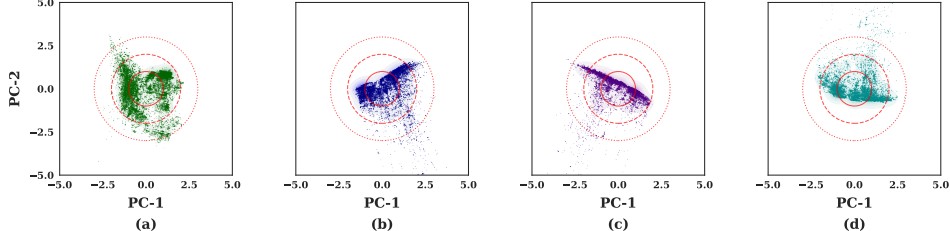

Figure 6: Visualization of the ablation study on joint latent distributions $P(Z_1, Z_2, Y)$ under different loss configurations: (a) full loss; (b) without the $\mathcal{L}_{\text{CS}}$ term; (c) without the $\mathcal{L}_{\text{Gauss}}$ term; (d) without both $\mathcal{L}_{\text{CS}}$ and $\mathcal{L}_{\text{Gauss}}$ terms. —, ⋯⋯ and - - - denote the $1\sigma$, $2\sigma$ and $3\sigma$ contours of a standard Gaussian distribution.

In Fig. 6(a), the full-loss embeddings cluster tightly at the origin. Removing $\mathcal{L}_{\text{CS}}$ (b) shifts the cloud along PC-1 with heavy tails beyond $2\sigma$ and $3\sigma$. Removing $\mathcal{L}_{\text{Gauss}}$ (c) preserves centering but produces a diagonally elongated distribution. Omitting both (d) yields a highly anisotropic cloud with tails scattered well beyond $3\sigma$. $\mathcal{L}_{\text{CS}}$ enforces mean zero and unit covariance, while $\mathcal{L}_{\text{Gauss}}$ corrects skewness and kurtosis; dropping any component causes mean shifts, anisotropic scaling, and non-Gaussian distortions as the encoder focuses on prediction error.

### D.1.2 Unique Information Extraction Guarantee

$\mathcal{L}_{CMI}$ does not act directly on the joint distribution, it ensures that each modality's latent encoding captures only the unique information of that modality, preventing irrelevant cross-modal redundancy from leaking into the other modality's representation. If the $\mathcal{L}_{CMI}$ is effective, then, in theory, $Z_1$ (resp. $Z_2$) should contain no or less information about $X_2$ (resp. $X_1$). Theoretically, consider the conditional mutual information $I(X_1; Z_2 \mid X_2)$. If $I(X_1; Z_2 \mid X_2) \approx 0$, then by the identity $I(X_1; Z_2 \mid X_2) = I(X_1; Z_2, X_2) - I(X_1; X_2)$, we obtain $I(X_1; Z_2, X_2) \approx I(X_1; X_2)$, i.e., the combined pair $(Z_2, X_2)$ provides no additional information about $X_1$ beyond $X_2$ alone. Consequently, the mapping $(Z_2, X_2) \to X_1$ behaves almost identically to $X_2 \to X_1$.

| | Full Loss | | $-\mathcal{L}_{CMI}$ | |
| --- | --- | --- | --- | --- |
| Prediction | MSE↓ | $R^2$↑ | MSE↓ | $R^2$↑ |
| $X_2 \to X_1$ | 0.073 | 0.928 | 0.073 | 0.928 |
| $Z_2 \to X_1$ | 0.670 | 0.334 | 0.155 | 0.846 |
| $(Z_2, X_2) \to X_1$ | 0.089 | 0.911 | 0.089 | 0.912 |
| $X_1 \to X_2$ | 0.619 | 0.380 | 0.619 | 0.380 |
| $Z_1 \to X_2$ | 0.779 | 0.166 | 0.664 | 0.407 |
| $(Z_1, X_1) \to X_2$ | 0.642 | 0.311 | 0.620 | 0.388 |

Table 5: Cross-prediction performance under Full Loss and without $\mathcal{L}_{CMI}$.

To verify this empirically, we record—for both the best-performing full-loss model and the model trained without $\mathcal{L}_{CMI}$—latent codes $Z_1, Z_2$ alongside their corresponding inputs $X_1, X_2$, and apply the cross-prediction schemes listed in Table 5. We then employ identical three-layer MLP probes (256 hidden units, LayerNorm, ReLU, dropout) for each of the six cross-modal regression tasks. The results align with our theoretical analysis: under full loss, $Z_2 \to X_1$ and $Z_1 \to X_2$ yield $R^2 \approx 0.34$

and $0.17$, respectively, indicating minimal information leakage; omitting $\mathcal{L}_{\mathrm{CMI}}$ causes $R^2$ to rise to $0.846$ and $0.407$, demonstrating severe leakage. Moreover, joint regressors $(Z_2, X_2) \to X_1$ and $(Z_1, X_1) \to X_2$ confirm that any leaked information is redundant with each modality's raw input.

The incorporation of $\mathcal{L}_{\mathrm{CS}}$, $\mathcal{L}_{\mathrm{Gauss}}$, and $\mathcal{L}_{\mathrm{CMI}}$ underpins the reliability of the PIDReg framework from both distributional alignment and information-theoretic interpretability perspectives, rendering each component indispensable.

## D.2 Modality Ablation

As presented in the main paper's Interpretability Analysis section, we quantify each modality's contribution to prediction performance via PIDReg on multiple datasets. Notably, on the ***Vision&Touch*** dataset, Visual provides substantial unique information while Touch's unique contribution is nearly zero; similarly, on ***Rest-meta-MDD***, sMRI yields significant unique content whereas fMRI does not. We thus hypothesize that (i) using only Visual on Vision&Touch will match the dual-modality baseline, but using only Touch will degrade performance markedly, and (ii) on Rest-meta-MDD, sMRI alone will suffice, whereas fMRI alone will fail. To test this, we conduct modal ablations by disabling the PID computation and fusion mechanism in PIDReg and instead assigning a fixed weight of 1 to the chosen modality, with all other components held constant.

| Metric | Full | Visual-only | Touch-only | | Metric | Full | sMRI-only | fMRI-only |
|---|---|---|---|---|---|---|---|---|
| MSE↓ ($\times 10^{-4}$) | **1.53** | 1.84 | 93.5 | | MAE↓ | **6.29** | 6.56 | 10.3 |
| Corr* ↑ | **0.98** | **0.98** | 0.05 | | Corr↑ | **0.75** | **0.75** | 0.04 |

Table 6: Modality ablation experiments results: Vision&Touch (left) and Rest-meta-MDD (right).

The results in Table 6 fully validate our hypothesis, demonstrating that the PIDReg-derived information decomposition provides valuable guidance for both modality fusion and modality selection.

## D.3 Linear-Noise Information Bottleneck Ablation

The linear noise injection in the information bottleneck framework, as formulated in Eq.( 1), serves three fundamental purposes in our multimodal learning architecture:

- **Ensuring meaningful computation of information-theoretic terms.** The injection of noise effectively transforms a deterministic mapping $\bar{X} \to Z$ into a stochastic transformation. This stochasticity is essential for enabling meaningful and robust information-theoretic estimation, as purely deterministic mappings can exhibit pathological behavior, including infinite mutual information values under continuous variable settings, as previously established by (Saxe et al., 2019).

- **Enhancing generalization capability.** The Information Bottleneck mechanism has been demonstrated, both empirically and theoretically (Kawaguchi et al., 2023), to discard irrelevant or noisy details in the input $\bar{X}$, thereby improving the model's generalization performance.

- **Facilitating PID computation.** The injection of Gaussian noise into $R_m$ shifts the marginal distribution of $Z_m$ toward a Gaussian distribution, which is a prerequisite for our Gaussian PID component optimization framework.

Rather than manually selecting the noise parameter, $\lambda_m$ is optimized through gradient-based learning. In our implementation, we introduce an unconstrained real-valued latent parameter $\lambda'_m$ and define the bounded noise parameter as:

$$\lambda_m = \sigma(\lambda'_m) = \frac{1}{1 + e^{-\lambda'_m}}, \quad \lambda'_m \in \mathbb{R}, \quad \lambda_m \in (0, 1), \tag{40}$$

where $\sigma$ denotes the sigmoid function. This reparameterization enables the utilization of unconstrained optimizers (e.g., Adam) to train the bounded variable $\lambda_m$ indirectly. The gradient of $\lambda'_m$ is computed with respect to the total loss $\mathcal{L}_{\mathrm{total}}$, yielding the update rule:

$$\lambda'_{m,(t+1)} = \lambda'_{m,(t)} - \eta_\lambda \mathrm{AdamUpdate}\left(\frac{\partial \mathcal{L}_{\mathrm{total}}}{\partial \lambda'_{m,(t)}}\right), \tag{41}$$

where $\eta_\lambda$ represents the learning rate for $\lambda_m$. Through this adaptive mechanism, the contributions of $R_m$ and $\epsilon_m$ are dynamically balanced, enabling the information bottleneck to discover the optimal trade-off between preserving predictive information and satisfying regularization and interpretability constraints.

Table 7 presents the convergence values of $\lambda_m$ across different datasets and modality configurations. The results demonstrate that noise injection consistently plays a significant role in MOSI and MOSEI datasets, particularly for the $X_1$ modality.

| Comp. | CT (Bone†, Air‡) | SC (prop.†, form.‡) | MOSI♡ (A†, T‡) | MOSI♠ (V†, T‡) | MOSI◇ (A†, V‡) | MOSEI♡ (A†, T‡) | MOSEI♠ (V†, T‡) | MOSEI◇ (A†, V‡) | V&T (Visual†, Touch‡) | MDD (fMRI†, sMRI‡) |
|---|---|---|---|---|---|---|---|---|---|---|
| $\lambda_1$ | 0.99 | 0.99 | 0.69 | 0.98 | 0.99 | 0.85 | 0.77 | 0.80 | 0.99 | 0.98 |
| $\lambda_2$ | 0.99 | 0.99 | 0.89 | 0.99 | 0.95 | 0.99 | 0.99 | 0.99 | 0.99 | 0.98 |

Table 7: $\lambda$ convergence values († and ‡ indicate $X_1$ and $X_2$, respectively).

To further substantiate the necessity of the information bottleneck component, we conducted an ablation study wherein $\lambda_m$ was fixed to 1 for both CT Slice and Superconductivity datasets, with all other hyperparameters held constant. The comparative results are presented in Table 8.

| **Metric** | CT | CT($\lambda_m = 1$) | SC | SC($\lambda_m = 1$) |
|---|---|---|---|---|
| RMSE↓ | 0.626 | 0.843 | 10.37 | 10.44 |
| Corr↑ | 1.000 | 0.999 | 0.940 | 0.912 |

Table 8: Linear-noise information bottleneck ablation results.

These experimental findings confirm that even when $\lambda_m$ is set to a relatively large value, the linear noise information bottleneck remains essential for enabling the critical deterministic-to-stochastic transition and maintaining the robustness of the PIDReg framework. The degradation in performance metrics when the information bottleneck is disabled ($\lambda_m = 1$) further substantiates the necessity of this component within our proposed methodology.

The Variational IB (VIB) (Alemi et al., 2016) is an alternative way to impose bottleneck regularization. Standard VIB learns a data-dependent variance $\sigma(x)$ with $Z \mid x \sim \mathcal{N}(\mu(x), \sigma(x)^2 I)$ and optimizes a KL term $D_{\mathrm{KL}}(q(z \mid x) \,\|\, \mathcal{N}(0, I))$ as an upper bound on the mutual information $I(X; Z)$.

We include a controlled comparison with a VIB-style variant (PIDReg-VIB) in the synthetic setting of Section 4.1. We replace our additive linear noise IB with VIB. We report RMSE, $R^2$, the Gaussianity statistic $W$ (from $\mathcal{L}_{\mathrm{Gauss}}$), and the aggregate variance $\sigma^2_{\mathrm{overall}}$ of the PID components:

| Variant | RMSE | $R^2$ | $U_1$ | $U_2$ | $R$ | $S$ | $\sigma^2_{\mathrm{overall}}$ | $W$ |
|---|---|---|---|---|---|---|---|---|
| PIDReg | 10.37 | 0.952 | 0.375 | 0 | 0.878 | 0.394 | 0.089 | 0.974 |
| PIDReg-VIB | 13.57 | 0.848 | 0.385 | 0 | 0.855 | 0.509 | 0.111 | 0.988 |

Table 9: Comparison between PIDReg and VIB-style variant.

Results show that, compared with VIB, our additive noise IB provides better structure preservation while still maintaining a good Gaussian approximation.

We further demonstrate a link between the convergence behavior of $\lambda$ and modality quality. In principle, a noisier input should induce a stronger effective bottleneck regularization. Following the synthetic setup in Section 4.1, we generate

$$X_1 = \tanh\big([R, U_1]W^{(1)} + b^{(1)}\big)W^{(2)} + b^{(2)} + \epsilon_1, \qquad \epsilon_1 \sim \mathcal{N}(0, \sigma_1^2 I_{d_x}), \tag{42}$$

and analogously for $X_2$.

We systematically vary $\sigma_1 = \sigma_2 \in \{0.1, 0.2, 0.3, 0.5, 0.7, 1.0, 1.2, 1.5\}$ under the setting $w_{u1} = 0$, $w_{u2} = 0$, $w_s = 0.75$, $w_r = 0.25$, and repeat the experiments. Results in Table 10 show that as input

noise increases, the learned bottleneck responds in a modality-aware manner: noisier inputs induce a stronger bottleneck, as reflected by the decreasing $\lambda$ values.

| Noise | $\lambda_1$ | $\lambda_2$ | RMSE | $R^2$ | $U_1$ | $U_2$ | $R$ | $S$ |
|---|---|---|---|---|---|---|---|---|
| 0.1 | 0.921 | 0.918 | 0.191 | 0.936 | 0.040 | 0.025 | 0.239 | 0.335 |
| 0.2 | 0.917 | 0.898 | 0.338 | 0.793 | 0.034 | 0.025 | 0.286 | 0.655 |
| 0.3 | 0.904 | 0.897 | 0.494 | 0.604 | 0.026 | 0.028 | 0.281 | 0.665 |
| 0.5 | 0.903 | 0.890 | 0.668 | 0.277 | 0.020 | 0.029 | 0.278 | 0.673 |
| 0.7 | 0.903 | 0.887 | 0.774 | 0.029 | 0.033 | 0.025 | 0.265 | 0.677 |
| 1.0 | 0.844 | 0.834 | 0.807 | -0.057 | 0.021 | 0.038 | 0.268 | 0.673 |
| 1.2 | 0.842 | 0.819 | 0.798 | -0.033 | 0.033 | 0.029 | 0.263 | 0.675 |
| 1.5 | 0.788 | 0.798 | 0.801 | -0.041 | 0.024 | 0.035 | 0.265 | 0.676 |

Table 10: Effect of input noise on learned bottleneck coefficients and PID estimates.

# E  EXPERIMENTAL DETAILS

## E.1  RAW FEATURE ENCODER ARCHITECTURE

All comparative experiments in this paper adopt a consistent encoder architecture to ensure fairness of comparison. The specific encoder used for each dataset is detailed as follows:

For both *CT Slice* and *Superconductivity*, each raw feature encoder is a symmetric three-stage MLP that successively compresses the original $D$-dimensional input into a $d_m$-dimensional embedding via $D \rightarrow HL \rightarrow \frac{HL}{2} \rightarrow d_m$, where $HL$ is the hidden-layer dimensionality (a tunable hyperparameter controlling model capacity). Each linear projection is followed by `BatchNorm1d`, in-place `ReLU`, and dropout (with dropout rate $p = 0.3$ after the first layer and $p = 0.2$ after the second), and the final $d_m$-dimensional output is again normalized via `BatchNorm1d` to stabilize feature statistics. This uniform, modality-agnostic design yields compact embeddings, ensures robust gradient flow, and mitigates overfitting.

For both *CMU-MOSI*, *CMU-MOSEI*, following Pennington et al. (2014); Degottex et al. (2014); Ma et al. (2021), we employ an identical two-stage raw feature encoder that maps the input sequence $\mathbf{X} \in \mathbb{R}^{T \times d_{\text{in}}}$ for each modality (text, audio, vision) to a compact, context-aware embedding. First, a point-wise convolution projects each time step via

$$\mathbf{U} = \text{Dropout}\big(\text{LeakyReLU}\big(\text{BatchNorm1d}\big(\text{Conv1d}_{d_{\text{in}} \rightarrow d_m}(\mathbf{X})\big)\big)\big). \tag{43}$$

Next, we transpose $\mathbf{U}$ to shape $(T, \text{batch}, d_m)$, add fixed sinusoidal positional embeddings, and feed the sum into $L$ stacked Transformer-encoder layers—each comprising $H$-head self-attention, residual connections, layer normalization, and a two-layer feed-forward network. Finally, the raw embedding is obtained by extracting the feature vector at the last time step, $\mathbf{h}_T \in \mathbb{R}^{d_m}$.

For *Vision&Touch*, we employ two modality-specific raw encoders that map the native sensor readings directly into feature vectors prior to information bottleneck mechanism: **Vision Encoder** uses a ResNet-18 pretrained on ImageNet (He et al., 2016) (truncated before the final classification layer): Conv1 ($7 \times 7$ kernel, stride 2, padding $3 \rightarrow 64$ channels; BatchNorm; ReLU), MaxPool ($3 \times 3$, stride 2), four residual stages (each with two basic blocks of conv $3 \times 3 \rightarrow$ BatchNorm $\rightarrow$ ReLU and downsampling at the start of stages 2–4), a global average pool to yield a 512-dimensional vector, and a final linear projection ($512 \rightarrow d_1$) with BatchNorm1d. **Touch Encoder** : raw force trajectories $\mathbf{F} \in \mathbb{R}^{T \times 6}$ are processed by a sequence of 1D convolutions over time (kernel size 5), each with BatchNorm1d and ReLU, then collapsed via global max- or average-pooling, and finally mapped to $d_2$ dimensions by a linear layer (Lee et al., 2020; Dufumier et al., 2025).

For *Rest-meta-MDD*, fMRI is modeled as $G = (V, E)$ with Fisher-$z$ edges. An optional $E$-dim node-ID embedding (per `emb_style`) is concatenated to features (Bullmore & Sporns, 2009). We apply $L$ GIN layers

$$x_v^{(l+1)} = \text{MLP}\big(x_v^{(l)} + \sum_{u \in \mathcal{N}(v)} x_u^{(l)}\big), \tag{44}$$

(hid. size $H$, out $O$, with dropout/BatchNorm), followed by two residual GATConv layers (4 heads, head-dim $O/4$), a node-wise MLP + sigmoid attention, and two SAGPooling stages (ratios 0.8, 0.6) each with global mean-pooling. Pooled vectors are summed and passed through Linear($O \rightarrow O$) – BN – ReLU – Dropout to yield the raw $O$-dim embedding. Normalized sMRI volumes feed a six-stage 3D CNN: the first $L - 1$ blocks use either standard conv ($3^3$, stride-2 pool, BN, ReLU) or depthwise-separable conv; the final block is a $1^3$ conv. After each conv (except last) we optionally apply ChannelAttention (avg/max→MLP→sigmoid) and SpatialAttention (avg/max→conv→sigmoid) (Woo et al., 2018). A final 3D avg-pool over $[5 \times 6 \times 5]$ produces a $C$-dim vector, then Linear($C \rightarrow C$)–BN–ReLU–Dropout yields the raw $C$-dim embedding.

## E.2 PREDICTOR ARCHITECTURE

The regression head is a three-stage MLP that nonlinearly maps the fused latent embedding of dimension $d$ to a scalar. The first stage applies Linear($d \rightarrow H$), followed by `BatchNorm1d`, in-place `ReLU`, and `Dropout`($p = 0.3$) to capture high-capacity feature interactions. The second stage performs Linear($H \rightarrow H/2$), plus `BatchNorm1d`, `ReLU`, and `Dropout`($p = 0.2$), thereby compressing the representation. The final stage is Linear($H/2 \rightarrow prediction$), which produces the scalar output. This *wide* → *narrow* → *scalar* design balances expressive power and regularization for downstream regression.

## E.3 TRAINING STRATEGIES AND HYPERPARAMETER

### E.3.1 EVALUATION PROTOCOL

The reported results are averaged over three independent runs. To further assess robustness on the largest REST-meta-MDD dataset, we additionally adopt the leave-two-site-out evaluation protocol, which is commonly used in the medical domain. Detailed results for REST-meta-MDD are reported in Appendix G.2.

### E.3.2 DATA PREPROCESSING

Prior to training, all input features (both modalities) and targets are independently standardized to zero mean and unit variance using scikit-learn's `StandardScaler`. For the synthetic data experiment, we generate a dataset of 10 000 samples following the prescribed formulation. The *Synthetic*, *CT Slice*, and *Superconductivity* datasets are each split into training, validation, and test sets using a fixed random seed. For *CMU-MOSI*, *CMU-MOSEI*, and *Vision&Touch*, we employ the official train/val/test partitions, while *Rest-meta-MDD* is divided by site.

### E.3.3 MODEL INITIALIZATION & LATENT DIMENSIONS

We initialize all `Linear` and `Conv` layer weights using Kaiming initialization and set biases to zero (He et al., 2015). For all experiments, the latent dimensions of $Z_1$ and $Z_2$ are set to 64 to provide sufficient representational capacity while controlling the computational overhead of PID decomposition.

### E.3.4 OPTIMIZER & LEARNING RATE SCHEDULING

We employ two Adam optimizers: one for the predictor and projection networks, and one for the information-bottleneck coefficient $\lambda^b$. A `ReduceLROnPlateau` scheduler is attached to the predictor optimizer, monitoring validation prediction loss with factor = 0.5, patience = 10 epochs, and min_lr = $10^{-6}$. The initial learning rates are $1 \times 10^{-3}$ for the predictor/projection optimizer and 0.1 for the $\lambda$ optimizer.

### E.3.5 LOSS WEIGHTS

We empirically observe that setting $\lambda_1 = \lambda_2 = \lambda_3 = 0.1$ delivers consistently robust performance across all datasets; and each regularizer's contribution is confirmed by our ablation studies. We do not consider any additional performance gains from dataset-specific $\lambda$ hyperparameter tuning to ensure concise and consistent.

### E.3.6 Regularization & Gradient Management

Dropout (rates between 0.2 and 0.5) is interleaved with `BatchNorm` in all MLP layers. We apply $L_2$-norm gradient clipping with a maximum norm of 1.0 to the predictor subnetwork.

### E.3.7 Early Stopping & Checkpointing

Training is halted when the validation total loss fails to decrease for 30 consecutive epochs. Whenever the validation loss reaches a new minimum, we save the model weights, optimizer state, the $\lambda^b$ parameter, and PID fusion weights. After training completes, the best checkpoint is reloaded for final test evaluation.

### E.3.8 Batch Size & Memory Requirements

We use a batch size of 32 for **CMU-MOSI**, 20 for **Rest-meta-MDD**, and 256 for all other datasets. Experiments on **Rest-meta-MDD** require at least 24 GB of GPU memory (e.g., NVIDIA RTX 4090 or equivalent).

## F Extension to Multivariate Gaussian and More than Two Modalities

Extending the PID framework to more than two input sources is a widely recognized open problem in information theory. Even with three sources, the complete Williams–Beer (Williams & Beer, 2010) lattice already comprises 18 distinct partial-information atoms that enumerate all unique, redundant, and synergistic interactions. the number of such atoms increases super-exponentially with the number of inputs. In what follows, we introduce a practical formulation for three modalities and outline two complementary optimization strategies, further highlighting the potential of PIDReg.

### F.1 Pragmatic Simplification for Three Modalities

Given this combinatorial complexity, we can adopt a pragmatic simplification. Specifically, we aggregate all redundancy-related atoms into a single overall redundancy term, and similarly, group all synergy-related atoms into a single overall synergy term (please refer to Fig.1(b) in Griffith & Koch (2014)). This enables us to decompose the total mutual information $I(Z_1, Z_2, Z_3; Y)$ as:

$$I(Z_1, Z_2, Z_3; Y) = R + U_{Z_1} + U_{Z_2} + U_{Z_3} + S, \tag{45}$$

where $R$ represents the common information redundantly available in all three sources, $U_{Z_i}$ represents the information uniquely available in source $Z_i$, $S$ captures all synergistic interactions that are only present when multiple sources are considered together.

In the meantime, we also have:

$$\begin{cases} I(Z_1; Y) = R + U_{Z_1} \\ I(Z_2; Y) = R + U_{Z_2} \\ I(Z_3; Y) = R + U_{Z_3} \end{cases} \tag{46}$$

However, Eqs. (45) and (46) involve five variables but only four equations, making the system underdetermined. To resolve this ambiguity, we formally specify the Union Information (the sum of all unique information and redundancy) as:

$$I^U := \min_{Q \in \Delta_P} I_Q(Z_1, Z_2, Z_3; Y). \tag{47}$$

Importantly, this aggregation not only alleviates the combinatorial complexity of the full PID lattice but also admits a natural optimization-based formulation, as elaborated in Appendix F.2.

### F.2 Tractable Optimization Objective

The above definitions generalize the two-source definition from Venkatesh et al. (2023); Bertschinger et al. (2014). In the special case where $Z_1, Z_2, Z_3, Y$ are jointly Gaussian, the optimization defining union information and unique information becomes particularly tractable. We

take $I^U$ as an example, since mutual information and conditional mutual information admit closed-form expressions over Gaussian variables, the optimization of $I^U$ reduces to the following convex optimization over covariance matrices:

$$\min_{\Sigma_Q \succeq 0} I_{\Sigma_Q}(Z_1, Z_2, Z_3; Y) \quad \text{s.t.} \quad \Sigma_Q^{Z_i, Y} = \Sigma_P^{Z_i, Y}, \; i = 1, 2, 3, \tag{48}$$

where $\Sigma_Q$ is the candidate joint covariance matrix, and $\Sigma_Q^{Z_i, Y}$ denotes its $(X_i, Y)$ marginal block.

By definition:

$$I(Y; Z_1, Z_2, Z_3) = \frac{1}{2} \log \left( \frac{\det(\Sigma_{Z_1 Z_2 Z_3})}{\det(\Sigma_{Z_1 Z_2 Z_3 | Y})} \right). \tag{49}$$

According to Venkatesh et al. (2023), the optimization problem can be reformulated as follows:

$$I^U := \min_{\Sigma_{Z_1 Z_2 Z_3 | Y}^Q} \frac{1}{2} \log \det \left( I + \sigma_Y^{-2} \begin{bmatrix} \Sigma_{Y Z_1}^P \\ \Sigma_{Y Z_2}^P \\ \Sigma_{Y Z_3}^P \end{bmatrix} \left( \Sigma_{Z_1 Z_2 Z_3 | Y}^Q \right)^{-1} \begin{bmatrix} \Sigma_{Y Z_1}^P \\ \Sigma_{Y Z_2}^P \\ \Sigma_{Y Z_3}^P \end{bmatrix}^T \right) \quad \text{s.t.} \quad \Sigma_{Z_1 Z_2 Z_3 | Y}^Q \succeq 0, \tag{50}$$

which has the same optimization form as Eq.( 5) and is likewise amenable to projected gradient descent.

### F.3 GRADIENTS AND OPTIMIZATION

For compactness, denote:

$$M := I + \sigma_Y^{-2} \begin{bmatrix} \Sigma_{Y Z_1}^P \\ \Sigma_{Y Z_2}^P \\ \Sigma_{Y Z_3}^P \end{bmatrix} \left( \Sigma_{Z_1 Z_2 Z_3 | Y}^Q \right)^{-1} \begin{bmatrix} \Sigma_{Y Z_1}^P \\ \Sigma_{Y Z_2}^P \\ \Sigma_{Y Z_3}^P \end{bmatrix}^\top, \qquad f \left( \Sigma_{Z_1 Z_2 Z_3 | Y}^Q \right) := \frac{1}{2} \log \det M. \tag{51}$$

The differential of $f$ with respect to $\Sigma_{Z_1 Z_2 Z_3 | Y}^Q$ is:

$$\mathrm{d}f = \frac{1}{2} \operatorname{tr}(M^{-1} \mathrm{d}M)$$

$$= -\frac{1}{2} \sigma_Y^{-2} \operatorname{tr} \left( \left( \Sigma_{Z_1 Z_2 Z_3 | Y}^Q \right)^{-1} \begin{bmatrix} \Sigma_{Y Z_1}^P \\ \Sigma_{Y Z_2}^P \\ \Sigma_{Y Z_3}^P \end{bmatrix}^\top M^{-1} \begin{bmatrix} \Sigma_{Y Z_1}^P \\ \Sigma_{Y Z_2}^P \\ \Sigma_{Y Z_3}^P \end{bmatrix} \left( \Sigma_{Z_1 Z_2 Z_3 | Y}^Q \right)^{-1} \mathrm{d}\Sigma_{Z_1 Z_2 Z_3 | Y}^Q \right). \tag{52}$$

Consequently,

$$\nabla_{\Sigma_{Z_1 Z_2 Z_3 | Y}^Q} f = -\frac{1}{2} \sigma_Y^{-2} \left( \Sigma_{Z_1 Z_2 Z_3 | Y}^Q \right)^{-1} \begin{bmatrix} \Sigma_{Y Z_1}^P \\ \Sigma_{Y Z_2}^P \\ \Sigma_{Y Z_3}^P \end{bmatrix}^\top M^{-1} \begin{bmatrix} \Sigma_{Y Z_1}^P \\ \Sigma_{Y Z_2}^P \\ \Sigma_{Y Z_3}^P \end{bmatrix} \left( \Sigma_{Z_1 Z_2 Z_3 | Y}^Q \right)^{-1}. \tag{53}$$

Starting from any feasible initialization $\Sigma_{Z_1 Z_2 Z_3 | Y}^{Q(0)} \succeq 0$, we apply a projected gradient descent scheme:

$$\widetilde{\Sigma}^{Q(t+1)} = \Sigma_{Z_1 Z_2 Z_3 | Y}^{Q(t)} - \eta_t \nabla_{\Sigma_{Z_1 Z_2 Z_3 | Y}^Q} f \left( \Sigma_{Z_1 Z_2 Z_3 | Y}^{Q(t)} \right), \qquad \Sigma_{Z_1 Z_2 Z_3 | Y}^{Q(t+1)} = \Pi_{\mathbb{S}_+} \left( \widetilde{\Sigma}^{Q(t+1)} \right), \tag{54}$$

where $\eta_t > 0$ is the step size. The operator $\Pi_{\mathbb{S}_+}$ denotes the projection onto the cone of positive semi-definite (PSD) matrices, ensuring that the constraint $\Sigma^Q \succeq 0$ is preserved at every iteration. Specifically, if $X = U \operatorname{diag}(\lambda_1, \ldots, \lambda_d) U^\top$ is the eigendecomposition of $X$, then:

$$\Pi_{\mathbb{S}_+}(X) \;=\; U \operatorname{diag}\big(\max\{\lambda_1, 0\}, \ldots, \max\{\lambda_d, 0\}\big) U^\top \;+\; \varepsilon I, \tag{55}$$

with a small $\varepsilon > 0$ added as a safeguard for numerical stability. To select a suitable step size $\eta_t$, we adopt a backtracking line search based on the Armijo rule, which guarantees a sufficient decrease of the objective (Nocedal & Wright, 2006), i.e., $f(\Sigma^{Q(t+1)}) \leq f(\Sigma^{Q(t)})$. The iteration terminates once the progress becomes negligible, $|f(\Sigma^{Q(t+1)}) - f(\Sigma^{Q(t)})| < \delta$, or when a budget of steps is reached. For robustness, all matrix inverses in Eq. (53) are evaluated via linear solves, and $\log \det(\cdot)$ is computed using slogdet to avoid numerical overflow or underflow.

**Synergy.** With the optimizer $\Sigma^{Q\star}_{Z_1 Z_2 Z_3 | Y}$ obtained by the projected gradient scheme in Eqs.( 53) to( 55), we derive the corresponding $I^{U\star}$. Together with the total mutual information given by Eq. (49), the synergy component can then be expressed as:

$$S \;=\; I(Z_1, Z_2, Z_3; Y) \;-\; I^{U\star}. \tag{56}$$

**Redundancy.** For each $i \in \{1, 2, 3\}$, the pairwise mutual information is computed directly from the empirical covariance $\Sigma^P$:

$$I(Z_i; Y) \;=\; \frac{1}{2} \log \det\Big(I + \sigma_Y^{-2} \Sigma^P_{Y Z_i} (\Sigma^P_{Z_i Z_i})^{-1} \Sigma^P_{Z_i Y}\Big). \tag{57}$$

Under the pragmatic three modality decomposition and the definition of union information for three modalities, we have:

$$I^U \;=\; R + U_{Z_1} + U_{Z_2} + U_{Z_3}, \qquad I(Z_i; Y) \;=\; U_{Z_i} + R \;\; (i = 1, 2, 3), \tag{58}$$

which leads to the following expression for redundancy:

$$R \;=\; \frac{I(Z_1; Y) + I(Z_2; Y) + I(Z_3; Y) \;-\; I^{U\star}}{2}. \tag{59}$$

By substituting Eq.( 57) into Eq.( 59), the redundancy can be numerically evaluated.

**Uniqueness.** Each unique information term admits a closed-form expression:

$$U_{Z_i} \;=\; I(Z_i; Y) - R \;=\; I(Z_i; Y) - \frac{I(Z_1; Y) + I(Z_2; Y) + I(Z_3; Y) \;-\; I^{U\star}}{2}, \qquad i \in \{1, 2, 3\}. \tag{60}$$

### F.4 NETWORK IMPLEMENTATION

In terms of network implementation, we can introduce a new modality-specific encoder for $X_3$ to generate $Z_3$. The final fused representation can then be formulated as:

$$Z = w_1 Z_1 + w_2 Z_2 + w_3 Z_3 + w_{123}(Z_1 \circ Z_2 \circ Z_3), \tag{61}$$

similar to Eq.( 2), the last term explicitly captures the synergistic effect. The weights $w_1, w_2, w_3$ and $w_{123}$ are adjusted by the computed PID terms accordingly.

### F.5 EXPERIMENTAL RESULTS OF TRI-MODAL PIDREG

We further evaluate on the ***CMU-MOSI*** and ***CMU-MOSEI***, both of which are inherently tri-modal (text, audio, and video), comparing the tri-modal extension of PIDReg against its bi-modal counterpart. The predictive results are reported in Table 11, while the corresponding information decomposition is provided in Table 12.

| Method | CMU-MOSI | | | | | CMU-MOSEI | | | | |
|---|---|---|---|---|---|---|---|---|---|---|
| | $A_7 \uparrow$ | $A_2 \uparrow$ | F1 $\uparrow$ | MAE $\downarrow$ | Corr $\uparrow$ | $A_7 \uparrow$ | $A_2 \uparrow$ | F1 $\uparrow$ | MAE $\downarrow$ | Corr $\uparrow$ |
| PIDReg$^{\heartsuit}$ | 32.0 | 80.0 | 79.7 | 0.938 | 0.662 | 47.4 | 80.2 | 80.0 | 0.634 | 0.662 |
| PIDReg$^{\spadesuit}$ | 37.2 | 80.8 | 80.9 | 0.947 | 0.664 | 47.0 | 80.6 | 80.2 | 0.642 | 0.661 |
| PIDReg$^{\diamondsuit}$ | 16.4 | 52.3 | 51.8 | 1.400 | 0.149 | 41.7 | 63.4 | 63.7 | 0.828 | 0.228 |
| PIDReg$^{\clubsuit}$ | **38.2** | **81.6** | **81.6** | **0.899** | **0.699** | **48.7** | **81.7** | **81.5** | **0.620** | **0.679** |

Table 11: Tri-Modal PIDReg results ($\clubsuit$ denotes modality combination of Text–Vision–Audio).

| Component | CMU-MOSI | | | | | CMU-MOSEI | | | | |
|---|---|---|---|---|---|---|---|---|---|---|
| | $U_1^{\triangle}$ | $U_2^{\square}$ | $U_3^{\circ}$ | R | S | $U_1^{\triangle}$ | $U_2^{\square}$ | $U_3^{\circ}$ | R | S |
| Value | 0.110 | 0.052 | 0.077 | 0.817 | 6.108 | 0.083 | 0.070 | 0.081 | 0.496 | 6.372 |

Table 12: Tri-modal Gaussian PID decomposition. $^{\triangle}$Text modality, $^{\square}$Vision modality, $^{\circ}$Audio modality.

It is noteworthy that the high synergy observed in Table 12 aligns with the inherent logic of multi-modal sentiment prediction in ***CMU-MOSI*** and ***CMU-MOSEI***. This further explains why the performance improvement from Table 2 to Table 11 primarily stems from the synergistic information across different modalities in predicting sentiment.

## G  EXTENDED RESULTS

### G.1  COMPUTATIONAL EFFICIENCY

PIDReg introduces no additional trainable parameters compared to standard fusion models and therefore does not increase model complexity. The encoder and predictor networks follow conventional architectures (e.g., MLPs, ResNet), as detailed in Appendices E.1 and Appendix E.2. PIDReg performs one PID optimization per iteration in the first phase, slightly increasing training time, but remains comparable to other methods. Table 13 reports the average training time per epoch across representative baselines on same device (all require 200 epochs of training).

| Method | DER | MoNIG | PIDReg |
|---|---|---|---|
| Time | 1.756 | 3.280 | 4.641 |

Table 13: Average training time per epoch (in seconds).

### G.2  FINE-GRAINED RESULTS ON REST-META-MDD

In Table 3 of the main paper, we present brain age regression results averaged across all sites. Detailed, site-specific performance metrics are reported in Table 14 (We have preserved the site-labeling conventions from Yan et al. (2019) and related works, no adjustments have been made). In brain-age prediction, models commonly exhibit a *regression-to-the-mean* effect—overestimating young subjects' ages and underestimating older subjects'. To eliminate this systematic bias and thereby ensure that the corrected age estimates yield more reliable and interpretable outcomes, we first fit a simple linear model on the training set (Beheshti et al., 2019; Cole et al., 2017):

$$\hat{y}_i = \alpha + \beta\, y_i + \varepsilon_i, \tag{62}$$

where $y_i$ is the chronological age and $\hat{y}_i$ the raw model prediction. The estimated intercept $\alpha$ and slope $\beta$ quantify the overall offset and compression of the prediction relative to true age.

Any new prediction $\hat{y}^{\text{raw}}$ is then bias-corrected via:

$$\hat{y}^{\text{corr}} = \frac{\hat{y}^{\text{raw}} - \alpha}{\beta}, \tag{63}$$

so that, by construction, $\hat{y}^{\mathrm{corr}} \approx y$ lies on the identity line within the training distribution (Smith et al., 2019). Equivalently, one may report the residual brain-age gap:

$$\Delta_i = \hat{y}_i^{\mathrm{raw}} - \left(\alpha + \beta\, y_i\right), \tag{64}$$

which is inherently uncorrelated with chronological age.

To avoid information leakage, this correction is performed independently. The resulting age-adjusted predictions (or brain-age gaps) are thus independent of true age, enhancing both interpretability and the validity of downstream associations with biological or clinical variables.

Fig. 7 further provides an intuitive visual summary of PIDReg's raw versus linear bias-corrected age predictions across all sites.

| Test Site(s) | Age | Gender(M/F) | Sample(MDD/HC) | MV | | MIB | | MoNIG | | NIG | | CoMM | | PIDReg | |
|---|---|---|---|---|---|---|---|---|---|---|---|---|---|---|---|
| | | | | MAE | Corr | MAE | Corr | MAE | Corr | MAE | Corr | MAE | Corr | MAE | Corr |
| S20 | $39.0 \pm 13.9$ | 157/322 | 250/229 | 7.900 | 0.794 | 9.048 | 0.608 | 7.960 | 0.792 | **7.498** | 0.628 | 7.749 | 0.592 | 7.518 | **0.797** |
| S7,S9 | $33.5 \pm 11.4$ | 83/85 | 83/85 | 9.497 | 0.479 | **6.374** | 0.679 | 9.591 | 0.379 | 10.619 | 0.107 | 10.187 | $-0.131$ | 7.776 | **0.723** |
| S14,S19 | $31.7 \pm 8.4$ | 53/89 | 79/63 | 6.793 | **0.824** | 5.103 | 0.633 | 8.232 | 0.611 | 8.912 | 0.579 | 10.438 | $-0.037$ | **4.284** | 0.751 |
| S1,S8 | $31.7 \pm 9.2$ | 106/156 | 127/135 | 7.946 | 0.592 | 6.696 | 0.513 | 9.223 | 0.426 | 9.855 | 0.364 | 9.574 | 0.253 | **5.967** | **0.687** |
| S17,S10 | $26.5 \pm 9.0$ | 65/88 | 86/67 | 7.292 | 0.637 | 5.757 | 0.739 | 6.610 | **0.785** | 6.872 | 0.773 | 7.316 | 0.675 | **4.687** | 0.776 |
| S23,S15 | $37.5 \pm 14.3$ | 44/68 | 52/60 | 7.759 | 0.764 | 7.727 | 0.708 | 9.454 | 0.761 | 17.693 | 0.771 | 11.375 | 0.357 | **6.678** | **0.849** |
| S22,S13 | $31.1 \pm 9.8$ | 34/40 | 38/36 | 10.123 | 0.334 | 8.476 | 0.448 | 11.250 | 0.525 | 9.901 | 0.456 | 8.476 | 0.373 | **6.553** | 0.562 |
| S21,S11 | $34.3 \pm 11.9$ | 79/102 | 99/82 | 7.959 | 0.622 | **7.039** | 0.657 | 9.894 | 0.446 | 11.932 | 0.313 | 9.825 | 0.373 | 7.977 | **0.754** |
| S2,S4 | $35.7 \pm 11.1$ | 24/47 | 34/37 | 5.196 | **0.847** | **4.556** | 0.790 | 6.130 | 0.580 | 6.931 | 0.825 | 10.229 | $-0.034$ | 5.172 | 0.843 |

Table 14: Cross-site results on Rest-meta-MDD.

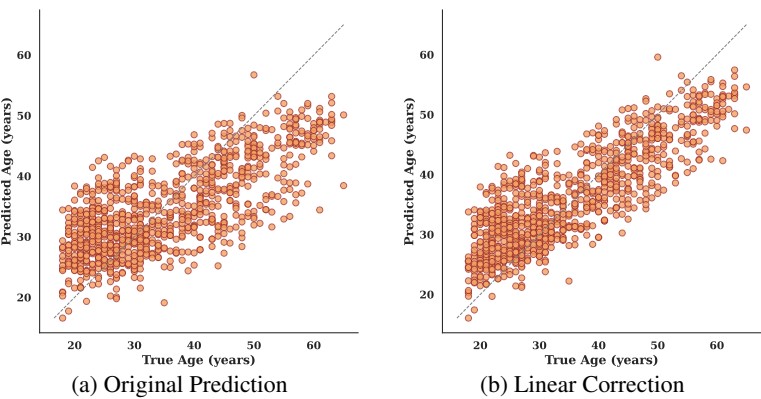

(a) Original Prediction        (b) Linear Correction

Figure 7: Brain age prediction results.

### G.3 LARGE SCALE BIMODAL MNIST REGRESSION BENCHMARK

Given the scarcity of large-scale, high-quality benchmarks for bimodal regression with known information contributions, we additionally design a new benchmark derived from the MNIST dataset. Specifically, starting from the 70,000 grayscale images of size $28 \times 28$, we construct a dataset of 140,000 samples as follows:

For each original image, sample a rotation angle $\theta \sim \mathcal{U}(-90°, 90°)$, which serves as the regression target. Rotate the image by $\theta$ degrees, add Gaussian noise whose standard deviation is drawn from $\mathcal{U}(0, 0.05)$, and apply a random contrast scaling sampled from $\mathcal{U}(0.8, 1.2)$. For each augmented image, we extract two heterogeneous feature modalities:

**Raw - pixel modality:** Flatten the $28 \times 28$ image into a 784-dimensional vector $X_1$, preserving all low-level visual information.

**Structured - feature modality:** Concatenate four classes of handcrafted descriptors, statistical moments, edge and gradient features, shape-contour descriptors, and frequency-domain features, into a 278-dimensional vector $X_2$, aimed at capturing higher-order structural information.

As shown in Fig 8, both the raw pixel modality (left) and the feature modality (right), when projected to two dimensions via PCA, exhibit clear deviations from Gaussianity.

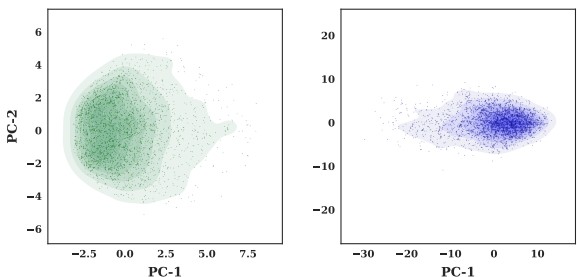

Figure 8: Non-Gaussianity of the raw data

Each sample is thus associated with a scalar target $\theta$ and two modalities $X_1 \in \mathbb{R}^{784}$ and $X_2 \in \mathbb{R}^{278}$. We then compare PIDReg against baseline methods in terms of predictive performance and information decomposition. The results are reported in Table 15. Fig. 9 further presents the visualizations of the latent representations $Z_1$, $Z_2$ learned by our encoder architecture.

| Metric | MIB | MoNIG | MEIB | DER | PIDReg |
|---|---|---|---|---|---|
| MAE↓ | 7.70 | 9.28 | 10.17 | 10.01 | **5.90** |
| $R^2$ ↑ | 0.95 | 0.94 | 0.86 | 0.91 | **0.97** |

| Component | $U_1^*$ | $U_2^\circ$ | $R$ | $S$ |
|---|---|---|---|---|
| Value | 0.866 | 0.000 | 0.250 | 0.076 |

Table 15: MNIST regression results (top) and Gaussian PID decomposition values (bottom). Pixel modality (*) and feature modality (°).

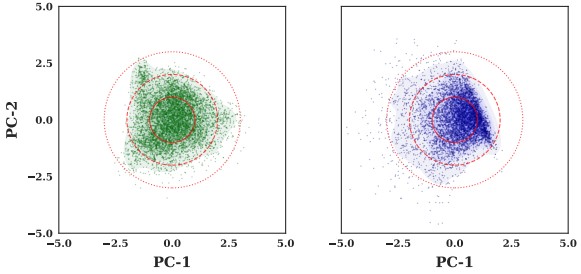

Figure 9: Gaussianity of latent representations

The raw-pixel modality retains the complete visual content of each image. Because the structured-feature modality is derived solely from descriptors that encode the same rotation-angle information present in the raw pixels, it does not contribute substantial additional information. The PID decomposition therefore unambiguously reveals the intrinsic relationship between these two modalities (redundant and no unique information from $X_2$). The results on MNIST further underscore the effectiveness and robustness of PIDReg on real world data.

### G.4 EVALUATION OF PIDREG UNDER EXTREME SCENARIOS AND NON-GAUSSIAN LATENT VARIABLES

To further validate the superiority of PIDReg in terms of predictive accuracy and information decomposition, we conduct the following additional experiments on synthetic data: (i) extreme boundary cases of information components (section G.4.1), (ii) severely skewed non-Gaussian information components (section G.4.2). (iii) discrete latent variables with non-trivial synergistic interaction patterns (section G.4.3). We further provide a physical interpretation of the relationship between the synthetic weights and the estimated PID components in section G.4.4.

### G.4.1 EXTREME SCENARIOS

First, we follow the same setup as in section 4.1, where synthetic data are generated according to Eq. (22) and Eq. (23). However, our focus here is on scenarios where a single PID component clearly dominates in the fusion process. Our goal is to verify whether the PID values computed by our method faithfully reflect the truly dominant component. We repeat the experiments under this setting, and the results shown in Fig. 10, demonstrate that PIDReg can accurately capture the information components even in such extreme cases.

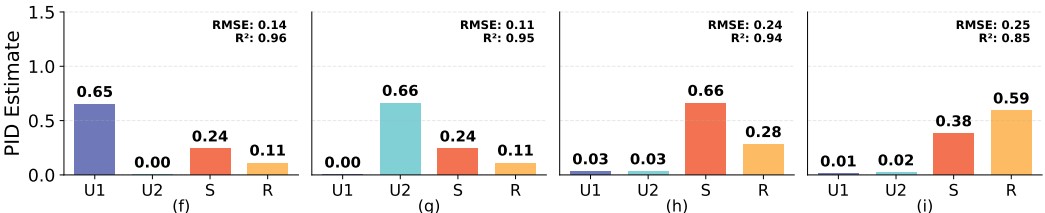

Figure 10: Estimated PID values when (f) $w_{u1} = 1.00$, $w_{u2} = 0.00$, $w_s = 0.00$, $w_r = 0.00$; (g) $w_{u1} = 0.00$, $w_{u2} = 1.00$, $w_s = 0.00$, $w_r = 0.00$; (h) $w_{u1} = 0.00$, $w_{u2} = 0.00$, $w_s = 1.00$, $w_r = 0.00$; (i) $w_{u1} = 0.00$, $w_{u2} = 0.00$, $w_s = 0.00$, $w_r = 1.00$.

### G.4.2 NON-GAUSSIAN LATENT VARIABLES

Furthermore, instead of drawing $R, U_1, U_2 \sim \mathcal{N}(0, 1)$, we sample them from a chi-squared distribution, $R, U_1, U_2 \sim \chi_4^2$, such that the variables significantly deviate from Gaussianity, the results are shown in Fig. 11. This allows us to provide an additional validation of PIDReg from the perspective of known ground truth, thereby assessing its reliability under skewed data distributions that often arise in real-world scenarios.

From these results, we conclude the following: 1) when a particular component is deactivated, PIDReg correctly identifies its absence, e.g.,

$$\omega_{u1} = 0 \quad \Longrightarrow \quad U_1 \approx 0; \tag{65}$$

and 2) when the relative strengths of two components are varied, the estimated PID values faithfully track the corresponding monotonic trends, e.g.,

$$w_s^{(2)} > w_s^{(1)}, \quad w_r^{(2)} < w_r^{(1)} \quad \Longrightarrow \quad S_{\text{est}}^{(2)} > S_{\text{est}}^{(1)}, \; R_{\text{est}}^{(2)} < R_{\text{est}}^{(1)}. \tag{66}$$

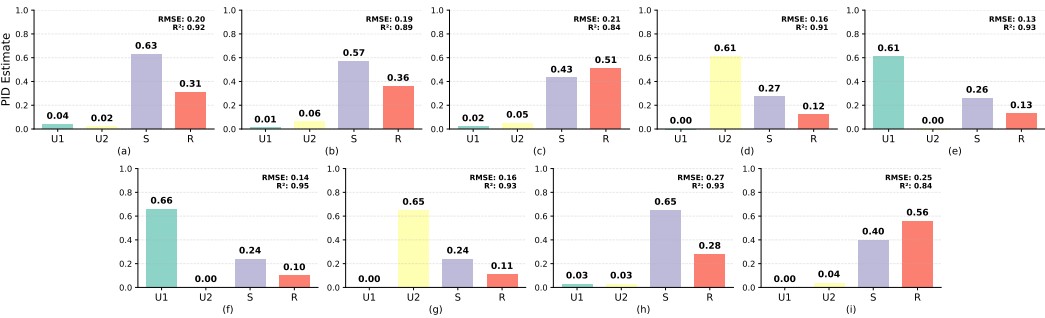

Figure 11: Estimated PID values when (a) $w_{u1} = 0.00$, $w_{u2} = 0.00$, $w_s = 0.75$, $w_r = 0.25$; (b) $w_{u1} = 0.00$, $w_{u2} = 0.00$, $w_s = 0.50$, $w_r = 0.50$; (c) $w_{u1} = 0.00$, $w_{u2} = 0.00$, $w_s = 0.25$, $w_r = 0.75$; (d) $w_{u1} = 0.00$, $w_{u2} = 0.80$, $w_s = 0.10$, $w_r = 0.10$; (e) $w_{u1} = 0.80$, $w_{u2} = 0.00$, $w_s = 0.10$, $w_r = 0.10$; (f) $w_{u1} = 1.00$, $w_{u2} = 0.00$, $w_s = 0.00$, $w_r = 0.00$; (g) $w_{u1} = 0.00$, $w_{u2} = 1.00$, $w_s = 0.00$, $w_r = 0.00$; (h) $w_{u1} = 0.00$, $w_{u2} = 0.00$, $w_s = 1.00$, $w_r = 0.00$; (i) $w_{u1} = 0.00$, $w_{u2} = 0.00$, $w_s = 0.00$, $w_r = 1.00$.

To further verify the numerical effect of Gaussian regularization on PID decomposition, we consider specifically when the weights are $w_{u1} = 0.00, w_{u2} = 0.00, w_s = 0.75, w_r = 0.25$. Under this setup, we repeat the experiment and progressively disable the $\mathcal{L}_{CS}$ and $\mathcal{L}_{\text{Gauss}}$ terms, either individually or jointly.

The resulting PID decompositions are summarized in Table 16. As shown in the results, imposing explicit Gaussian regularization in the latent space enables a more principled and rigorous implementation of PIDReg. Nevertheless, even when the Gaussian regularization is removed (i.e., set to zero), our Gaussian-PID estimator relies only on the first- and second-order statistics of the latent variables. This relaxation does not lead to any significant deviation from the results obtained with Gaussian regularization. This is expected, as in many practical settings the first- and second-order moments provide a sufficiently accurate characterization of the underlying distributions.

When both regularizers are removed, the estimated value of $S$ shows a modest decrease, but the drop remains moderate. This behavior is expected: ignoring higher–order information naturally leads to a slight underestimation of synergy, since synergy is particularly sensitive to joint dependencies that cannot be captured by first- and second-order statistics alone.

| | $U_1$ | $U_2$ | $R$ | $S$ |
|---|---|---|---|---|
| Full model | 0.04 | 0.02 | 0.31 | 0.63 |
| No $\mathcal{L}_{CS}$ | 0.06 | 0.03 | 0.26 | 0.56 |
| No $\mathcal{L}_{\text{Gauss}}$ | 0.05 | 0.03 | 0.25 | 0.54 |
| No $\mathcal{L}_{CS}, \mathcal{L}_{\text{Gauss}}$ | 0.03 | 0.03 | 0.26 | 0.49 |

Table 16: PID Stability under Regularization Ablation.

### G.4.3 DISCRETE LATENT VARIABLES AND NON-TRIVIAL SYNERGY INTERACTION PATTERNS

To assess whether PIDReg may underestimate the importance of synergy when the true latent variables include discrete components and exhibit non-trivial synergistic interactions, we further set $R, U_2 \sim \mathcal{N}(0,1)$   $U_1 \sim \text{Rademacher}(0.5)$, i.e., $P(U_1 = k) = 0.5$ for $k \in \{-1, 1\}$, and kept the rest of the synthetic pipeline unchanged, with synergy implemented as a sign-flip interaction: the synergistic term depends on $\text{sign}(U_1)\,U_2$. All other settings remain unchanged, and the experiment is repeated. The results are shown in Figure 12.

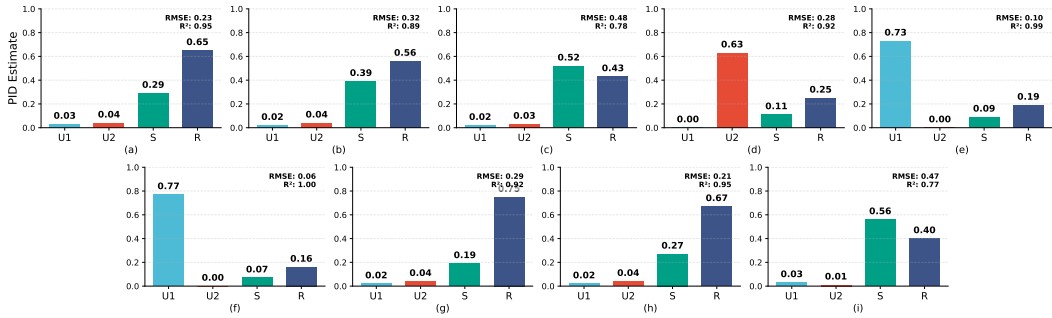

Figure 12: Estimated PID values when (a) $w_{u1} = 0.00, w_{u2} = 0.00, w_s = 0.75, w_r = 0.25$; (b) $w_{u1} = 0.00, w_{u2} = 0.00, w_s = 0.50, w_r = 0.50$; (c) $w_{u1} = 0.00, w_{u2} = 0.00, w_s = 0.25, w_r = 0.75$; (d) $w_{u1} = 0.00, w_{u2} = 0.80, w_s = 0.10, w_r = 0.10$; (e) $w_{u1} = 0.80, w_{u2} = 0.00, w_s = 0.10, w_r = 0.10$; (f) $w_{u1} = 1.00, w_{u2} = 0.00, w_s = 0.00, w_r = 0.00$; (g) $w_{u1} = 0.00, w_{u2} = 1.00, w_s = 0.00, w_r = 0.00$; (h) $w_{u1} = 0.00, w_{u2} = 0.00, w_s = 1.00, w_r = 0.00$; (i) $w_{u1} = 0.00, w_{u2} = 0.00, w_s = 0.00, w_r = 1.00$.

Next, let $U_1$ follow a Gaussian mixture distribution:

$$R, U_2 \sim \mathcal{N}(0,1), \qquad U_1 \sim \tfrac{1}{2}\mathcal{N}(\mu, \sigma^2) + \tfrac{1}{2}\mathcal{N}(-\mu, \sigma^2), \tag{67}$$

where, $\mu = 2, \sigma^2 = 0.2$. Additional experiments are performed with this setting. The results are shown in Figure 13.

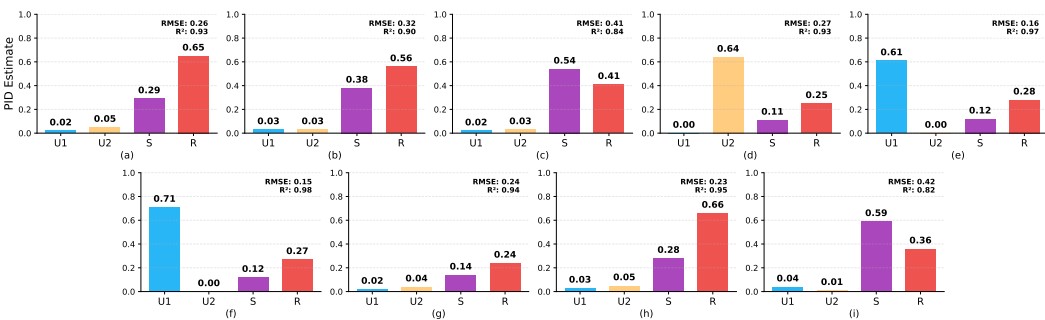

Figure 13: Estimated PID values when (a) $w_{u1} = 0.00$, $w_{u2} = 0.00$, $w_s = 0.75$, $w_r = 0.25$; (b) $w_{u1} = 0.00$, $w_{u2} = 0.00$, $w_s = 0.50$, $w_r = 0.50$; (c) $w_{u1} = 0.00$, $w_{u2} = 0.00$, $w_s = 0.25$, $w_r = 0.75$; (d) $w_{u1} = 0.00$, $w_{u2} = 0.80$, $w_s = 0.10$, $w_r = 0.10$; (e) $w_{u1} = 0.80$, $w_{u2} = 0.00$, $w_s = 0.10$, $w_r = 0.10$; (f) $w_{u1} = 1.00$, $w_{u2} = 0.00$, $w_s = 0.00$, $w_r = 0.00$; (g) $w_{u1} = 0.00$, $w_{u2} = 1.00$, $w_s = 0.00$, $w_r = 0.00$; (h) $w_{u1} = 0.00$, $w_{u2} = 0.00$, $w_s = 1.00$, $w_r = 0.00$; (i) $w_{u1} = 0.00$, $w_{u2} = 0.00$, $w_s = 0.00$, $w_r = 1.00$.

Across both figures, PIDReg consistently recovers the correct qualitative structure of the underlying information components. In particular, even when synergy is generated by highly non-Gaussian or discontinuous interaction mechanisms, PIDReg accurately captures its relative contribution and preserves a stable, interpretable decomposition.

### G.4.4 Interpretation of estimated PID components

The true weights $\omega_{u1}, \omega_{u2}, \omega_R, \omega_S$ and the estimated PID components $U_1, U_2, R, S$ differ in both their physical units and the mathematical spaces they inhabit, so numerical discrepancies are expected. In Eq.( 23), $\omega_{u1}, \omega_{u2}, \omega_R, \omega_S$ are linear scaling factors applied to latent variables in the generative process. For example, modality 1 follows:

$$X_1 = f_{\text{gen}}([R, U_1]), \quad Z_1 = h_{\phi_1}(X_1),$$

so that $\omega$ modulates the composite function $h_{\phi_1} \circ f_{\text{gen}}$. This makes the true contributions of unique information and redundancy highly nonlinear, and the estimated components $U_1, U_2, R, S$ capture the resulting influence after such transformations. Hence, exact numerical consistency with the original linear weights is not expected.

Nevertheless, we design our experiments to test whether the estimated PID components *reliably reflect the relative strengths* of the underlying generative factors, which we expect to be positively correlated with the true weights. The results in Fig. 2, Fig. 10 and Fig. 11 confirm clearly that: 1) when a particular component is deactivated, PIDReg correctly identifies its absence, e.g.,

$$\omega_{u1} = 0 \implies U_1 \approx 0; \tag{68}$$

and 2) when the relative strengths of two components are varied, the estimated PID values faithfully track the corresponding monotonic trends, e.g.,

$$w_s^{(2)} > w_s^{(1)}, \quad w_r^{(2)} < w_r^{(1)} \implies S_{\text{est}}^{(2)} > S_{\text{est}}^{(1)}, \ R_{\text{est}}^{(2)} < R_{\text{est}}^{(1)}. \tag{69}$$

### G.5 Sensitivity Analysis of $\lambda$

We conduct a sensitivity analysis of the Gaussian regularization using the ***Superconductivity*** as an example. Following the same setup as in the main paper, we set $\lambda_1 = \lambda_2 = \lambda_3 = \lambda$ and evaluate

| $\lambda$ | RMSE | Corr | $U_1$ | $U_2$ | $R$ | $S$ |
|---|---|---|---|---|---|---|
| 0.01 | 10.405 | 0.952 | 0.369 | 0.001 | 0.847 | 0.378 |
| 0.05 | 10.492 | 0.951 | 0.372 | 0.000 | 0.856 | 0.380 |
| 0.1 | 10.370 | 0.952 | 0.375 | 0.001 | 0.878 | 0.394 |
| 0.5 | 10.593 | 0.950 | 0.375 | 0.001 | 0.854 | 0.379 |
| 1.0 | 10.588 | 0.951 | 0.375 | 0.001 | 0.853 | 0.380 |
| Relative Change | 2.15% | 0.21% | 1.63% | 0 | 3.66% | 4.23% |

Table 17: Sensitivity Analysis of $\lambda$.

$\lambda \in 0.01, , 0.05, , 0.1, , 0.5, , 1.0$. The corresponding predictive performance and PID decomposition results are reported in Table 17.

As $\lambda$ varies, all predictive metrics fluctuate by less than $2.2\%$, while the PID decomposition values change by less than $4.3\%$. The overall structure remains stable, and the interpretability is not affected. These results indicate that the model is highly robust with respect to $\lambda$.

### G.6 ALTERNATIVE WAYS FOR MODELING SYNERGY

The Hadamard product is the mechanism we adopt in Eq. 2 to model synergy. To further justify the effectiveness and advantage of this design, we compare the Hadamard product with several alternative interaction mechanisms on CMU-MOSEI:

1. **Tensor product:** $T = Z_1 \otimes Z_2$, followed by a linear projection $\widetilde{Z} = W_{\text{proj}} \cdot \text{vec}(T)$ back to $\mathbb{R}^d$.

2. **2D convolutions:** reshape $Z_1, Z_2$ into $\sqrt{d} \times \sqrt{d}$ maps, stack them as a 2-channel input, apply a small Conv2D–BN–ReLU stack, and adaptively pool back to $\mathbb{R}^d$.

3. **Bilinear fusion:** learns a set of $d$ matrices $\{W_k\}_{k=1}^d$, each of size $d \times d$, and computes the $k$-th dimension of the fused feature as $\widetilde{Z}_k = Z_1^\top W_k Z_2 + b_k = \sum_{i=1}^d \sum_{j=1}^d Z_{1,i} W_{k,ij} Z_{2,j} + b_k$. This bilinear form explicitly captures second-order interactions between the two latent vectors, allowing $\widetilde{Z}$ to encode cross-dimension multiplicative dependencies that cannot be modeled by additive fusion (e.g., concatenation or averaging).

4. **Concat + MLP:** $\widetilde{Z} = \text{MLP}([Z_1; Z_2])$.

5. **Concat + attention:** a cross-attention block with $Q = Z_1$, $K = Z_2$, $V = Z_2$, followed by multi-head attention, FFN, and residual LayerNorm to obtain $\widetilde{Z}$.

All variants are plugged into the same fusion template:

$$Z = w_1 \cdot Z_1 + w_2 \cdot Z_2 + w_3 \cdot \widetilde{Z}. \tag{70}$$

To approximate synergy-heavy scenarios, we construct a conflict subset where text-only and vision-only predictors disagree. Concretely, we first train unimodal predictors for text and vision; a sample is flagged as conflict if:

$$\text{sign}(\hat{y}_i^{\text{text}}) \neq \text{sign}(\hat{y}_i^{\text{vision}}) \quad \text{or} \quad \hat{y}_i^{\text{text}} - \hat{y}_i^{\text{vision}} > \delta, \tag{71}$$

where:

$$\delta = \max\big(0.3, \, \text{median}(\hat{y}^{\text{text}} - \hat{y}^{\text{vision}})\big). \tag{72}$$

These conflict samples require stronger cross-modal reasoning to resolve modality disagreement, and are a reasonable proxy for sarcasm-like synergy cases. The results are shown in Table 18.

As can be seen, the Hadamard fusion provides the most stable and competitive performance across both the full *Text+Vision* dataset and the challenging *Conflict Subset*. Especially, on the *Conflict Subset*, where synergistic reasoning is crucial, the performance of all baselines degrades significantly, yet Hadamard fusion remains one of the most resilient. While more complex fusion strategies (e.g., tensor products, bilinear layers, 2D convolutions, or attention-based fusion) often degrade substantially on conflict samples, Hadamard fusion retains the best correlation and consistently ranks first

|  | $A_7$ | $A_2$ | $F_1$ | MAE | Corr |
|---|---|---|---|---|---|
| **Text+Vision** | | | | | |
| Hadamard | 47.0 | **80.6** | **80.2** | 0.642 | 0.661 |
| Tensor Products | 45.7 | 72.2 | 72.4 | 0.682 | 0.632 |
| 2D Convolutions | 28.6 | 66.1 | 52.6 | 0.894 | 0.616 |
| Bilinear | **47.6** | 67.3 | 73.1 | **0.636** | **0.671** |
| Concat+MLP | 46.5 | 72.2 | 74.5 | 0.669 | 0.656 |
| Concat+Attention | 46.7 | 69.7 | 73.4 | 0.645 | 0.647 |
| **Conflict Subset** | | | | | |
| Hadamard | 45.3 | 62.0 | 59.9 | 0.680 | **0.459** |
| Tensor Products | 46.2 | 60.8 | 60.3 | **0.672** | 0.450 |
| 2D Convolutions | **46.9** | 63.0 | 55.0 | 0.711 | 0.406 |
| Bilinear | 30.4 | 62.9 | 29.7 | 0.894 | 0.389 |
| Concat+MLP | 45.4 | 56.8 | **60.6** | 0.705 | 0.439 |
| Concat+Attention | 45.3 | **64.2** | 60.4 | 0.755 | 0.431 |

Table 18: Synergy Modeling Comparison.

or second on the classification metrics. Its advantages stem from being parameter-free and computationally efficient, avoiding the heavy $O(d^2)$-$O(d^3)$ parameterization required by bilinear or tensor-product fusion. This lightweight multiplicative interaction generalizes better and prevents overfitting to spurious cross-modal patterns, making Hadamard fusion particularly robust when modalities disagree.

### G.7 COMPARISON WITH POST-HOC EXPLAINATION

To assess whether our computed PID values (e.g., whether redundancy or synergy dominates, or whether one of the unique information terms tends toward zero) align with existing post-hoc explanation approaches, we compare against PID-BATCH (Liang et al., 2023), which quantifies interaction information only after the model has been trained. In contrast to PID-BATCH, where PID estimation is performed independently from the design of the multimodal fusion architecture, our PIDReg integrates both aspects into a unified framework. It develops a novel multimodal fusion model together with a principled, built-in PID computation mechanism, thereby providing *built-in* explainability by design rather than relying on *post-hoc* analysis.

We first consider a synthetic data following the experimental protocol in Section G.4.2. We compute the information decomposition values with PID-BATCH, with results summarized in Fig. 14. Notably, under skewed and non-Gaussian latent distributions, the estimated values fail to reflect the intended one-dimensional variations across all settings: modifying a single ground-truth component often induces changes in multiple estimated components; the relative strengths of synergy and redundancy are frequently reversed; and non-zero estimates appear even when the true weight is zero. These findings indicate that the post-hoc PID method is unable to reliably recover the underlying interaction structure.

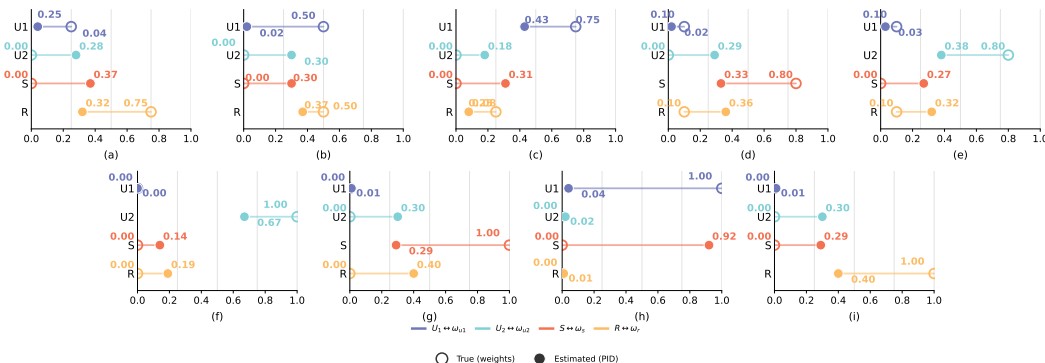

Figure 14: Post-hoc Diagnostic Estimation of PID Components.

Furthermore, on **CMU-MOSI** and **CMU-MOSEI**, we compare the PID values produced by PIDReg during the regression process with those from PID-BATCH. The results are summarized in Table 19.

| | CMU-MOSI | | | | | | | |
|---|---|---|---|---|---|---|---|---|
| | PID-BATCH | | | | PIDReg | | | |
| **Modality Pair** | $U_1$ | $U_2$ | $R$ | $S$ | $U_1$ | $U_2$ | $R$ | $S$ |
| Text + Vision | 0 | 0.04 | 0.03 | 0.24 | 0.06 | 0.05 | 1.38 | 4.23 |
| Text + Audio | 0.01 | 0 | 0.03 | 0.24 | 0.21 | 0.10 | 9.15 | 0.31 |
| Vision + Audio | 0.05 | 0 | 0.03 | 0.20 | 0.15 | 0.02 | 9.15 | 0.33 |
| | CMU-MOSEI | | | | | | | |
| | PID-BATCH | | | | PIDReg | | | |
| **Modality Pair** | $U_1$ | $U_2$ | $R$ | $S$ | $U_1$ | $U_2$ | $R$ | $S$ |
| Text + Vision | 0.09 | 0.04 | 0.22 | 0.13 | 0.03 | 0.02 | 0.32 | 0.69 |
| Text + Audio | 0.70 | 0 | 0.30 | 0 | 0.03 | 0.02 | 0.19 | 0.30 |
| Vision + Audio | 0.74 | 0 | 0.26 | 0 | 0.15 | 0.02 | 9.15 | 0.33 |

Table 19: Comparison of PID values between PID-BATCH and PIDReg on CMU-MOSI and CMU-MOSEI.

We then extend evaluation to real-world data. On **CMU-MOSEI**, we observe that the two methods are highly consistent: both identify Text as the dominant modality for the task, Vision as carrying more unique information than Audio, and both modalities exhibiting a high degree of overlap (redundancy).

On **CMU-MOSI**, both PID-BATCH and PIDReg indicate strong synergy for the "Text + Vision" pair. However, for "Text + Audio" and "Audio + Vision," PID-BATCH consistently attributes the interaction to high synergy, whereas PIDReg identifies these pairs as being primarily redundancy-dominated. To assess which decomposition better reflects the actual predictive behavior, we train three unimodal predictors (using the same encoders and predictors as in PIDReg) based on Text, Vision, and Audio alone. The results are reported in Table 20.

As shown in the table, combining "Text + Vision" indeed yields a noticeable improvement over either modality alone. In contrast, combining "Audio + Vision" does not provide any performance gain beyond the Audio modality, suggesting near-zero synergy, which is contrary to the high synergy assigned by PID-BATCH. A similar pattern is observed for "Text + Audio," where the performance improvement over Text alone is marginal. These observations align with our PIDReg's redundancy-dominated decomposition and indicate that its PID estimates more faithfully reflect the model's actual multimodal predictive contributions.

| **Modality** | $A_2$ | $A_5$ | $F_1$ | **MAE** | **Corr** |
|---|---|---|---|---|---|
| Text | 29.6 | 77.3 | 77.4 | 0.999 | 0.659 |
| Vision | 15.3 | 53.8 | 53.9 | 1.422 | 0.073 |
| Audio | 15.5 | 52.1 | 47.7 | 1.427 | 0.229 |
| Text + Vision | 37.2 | 80.8 | 80.9 | 0.947 | 0.664 |
| Text + Audio | 32.0 | 80.0 | 79.7 | 0.938 | 0.662 |
| Audio + Vision | 16.4 | 52.3 | 51.8 | 1.400 | 0.149 |

Table 20: Unimodal and multimodal performance comparison on **CMU-MOSI**.

## H  LIMITATIONS AND FUTURE WORK

The current PIDReg framework provides modality-level or dataset-level interpretability by revealing how different higher-order modality interactions contribute to the final prediction. For future work,

we aim to extend this capability toward sample-level (instance-level) interpretability. In this setting, when making inference for a specific sample, PIDReg would be able to identify the most informative unique modality or modality interaction that drives the prediction.

On the other hand, although the present paper focuses on regression, our work is the first to embed PID directly within a multimodal model, rather than relying on PID solely as a post-hoc explanatory tool. That is, the overall framework (i.e., the iterative coupling between PID computation and predictive model training) can be naturally extended to classification settings or regression tasks with discrete targets.

To achieve this, one only needs a way to compute the PID terms in $I(Z_1, Z_2; Y)$ when $Y$ is discrete. This can be accomplished in two ways:

- **Discretization of latent variables.** One may discretize the continuous latent representations $Z_1$ and $Z_2$, such that all three variables $(Z_1, Z_2, Y)$ become discrete. In this case, the joint distribution $P(Z_1, Z_2, Y)$ is a probability mass function, and the PID components can be computed directly using the convex optimization formulation (CVX-PID) introduced in (Liang et al., 2023). The approximation accuracy depends on the number of discretization bins used for $Z_1$ and $Z_2$.
- **Logit-based continuous approximation.** Alternatively, instead of working with the discrete target $Y$ directly, one may use its logit (or pre-softmax) representation as a continuous surrogate.

## I  LLM USAGE STATEMENT

This study does not incorporate LLMs as a key, novel, or unconventional component of the proposed approach, nor in any experiments. Any use of LLMs was limited to writing refinement and had no influence on the fundamental methodology or the results.

