# OpenReview forum: "Explainable Multimodal Regression via Information Decomposition"
_ICLR.cc/2026/Conference — Submitted to ICLR 2026_

### Official Review · Reviewer_HX1r · 2025-10-21

**Soundness:** 3
**Presentation:** 3
**Contribution:** 3
**Rating:** 6
**Confidence:** 4

**Summary:**

This paper introduces PIDReg, an explainable multimodal regression framework grounded in Partial Information Decomposition (PID). The key objective is to quantify and disentangle modality contributions—unique, redundant, and synergistic—in multimodal regression tasks. To make PID tractable for high-dimensional, continuous data, the authors enforce Gaussianity in the joint latent space of modality-specific representations and the transformed target variable. This yields a closed-form analytical decomposition.

**Strengths:**

- PIDReg provides a mathematically grounded decomposition of information flow in multimodal learning.

- The Gaussian PID formulation eliminates reliance on computationally expensive variational or Monte Carlo approximations.

- PID components (unique, redundant, synergy) correspond directly to measurable modality behaviors—clearly demonstrated in Table 4 and Figures 3b–c.

- Six datasets spanning physics, healthcare, and affective computing show consistent gains in both RMSE and correlation metrics (Tables 1–3).

- Interpretability outcomes match known domain insights (e.g., sMRI dominance in brain age prediction, vision dominance in robotics).

**Weaknesses:**

- Although practical, the Gaussian approximation may restrict performance on highly multimodal or non-linear latent distributions. An empirical robustness analysis with non-Gaussian data would strengthen the claim.

- The combination of multiple regularizers (CS, Gaussianity, and conditional independence) could increase training time—though no runtime comparison is reported.

- The Hadamard product captures pairwise synergy but may overlook higher-order nonlinear dependencies.

- The framework involves several λ-weights (Eq. 17). A sensitivity study or adaptive weighting scheme could improve reproducibility.

- While Section 3.1.1 formalizes Gaussian PID, more intuition on how union information constrains redundancy would aid accessibility for non-specialists.

**Questions:**

- How sensitive is the PIDReg performance to deviations from Gaussianity? Could normalizing flows or copula-based Gaussianization improve flexibility?

- How does the computational overhead of PIDReg compare to baselines like DER or CoMM?

- Could the synergy modeling (Hadamard product) be replaced with more expressive neural cross-interaction layers?

- Can PIDReg be adapted for classification or discrete regression tasks where mutual information is not directly differentiable?

- How scalable is PIDReg to scenarios with >3 modalities and large feature dimensions?

**Details Of Ethics Concerns:**

All datasets are publicly available and ethically cleared (e.g., REST-meta-MDD and CMU-MOSEI). The method poses no ethical concerns and promotes transparency in model interpretability.

---

> ### Author Response · Authors · 2025-11-29
> **Reply to your Weaknesses**
>
> ### **Weakness 1: Empirical Results on Non-Gaussian Data**
>
> We are grateful for the reviewer's question. We would like to clarify that PIDReg has **already** been tested on data in which the latents are non-Gaussian or even discrete:
>
> **(a) Strongly non-Gaussian latent factors.** In **Appendix G.4.2** and **Appendix G.4.3**, we construct a synthetic setting where the true information-carrying latent sources are strongly non-Gaussian and even discrete, as can be seen in **Figs. (11)-(13)**, PIDReg still faithfully recovers the correct relative structure of the true PID decomposition.
>
> **(b) Real-world non-Gaussian data.** In all real-data experiments (superconductivity, CT slice, MOSI, MOSEI, neuroscience), the original data distributions and their underlying latents are typically away from Gaussian. Our PIDReg just assumes the latent representations follow a joint Gaussian, which essentially implies that we rely on the 1st and 2nd order data statistics to estimate PID values, which is reasonable given that the first two moments typically capture the majority of meaningful data structure.
>
> ---
>
> ### **Weakness 2: Clarification on the Existence of the Reported Runtime Comparison**
>
> As shown in **Appendix G.1** (Table 13), we have **already** provided a detailed runtime comparison. The computational overhead is modest given that PIDReg provides stable and theoretically grounded interpretability, a property absent in other methods, additional computational cost introduced by the three regularization terms $𝓛_{\\mathrm{CS}}$, $𝓛_{\\mathrm{Gauss}}$, $𝓛_{\\mathrm{CMI}}$ is incurred only in the low-dimensional bottleneck space $Z_1, Z_2, Y$, and does not introduce any extra trainable parameters.
>
> ---
>
> ### **Weakness 3: Synergy Modeling**
> Hadamard product is a standard, efficient, and parameter-free interaction mechanism widely used in multimodal fusion (drug synergy, VQA, sentiment analysis, etc.; see Sec. 3.1).
>
> We further evaluate our approach on both the full CMU-MOSEI dataset and a modality-conflict “sarcasm” subset, where synergistic reasoning is more crucial because two modalities may provide conflicting or incomplete cues, making no single modality reliable on its own. As shown in **Appendix G.6** or the following table, we compare our Hadamard product with five alternatives, including tensor product, bilinear fusion, and attention-based fusion, Hadamard fusion consistently delivers a clear and stable performance advantage on both the full dataset and the subset. Bilinear fusion performs well on the full dataset but degrades on the sarcasm subset, whereas tensor product fusion shows strong synergy detection but weaker overall performance.
>
> |  | $A_7$ | $A_2$ | $F1$ | $MAE$ | $Corr$ |
> |:---:|:---:|:---:|:---:|:---:|:---:|
> |  |  |  | **Text+Vision** |  |  |
> | Hadamard | 47.0 | 80.6 | 80.2 | 0.642 | 0.661 |
> | Tensor Products | 45.7 | 72.2 | 72.4 | 0.682 | 0.632 |
> | 2D Convolutions | 28.6 | 66.1 | 52.6 | 0.894 | 0.616 |
> | Bilinear | 47.6 | 67.3 | 73.1 | 0.636 | 0.671 |
> | Concat+MLP | 46.5 | 72.2 | 74.5 | 0.669 | 0.656 |
> | Concat+Attention | 46.7 | 69.7 | 73.4 | 0.645 | 0.647 |
> |  |  |  | **Conflict Subset** |  |  |
> | Hadamard | 45.3 | 62.0 | 59.9 | 0.680 | 0.459 |
> | Tensor Products | 46.2 | 60.8 | 80.3 | 0.672 | 0.450 |
> | 2D Convolutions | 46.9 | 63.0 | 55.0 | 0.711 | 0.466 |
> | Bilinear | 30.4 | 62.9 | 29.7 | 0.894 | 0.389 |
> | Concat+MLP | 45.4 | 56.8 | 60.6 | 0.705 | 0.439 |
> | Concat+Attention | 45.3 | 64.2 | 60.4 | 0.755 | 0.431 |
>
> ---
>
> ### **Weakness 4: Sensitivity study of $\lambda$**
> We observed that, within each mini-batch, the raw values of $𝓛_{\mathrm{pred}}$, $𝓛_{\mathrm{Gauss}}$, $𝓛_{\mathrm{CMI}}$, and $𝓛_{\mathrm{CS}}$  naturally fall within a similar scale and are of MSE-comparable magnitude. This observation motivates using a shared $\lambda$ for all three regularization terms, which also reduces the number of hyperparameters. We provide a detailed $\lambda$-sensitivity analysis in **Appendix G.5** using the superconductivity real-world dataset, which further justifies its robustness.
>
> ---
>
> ### **Weakness 5: Intuition Explanation of Union Information Constrains Redundancy**
>
> We thank the reviewer for this valuable suggestion. We provide a more intuitive explanation of how union information constrains redundancy in Section 3.1. Please also refer to **Appendix A.1** in the revised manuscript for details.
>
> In short, union information provides an operational lower bound on “shared evidence” that cannot be removed by any permissible coupling, which is why it serves as the redundancy term in PID.

---

> ### Author Response · Authors · 2025-11-29
> **Reply to your Questions**
>
> ### **Question 1: Empirical Results on Non-Gaussian Data and Alternatives to Gaussianization**
>
> Regarding performance under deviations from Gaussian, please refer to **Weakness 1**. We appreciate the theoretical suggestion. However, normalizing flows risk distorting encoder fidelity through parameter overhead, and copula-based CDF estimation is numerically unstable on high-dimensional minibatches. Our lightweight design avoids these complexities, robustly achieving distribution alignment with superior efficiency and stability.
>
> ---
>
> ### **Question 2: Computational Efficiency**
>
> For clarifications regarding computational efficiency, please refer to **Weakness 2**. Additionally, we have supplemented Table 13 with the computational cost of CoMM to address the reviewer's concerns about its efficiency.
>
> | Method | DER | MoNIG | CoMM | PIDReg |
> |--------|-----|-------|------|--------|
> | Epoch Time (s) | 1.756 | 3.280 | 3.240 | 4.641 |
>
> ---
>
> ### **Question 3: Synergy Modeling**
>
> Please refer to **Weakness 3**.
>
> ---
>
> ### **Question 4: Extension to classification or discrete regression tasks**
>
> We thank the reviewer for the insightful question.
>
> As noted by Reviewer KZFk, our work is the first to embed a partial information decomposition directly into a multimodal model, rather than employing PID purely as a post-hoc explanatory tool. The overall framework (i.e., the iterative interaction between PID computation and predictive model training) naturally extends to classification or discrete regression tasks.
>
> To achieve this, one just needs to have a way to estimate PID terms in $I(Z1,Z2; Y)$ when $Y$ is discrete.
> This can be achieved in two possible ways:
> 1) **Discretization of the latent variables**: To handle a discrete target variable $Y$, one may also discretize the continuous latent representations $Z_1$ and $Z_2$, such that all three variables $(Z_1, Z_2, Y)$ become discrete. In this case, the joint distribution $P(Z_1, Z_2, Y)$ is a probability mass function, and the PID components can be computed directly using the convex optimization formulation described in [1]. The approximation accuracy depends on the number of discretization bins applied to $Z_1$ and $Z_2$.
>
>
> 2) Approximate the discrete label $Y$ using its logit representation, which is continuous.
>
> ---
>
> ### **Question 5: Scalable to Large Feature Dimensions and Modalities**
>
> **Scaling to More Than Three Modalities ($N \\rightarrow \\infty$)**
>
> We feel there is a **misunderstanding** here, as we have clearly explains the way of extending PIDReg to more than three modalities in Appendix F.
> The experimental results are also provided.
>
> Extending PID to $N > 3$ modalities is theoretically challenging due to a combinatorial explosion in the number of PID lattice terms. As noted in Appendix F, the number of terms grows super-exponentially with $N$: already 18 terms for $N = 3$, and hundreds for $N = 4$.
> Any attempt to compute the full PID for all terms becomes theoretically infeasible when $N > 3$.
> The only viable path is the pragmatic simplification we proposed in Appendix F, which reduces the PID structure for $N$ modalities to $N + 2$ key quantities:
>
> 1. $U_1, U_2, \\ldots, U_N$ (unique information from each modality)
> 2. A global redundancy term $R$
> 3. A global synergy $S$
>
> This simplification preserves interpretability while avoiding the combinatorial blow-up inherent in full multivariate PID. We further demonstrate the scalability of PIDReg to multimodal inputs in Appendix F.5, where we evaluate the three-modality version of PIDReg on the CMU-MOSI and CMU-MOSEI datasets.
>
> **Scaling to High-Dimensional Features ($D \\rightarrow \\infty$)**
>
> Our method scales well with feature dimensionality. By design, PIDReg is entirely free from computational bottlenecks tied to the dimensionality of the raw input. Regardless of how large $D$ is, each high-dimensional input $X_m \\in \\mathbb{R}^D$ is first projected by a deep nonlinear encoder $h_{\\phi_m}$ into a fixed-size, low-dimensional latent space $Z_m \\in \\mathbb{R}^d$. All core components including $𝓛_{\\mathrm{Gauss}}$, $𝓛_{\\mathrm{CS}}$, $𝓛_{\\mathrm{CMI}}$, and the Gaussian PID solver, operate solely within this latent space.
>
> We have already demonstrated this scalability in our experiments, such as REST-meta-MDD: handles extremely high-dimensional 3D sMRI volumetric data (millions of voxels) and graph-structured fMRI data, as well as Vision & Touch etc.
>
> [1] Liang, Paul Pu, et al. "Quantifying & modeling multimodal interactions: An information decomposition framework." NeurIPS, 2023.

---

### Official Review · Reviewer_DLax · 2025-10-28

**Soundness:** 3
**Presentation:** 3
**Contribution:** 3
**Rating:** 2
**Confidence:** 3

**Summary:**

This paper proposes PIDReg, a multimodal regression framework integrating Partial Information Decomposition (PID) to address interpretability gaps. It decomposes modality-specific info into unique (U), redundant (R), and synergistic (S) components, enforces Gaussianity on latent distributions for analytical PID computation, and designs CS divergence/CMI regularizers. Experiments on 6 datasets (healthcare, physics, etc.) show it outperforms SOTA in accuracy and interpretability.

**Strengths:**

1.	The paper innovatively integrates Partial Information Decomposition (PID) into multimodal regression, solving PID’s underdeterminacy via Gaussianity enforcement on latent distributions (enabling analytical PID computation) and combining CS divergence/CMI regularizers—overcoming prior limitations of PID in high-dimensional continuous data. It also proposes a two-stage optimization for stable fusion weight learning, a creative combination of interpretability and regression.

2.	Theoretically rigorous (e.g., closed-form CMI estimator, Gaussian PID derivation); experimentally comprehensive (6 cross-domain datasets, synthetic data validation, ablation studies for regularizers/bottleneck) to confirm reliability; results align with domain knowledge (e.g., sMRI dominance in brain age prediction).

3.	Well-structured (abstract→method→experiments→conclusion); key terms (PID components, CS divergence) defined clearly; appendices detail proofs/experimental setups; tables/figures (e.g., PID estimates, ablation results) intuitively support claims.

4.	Fills multimodal interpretability gaps (intrinsic vs. post-hoc XAI); yields domain insights (e.g., Vision+Text synergy in sentiment analysis); scalable to multi-modality; aids practical tasks (modality selection for efficient inference), with value for healthcare/robotics.

**Weaknesses:**

1.	Lacks validation on data with inherently non-Gaussian latents (e.g., discrete event-based robotics data). No experiments on scenarios where latent non-Gaussianity (e.g., multi-modal signals) might bias PID decomposition, missing supplementary tests (e.g., bimodal MNIST) to quantify impact.

2.	Critical params (PID convergence $K=5/\delta^t<0.01$, regularization $\lambda_1=\lambda_2=\lambda_3=0.1$) lack sensitivity analysis. No comparison of performance across param values (e.g., $K=3$ or $\lambda=0.01$) to confirm robustness, hindering reproducibility.

3.	Higher training time (4.64s/epoch vs. DER’s 1.76s) lacks optimization exploration (e.g., mini-batch covariance estimation). No data on scalability to large datasets (100k+ samples) or high-dimensional latents ($d>256$), limiting real-world adoption.

4.	The linear-noise IB ($Z_m=\lambda_m R_m+(1-\lambda_m)\epsilon_m$) lacks comparison to nonlinear IB variants (e.g., adaptive noise scaling). No analysis of how $\lambda_m$ convergence (Table 7) correlates with modality quality (e.g., noisy vs. clean modalities), missing opportunities to link bottleneck behavior to data characteristics.

**Questions:**

1.	Did you conduct ablation experiments comparing the Hadamard product to these alternatives (e.g., tensor products, 2D convolutions for interaction modeling) on datasets with known synergistic patterns? For example, in CMU-MOSEI (Section 4.2), where Vision+Text synergy aids sarcasm detection, does the Hadamard product outperform other methods in capturing this synergy (e.g., via higher S alignment with human-annotated sarcasm cases)?

2.	Did you evaluate PIDReg on a dataset with known non-Gaussian latents—e.g., a robotics dataset where latent $Z_1$ (visual) is continuous but $Z_2$ (tactile) is discrete (binary contact/no-contact)? If so, how did Gaussian enforcement impact PID decomposition (e.g., did it underestimate discrete tactile synergy) and predictive accuracy?

3.	For a dataset like CMU-MOSI (Audio &Text), did you run a post-hoc method (e.g., kernel SHAP) on PIDReg’s predictions to quantify explanation consistency?

4.	For datasets with already Gaussian targets (e.g., synthetic data in Section 4.1), does the transformation provide any benefit (e.g., faster PID convergence) or introduce unnecessary bias?

5.	Have you tested PIDReg on datasets where latent representations are inherently non-Gaussian? If yes, does Gaussianity enforcement lead to PID decomposition biases  or a decline in prediction accuracy? If not tested, have you considered relaxing the Gaussian assumption  to adapt to such scenarios while still ensuring the analyticity of PID computation?

---

> ### Author Response · Authors · 2025-11-29
> **Reply to your Weaknesses 1 and 2**
>
> We appreciate that review feel our work has theoretical rigorous, is well-structured and fills multimodal interpretability gaps.
> However, we find that there is a **factual misunderstanding** regarding our experiment (especially Weakness 1). **We DO have evaluations on non-Gaussian latents for both synthetic and real-world data**. Please also refer to our following reply for details.
>
> ### **Weakness 1: Clarification on Non-Gaussian Latents and Bimodal MNIST Validation**
>
> PIDReg has been tested on data in which the latents are non-Gaussian or even discrete:
>
> **(a) Strongly non-Gaussian latent factors.** In **Appendix G.4.2** and **Appendix G.4.3**, we construct a synthetic setting where the true information-carrying latent sources are strongly non-Gaussian and even discrete, as can be seen in Figs. (11)-(13), PIDReg still faithfully recovers the correct relative structure of the true PID decomposition.
>
> **(b) Real-world non-Gaussian data.** In all real-data experiments (superconductivity, CT slice, MOSI, MOSEI, neuroscience), the original data distributions and their underlying latents, are typically away from Gaussian.
> Our PIDReg just assumes the latent representations follow a joint Gaussian, which essentially implies that we rely on the 1st and 2nd order data statistics to estimate PID values, which is reasonable given that the first two moments typically capture the majority of meaningful data structure.
>
> To further demonstrate that our Gaussian regularization ensures a rigorous implementation of PIDReg, we additionally perform a Gaussian regularizations ablation on a synthetic dataset with non-Gaussian latent variables (please refer to **Appendix G.4.2**, Table 16 or the table below)
>
> |  | $U_1$ | $U_2$ | $R$ | $S$ |
> |:---:|:---:|:---:|:---:|:---:|
> | Baseline | 0.04 | 0.02 | 0.31 | 0.63 |
> | No $𝓛_{CS}$ | 0.06 | 0.03 | 0.26 | 0.56 |
> | No $𝓛_{Gauss}$ | 0.05 | 0.03 | 0.25 | 0.54 |
> | No $𝓛_{CS}, 𝓛_{Gauss}$ | 0.03 | 0.03 | 0.26 | 0.49 |
>
> As can be seen, removing the Gaussian regularizations does not lead to large deviations in the estimated PID values.
> We only see a slight underestimation of synergy, which is expected when relying solely on first- and second-order statistics.
>
> Regarding missing supplementary tests like bimodal MNIST: **this experiment is already included in Appendix G.3 of the original manuscript**.
> PIDReg achieves state-of-the-art performance, with an intuitive and verifiable PID decomposition: the raw pixel modality $U_1 = 0.866$ carries all unique information, while the derived feature modality $U_2 = 0.000$ adds no new unique information. In the revised version, **Appendix G.3** also visualizes the non-Gaussianity of $X_1$, $X_2$ and the near-Gaussianity of the learned $Z_1$, $Z_2$, providing direct evidence for our representation-level Gaussianization view.
>
> ---
>
> ### **Weakness 2: Hyperparameter Sensitivity and Robustness**
>
> We observed that, within each mini-batch, the raw values of $𝓛_{\mathrm{pred}}$, $𝓛_{\mathrm{Gauss}}$, $𝓛_{\mathrm{CMI}}$, and $𝓛_{\mathrm{CS}}$
> naturally fall within a similar scale and are of MSE-comparable magnitude. This observation motivates using a shared $\lambda$ for all three regularization terms, which also reduces the number of hyperparameters.
>
>
> We provide a detailed $\lambda$-sensitivity analysis in **Appendix G.5** using the superconductivity real-world dataset, which further justifies its robustness (see also Table below).
>
> | $\lambda$ | RMSE | Corr | U$_1$ | U$_2$ | R | S |
> |:---:|:---:|:---:|:---:|:---:|:---:|:---:|
> | 0.01 | 10.405 | 0.952 | 0.369 | 0.001 | 0.847 | 0.378 |
> | 0.05 | 10.492 | 0.951 | 0.372 | 0.000 | 0.856 | 0.380 |
> | 0.1  | 10.370 | 0.952 | 0.375 | 0.001 | 0.878 | 0.394 |
> | 0.5  | 10.593 | 0.950 | 0.375 | 0.001 | 0.854 | 0.379 |
> | 1.0  | 10.588 | 0.951 | 0.375 | 0.001 | 0.853 | 0.380 |
> | Relative Change | 2.15% | 0.21% | 1.63% | 0 | 3.66% | 4.23% |
>
>
> Regarding the sensitivity of the PID convergence parameter $K$, it simply controls the number of epochs in Stage I training.
> As illustrated in Fig. 3(b)–(c), the PID decomposition in Stage I is intended to eventually converge to a stable plateau. Consequently, if $K$ is set to a relatively large value, it does not affect the final optimization outcome.
> That is, as long as training proceeds for enough epochs, different choices of $K$ yields negligibly different final performance and encoder representations.

---

> ### Author Response · Authors · 2025-11-29
> **Reply to your Weakness 3**
>
> ### **Weakness 3: Training Time, Optimization Strategy, and Scalability, high-dimensional latents**
>
> **Training Time:** PIDReg (4.64 s/epoch) is slower than DER (1.76 s) and MoNIG (3.28 s), but all methods remain in the same order of magnitude. DER and MoNIG behave as black-box models, whereas PIDReg trades a moderate and controlled overhead for built-in PID decomposition and interpretability. In practice, this gap does not constitute a practical barrier to adoption, especially given the additional explanatory power PIDReg provides.
>
> **Optimization Strategy:** On optimization and mini-batch covariance estimation, all covariance matrices $\Sigma^2$ and kernel matrices used in $𝓛_{Gauss}$, $𝓛_{CMI}$, and the Gaussian PID module are computed on mini-batches.
>
> **Scalability:** **The statement that we provide no data on scalability to large datasets (100k+ samples) reflects a factual misunderstanding**. PIDReg has already been evaluated on several large-scale settings. The *REST-meta-MDD* dataset is the largest publicly available multimodal neuroimaging dataset for major depressive disorder. It contains over 2400 subjects with both sMRI and rs-fMRI data. Even when considering only sMRI and counting individual 2D slices, the dataset includes more than $2k \times 180 = 360k$ sliced images, far exceeding the 100k scale. Similarly, the *Vision\&Touch* robotic dataset includes 150 trajectories totaling $147k$ synchronized timesteps of visual, tactile, and proprioceptive signals. In addition, we constructed a *bimodal MNIST* regression benchmark (Appendix G.3) with $140k$ samples. Moreover, *CMU-MOSEI* remains one of the largest multimodal sentiment analysis datasets in the current literature. We conducted an exhaustive search and did not find any other larger multimodal datasets suitable for regression tasks.
>
> **High-dimensional latents.** We see a conceptual mismatch between generative modeling and our regression-focused setting. In our case, a latent dimension d=64 is already heavily over-parameterized relative to the target dimension $d_y$, and fully sufficient to encode all task-relevant information. Choosing a moderate latent dimension forces the encoder to extract predictive information, improving the signal-to-noise ratio of the PID decomposition. Hence, our latent size reflects a principled design choice, not a scalability limitation. We additionally report results on the Superconductivity dataset where the latent dimensionality is expanded to 92. The overall trend remains the same: $U_2$ (the unique information from the 2nd modality) is negligible, and redundancy always outweighs synergy. A slight increase in redundancy and synergy is expected, given that the increased latent dimensionality encourages more multimodal interactions.
>
> | Latent | $RMSE$ | $Corr$ | $U_1$ | $U_2$ | $R$ | $S$ |
> |--------|--------|--------|-------|-------|-----|-----|
> | 64     | 10.37  | 0.953  | 0.373 | 0.001 | 0.878 | 0.394 |
> | 92     | 10.18  | 0.954  | 0.273 | 0.000 | 1.098 | 0.710 |

---

> ### Author Response · Authors · 2025-11-29
> **Reply to your Weakness 4**
>
> ---
>
> ### **Weakness 4: On Linear-Noise IB, Nonlinear Variants, and Modality Quality Behavior**
>
> First, we would like to clarify that our IB parameterization is already adaptive and nonlinear: for each modality, the bottleneck is $Z_m = \lambda_m R_m + (1 - \lambda_m)\epsilon_m$, where $R_m$ is deep nonlinearity encoder. Thus, the effective noise level is learned at the modality level via $\lambda_m$; the linear part is only a simple gating, not a fixed, hand-designed noise schedule.
>
> We acknowledge that variational IB (VIB) [1] is an alternative way to impose bottleneck regularization. We add a controlled comparison with a VIB-style variant (PIDReg-VIB) in the synthetic setting of Section 4.1(a) and report RMSE, R², the Gaussianity statistic W (from $𝓛_{Gauss}$), the aggregate variance $\sigma_{\text{overall}}^2$ of the PID components:
>
> | Variant | $RMSE$ | $R^2$ | $U_1$ | $U_2$ | $R$ | $S$ | $\sigma_{\text{overall}}^2$ | $W$ Statistic |
> |---------|--------|-------|-------|-------|-----|-----|------------------------|---------------|
> | PIDReg  | 10.37  | 0.952 | 0.375 | 0    | 0.878 | 0.394 | 0.089 | 0.974 |
> | PIDReg-VIB | 13.57 | 0.848 | 0.385 | 0 | 0.855 | 0.509 | 0.111 | 0.988 |
>
> Results show that, compared with VIB, Gating IB provides better structure preservation while still maintaining a good Gaussian approximation.
>
> Regarding the link between convergence behavior and modality quality, a noisier input should induce a stronger effective bottleneck. Following the synthetic setup in Section 4.1, we generate
>
> $$X_1 = \tanh\left([R, U_1] W^{(1)} + b^{(1)}\right)W^{(2)} + b^{(2)} + \epsilon_1, \epsilon_1 \sim \mathcal{N}(0, \sigma_1^2 I_{d_x})$$
>
> and analogously for $X_2$. We systematically vary $\sigma_1 = \sigma_2 \in \{0.1, 0.2, 0.3, 0.5, 0.7, 1.0, 1.2, 1.5\}$ under the setting $w_{u1} = 0, w_{u2} = 0, w_s = 0.75, w_r = 0.25$, and repeat the experiments:
>
> | $noise$ | $\lambda_1$ | $\lambda_2$ | $RMSE$ | $R^2$ | $U_1$ | $U_2$ | $R$ | $S$ |
> |---------|-------------|-------------|--------|-------|-------|-------|-----|-----|
> | 0.1     | 0.921       | 0.918       | 0.191  | 0.936 | 0.040 | 0.025 | 0.239 | 0.335 |
> | 0.2     | 0.917       | 0.898       | 0.338  | 0.793 | 0.034 | 0.025 | 0.286 | 0.655 |
> | 0.3     | 0.904       | 0.897       | 0.494  | 0.604 | 0.026 | 0.028 | 0.281 | 0.665 |
> | 0.5     | 0.903       | 0.890       | 0.668  | 0.277 | 0.020 | 0.029 | 0.278 | 0.673 |
> | 0.7     | 0.903       | 0.887       | 0.774  | 0.029 | 0.033 | 0.025 | 0.265 | 0.677 |
> | 1.0     | 0.844       | 0.834       | 0.807  | -0.057 | 0.021 | 0.038 | 0.268 | 0.673 |
> | 1.2     | 0.842       | 0.819       | 0.798  | -0.033 | 0.033 | 0.029 | 0.263 | 0.675 |
> | 1.5     | 0.788       | 0.798       | 0.801  | -0.041 | 0.024 | 0.035 | 0.265 | 0.676 |
>
> **Results show that as input noise increases, the learned bottleneck behaves in a modality-aware way. The noisier the input, the stronger the bottleneck.**
>
> We add the above analysis to **Appendix D.3**.
>
> [1] Alemi, Alexander A., et al. "Deep variational information bottleneck." ICLR, 2017.

---

> ### Author Response · Authors · 2025-11-29
> **Reply to your Questions 1 and 2**
>
> ---
>
> ### **Q1: Hadamard Product, Alternative Synergy Mechanisms, Sarcasm Detection**
>
> Using the Hadamard product to model synergy is simple, but it is not an arbitrary design choice. As discussed in Section 3.1, it has been successfully applied in domains, such as drug synergy prediction, visual question answering, and multimodal sentiment analysis.
>
> In the rebuttal, we compare the Hadamard product with five alternative synergy mechanisms, including tensor product, bilinear fusion, and attention-based fusion. As shown in **Appendix G.6**, the Hadamard product demonstrates a clear and consistent performance advantage.
>
> According to Table 16 or the following table, Hadamard fusion achieves the most stable and competitive performance on both the full CMU-MOSEI dataset and the challenging sarcasm subset, where synergistic reasoning is particularly critical. Bilinear fusion performs well on the full dataset but degrades on the sarcasm subset, whereas tensor product fusion shows strong synergy detection but weaker overall performance.
>
> Finally, unlike these alternative mechanisms, the Hadamard product is parameter-free, which makes it computationally efficient and less prone to overfitting.
>
> Regarding datasets with known synergistic patterns and the sarcasm scenario: we clarify that MOSI/MOSEI do not contain official sarcasm annotations.
> However, we approximate this phenomenon by constructing a modality-conflict subset that simulates sarcastic interactions, as also demonstrated in **Appendix G.6**.
>
> |  | $A_7$ | $A_2$ | $F1$ | $MAE$ | $Corr$ |
> |:---:|:---:|:---:|:---:|:---:|:---:|
> |  |  |  | **Text+Vision** |  |  |
> | Hadamard | 47.0 | 80.6 | 80.2 | 0.642 | 0.661 |
> | Tensor Products | 45.7 | 72.2 | 72.4 | 0.682 | 0.632 |
> | 2D Convolutions | 28.6 | 66.1 | 52.6 | 0.894 | 0.616 |
> | Bilinear | 47.6 | 67.3 | 73.1 | 0.636 | 0.671 |
> | Concat+MLP | 46.5 | 72.2 | 74.5 | 0.669 | 0.656 |
> | Concat+Attention | 46.7 | 69.7 | 73.4 | 0.645 | 0.647 |
> |  |  |  | **Conflict Subset** |  |  |
> | Hadamard | 45.3 | 62.0 | 59.9 | 0.680 | 0.459 |
> | Tensor Products | 46.2 | 60.8 | 80.3 | 0.672 | 0.450 |
> | 2D Convolutions | 46.9 | 63.0 | 55.0 | 0.711 | 0.466 |
> | Bilinear | 30.4 | 62.9 | 29.7 | 0.894 | 0.389 |
> | Concat+MLP | 45.4 | 56.8 | 60.6 | 0.705 | 0.439 |
> | Concat+Attention | 45.3 | 64.2 | 60.4 | 0.755 | 0.431 |
>
> ---
>
> ### **Q2: On Discrete, Non-Gaussian Latents Scenarios**
>
> **We have tested in real-world robotics data**, in which the true latents are not likely to be Gaussian.
> In the Vision\&Touch data, the tactile modality is a continuous time series that not only encodes binary contact versus no-contact events,
> but also provides continuous information such as force, pressure.

---

> ### Author Response · Authors · 2025-11-29
> **Reply to your Question 3**
>
> ---
>
> ### **Q3: On Post-hoc Explanation Consistency**
>
> We agree that testing explanation consistency with post-hoc methods is important. However, kernel SHAP (and similar methods) are designed for per-sample, feature-level attributions under approximately independent features. For multimodal sequences/data such as CMU-MOSI and CMU-MOSEI, it is nontrivial to define realistic modality-level perturbations that (i) respect temporal structure, (ii) preserve cross-modal alignment, and (iii) remain faithful to the data manifold. As a result, directly applying kernel SHAP at the modality level risks producing artifacts rather than meaningful decompositions.
>
> Using CMU-MOSI and CMU-MOSEI as examples, we employ the PID-BATCH method [1], a PID estimation approach based on convex optimization and neural estimators to obtain post-hoc PID decomposition values. We compare these with the PID values produced by PIDReg during the regression process. The results are summarized in the table below.
> **CMU-MOSI:**
> | | **PID-BATCH** | | | | **PIDReg** | | | |
> | :--- | :---: | :---: | :---: | :---: | :---: | :---: | :---: | :---: |
> | | $U_1$ | $U_2$ | $R$ | $S$ | $U_1$ | $U_2$ | $R$ | $S$ |
> | Text(Modal 1)+Vision(Modal 2) | 0 | 0.04 | 0.03 | 0.24 | 0.06 | 0.05 | 1.38 | 4.23 |
> | Text(Modal 1)+Audio(Modal 2) | 0.01 | 0 | 0.03 | 0.24 | 0.21 | 0.10 | 9.15 | 0.31 |
> | Vision(Modal 1)+Audio(Modal 2)| 0.05 | 0 | 0.03 | 0.20 | 0.15 | 0.02 | 9.15 | 0.33 |
>
> **CMU-MOSEI:**
>
> | | **PID-BATCH** | | | | **PIDReg** | | | |
> | :--- | :---: | :---: | :---: | :---: | :---: | :---: | :---: | :---: |
> | | $U_1$ | $U_2$ | $R$ | $S$ | $U_1$ | $U_2$ | $R$ | $S$ |
> | Text(Modal 1)+Vision(Modal 2) | 0.09 | 0.04 | 0.22 | 0.13 | 0.03 | 0.02 | 0.32 | 0.69 |
> | Text(Modal 1)+Audio(Modal 2) | 0.70 | 0 | 0.30 | 0 | 0.03 | 0.02 | 0.19 | 0.30 |
> | Vision(Modal 1)+Audio(Modal 2)| 0.74 | 0 | 0.26 | 0 | 0.15 | 0.02 | 9.15 | 0.33 |
>
> On CMU-MOSEI, we observe that the two methods are highly consistent: both identify Text as the dominant modality for the task, Vision as carrying more unique information than Audio, and both modalities exhibiting a high degree of overlap (redundancy).
>
> On CMU-MOSI, both PID-BATCH and PIDReg indicate strong synergy for the ``Text + Vision`` pair. However, for ``Text + Audio`` and ``Audio + Vision``, PID-BATCH consistently attributes the interaction to high synergy, whereas PIDReg identifies these pairs as being primarily redundancy-dominated. To assess which decomposition better reflects the actual predictive behavior, we train three unimodal predictors (using the same encoders and predictors as in PIDReg) based on Text, Vision, and Audio alone. The results are reported as:
>
> | Metric | $A_2$ | $A_7$ | $F_1$ | $MAE$ | $Corr$ |
> |--------|-------|-------|-------|-------|--------|
> | Text   | 29.6  | 77.3  | 77.4  | 0.999 | 0.659  |
> | Vision | 15.3  | 53.8  | 53.9  | 1.422 | 0.073  |
> | Audio  | 15.5  | 52.1  | 47.7  | 1.427 | 0.229  |
> | Text+Vision | 37.2 | 80.8 | 80.9 | 0.947 | 0.664 |
> | Text+Audio | 32.0 | 80.0 | 79.7 | 0.938 | 0.662 |
> | Audio+Vision | 16.4 | 52.3 | 51.8 | 1.400 | 0.149 |
>
> As shown in the table, combining ``Text + Vision`` indeed yields a noticeable improvement over either modality alone. In contrast, combining ``Audio + Vision`` does not provide any performance gain beyond the Audio modality, suggesting near-zero synergy, which is contrary to the high synergy assigned by PID-BATCH. A similar pattern is observed for ``Text + Audio``, where the performance improvement over Text alone is marginal. These observations align with our PIDReg’s redundancy-dominated decomposition and indicate that its PID estimates more faithfully reflect the model's actual multimodal predictive contributions.
>
> [1] Liang, Paul Pu, et al. "Quantifying & modeling multimodal interactions: An information decomposition framework." NeurIPS, 2023.

---

> ### Author Response · Authors · 2025-11-29
> **Reply to your Questions 4 and 5**
>
> ---
>
> ### **Q4: Already Near-Gaussian Scenarios**
>
> Consistent with Section 4.1, we again draw $U_1, U_2, R \sim \mathcal{N}(0, I)$. However, in this experiment the synthesis of $X_1, X_2$ no longer introduces nonlinear activations such as tanh. Instead, we construct the inputs as follows. Let $z_1 = [R, U_1]^{\top}, z_2 = [R, U_2]^{\top}$. We generate
>
> $$X_1 = \alpha(W_1 z_1 + b_1) + \epsilon_1, \epsilon_1 \sim \mathcal{N}(0, \sigma_1^2 I_{d_x})$$
> $$X_2 = \alpha(W_2 z_2 + b_2) + \epsilon_2, \epsilon_2 \sim \mathcal{N}(0, \sigma_2^2 I_{d_x})$$
>
> where $W_1, W_2 \in \mathbb{R}^{d_x \times 2}$ are random orthogonal matrices generated via QR decomposition, and $\alpha = 0.5$ is a scaling factor used to maintain numerical stability. This construction ensures that $P(X_1, X_2)$ is jointly Gaussian. The target variable $Y$ is defined as a linear combination of the latent variables, with only the multiplicative term retained to introduce synergy:
>
> $$Y = w_r R + w_{u1} U_1 + w_{u2} U_2 + w_s S + \epsilon_y, \quad \epsilon_y \sim \mathcal{N}(0, \sigma_y^2)$$
>
> where, $S = \frac{U_1 U_2 - \mu_{product}}{\sigma_{product}}$, which follows a product-normal distribution. All other settings remain unchanged, and we repeat the experiment. The results are reported in the table below:
>
> | | $w_1$ | $w_2$ | $w_{u}$ | $w_s$ | $RMSE$ | $R^2$ | $U_1$ | $U_2$ | $R$ | $S$ |
> |---|-------|-------|---------|-------|--------|-------|-------|-------|-----|-----|
> | 1 | 0.000 | 0.000 | 0.250 | 0.750 | 0.390 | 0.857 | 0.044 | 0.028 | 0.301 | 0.627 |
> | 2 | 0.000 | 0.000 | 0.500 | 0.500 | 0.480 | 0.774 | 0.029 | 0.027 | 0.436 | 0.508 |
> | 3 | 0.000 | 0.000 | 0.750 | 0.250 | 0.562 | 0.687 | 0.022 | 0.021 | 0.514 | 0.444 |
> | 4 | 0.000 | 0.800 | 0.100 | 0.100 | 0.277 | 0.921 | 0.000 | 0.721 | 0.086 | 0.193 |
> | 5 | 0.800 | 0.000 | 0.100 | 0.100 | 0.268 | 0.925 | 0.717 | 0.000 | 0.090 | 0.193 |
> | 6 | 1.000 | 0.000 | 0.000 | 0.000 | 0.247 | 0.941 | 0.756 | 0.000 | 0.076 | 0.169 |
> | 7 | 0.000 | 1.000 | 0.000 | 0.000 | 0.249 | 0.938 | 0.000 | 0.755 | 0.078 | 0.169 |
> | 8 | 0.000 | 0.000 | 1.000 | 0.000 | 0.352 | 0.875 | 0.030 | 0.038 | 0.283 | 0.649 |
> | 9 | 0.000 | 0.000 | 1.000 | 0.000 | 0.317 | 0.713 | 0.018 | 0.028 | 0.514 | 0.440 |
>
> The results show that the PID decomposition still accurately captures the relative information structure among the nearly Gaussian variables.
>
> ---
>
> ### **Q5: On Discrete, Non-Gaussian Latents Scenarios**
>
> Please refer to **Weakness 1** and **Q3**.

---

### Official Review · Reviewer_KZFk · 2025-10-29

**Soundness:** 3
**Presentation:** 3
**Contribution:** 3
**Rating:** 4
**Confidence:** 4

**Summary:**

This paper proposes PIDReg, a multimodal regression model that enforces a partial information decomposition (PID) of the learned representations. Concretely, two encoders produce latent embeddings $Z_1,Z_2$ for the two input modalities, and a fused prediction is formed as $Z = w_1Z_1 + w_2Z_2 + w_3(Z_1 \odot Z_2)$ (where $Z_1\odot Z_2$ is elementwise product modeling synergy). To make PID tractable in continuous, high-dimensional data, the authors force the joint distribution of $(Z_1,Z_2,Y)$ to be (approximately) Gaussian, via (1) a rank‐based inverse-normal transform on $Y$ and (2) a Shapiro–Wilk Gaussianity regularizer on $(Z_1,Z_2,Y)$. They then analytically compute Gaussian‐PID terms (redundancy $R$, unique $U_1,U_2$, synergy $S$) and incorporate a Cauchy–Schwarz divergence (CS) regularizer to encourage $Z_1$ to be independent of the other modality (and vice versa). Experimentally, PIDReg is evaluated on six tasks, where it reportedly achieves very low error and yields nontrivial modality‐level decompositions (e.g. high redundancy or synergy) aligned with domain expectations.

**Strengths:**

1.	**Ambitious Idea with End-to-End PID:** Embedding a partial information decomposition directly into a multimodal regression model is novel. The paper tackles the underdetermined nature of PID in continuous domains by introducing Gaussian latent constraints, which is a bold approach. As a result, PIDReg delivers intrinsic interpretability: the learned weights $(w_1,w_2,w_3)$ and computed PID terms give a clear “unique vs. redundant vs. synergistic” attribution of each modality to the prediction, rather than a black‐box fusion. This end-to-end interpretability is conceptually appealing and aligns with recent interest in multimodal interaction analysis[1].
2.	**Comprehensive Experiments:** The authors test PIDReg on diverse real-world datasets and a synthetic benchmark. They compare against some strong baselines. In every case, PIDReg attains the lowest error and highest correlation, suggesting it effectively uses multimodal information. The empirical section is thorough, covering both quantitative prediction metrics and qualitative PID decompositions.
3. **Reproducibility:** The authors report implementation details in appendices. The design choices (e.g. latent size, learning rates, batch sizes) are clearly documented. This transparency is laudable and should facilitate replication.

[1] Quantifying & Modeling Multimodal Interactions: An Information Decomposition Framework

**Weaknesses:**

1.	**Strong Gaussian Assumption:** A core assumption is that $(Z_1,Z_2,Y)$ is jointly Gaussian so that PID terms admit an analytic solution. However, this assumption is very strong and typically invalid in realistic multimodal tasks. The authors attempt to enforce it via a Shapiro–Wilk test loss, but this only tests for marginal normality. Even if each of $Z_1,Z_2,Y$ is marginally Gaussian, the joint distribution may still be far from multivariate normal (as the paper itself notes). There is no guarantee that the latent $Z_i$ have Gaussian joint statistics, especially in high dimensions and under nonlinear encoders. If the Gaussianity assumption fails, the computed PID values ($R,U,S$) are not theoretically valid.
2.	**Target Transformation Unclear:** The authors apply a rank-based inverse-normal transform to the regression target $Y$. This makes $Y$ marginally Gaussian, but it is highly unusual in regression to alter the target scale without discussion. It is unclear whether they apply the model to the transformed $Y$ and then invert back for error metrics, or simply train on the “normal” $Y$ directly. If the latter, the reported RMSE is on a transformed scale (e.g. the CT target “axial position” became normally distributed), which may not reflect true predictive accuracy on the raw metric. Conversely, inverting predictions from the normal space back to the original scale introduces its own approximation. The paper should clarify how target transformation interacts with training and evaluation. This procedural complexity raises doubts about both the validity of the Gaussianity enforcement and the comparability of reported errors.
3.	**Heavy Regularization and Hyperparameters:** PIDReg introduces multiple nonstandard components: the linear‐noise IB (Eq.1), the Gaussianity loss $L_{\text{Gauss}}$, the CS‐based conditional‐independence loss $L_{\text{CMI}}$, and stochastic redundancy assignment. Together, these add at least three new hyperparameters ($\lambda_1,\lambda_2,\lambda_3$) plus kernel width $\sigma$ for the CS estimator (Eq.15). Although the authors claim to fix $\lambda=0.1$ universally, it is surprising that one can balance these terms without per-task tuning. The paper gives no details on how gradients are computed through $W$ or how sensitive $W$ is to batch size. The empirical runtime (Table 11) shows PIDReg is considerably slower (4.64s/epoch vs 1.76 for DER), indicating nontrivial computational overhead. In summary, the method is complex and may be brittle: modest changes to any regularizer or its weight could substantially change the learned decomposition, but this is not explored.
4.	**Ambiguous Interpretability Claims:** The key selling point is that $(w_1,w_2,w_3)$ and the PID components give insight into modality contributions. But this interpretation relies on all assumptions holding perfectly. For example, synergy is modeled purely via $Z_1\odot Z_2$, yet true statistical synergy could exist in other forms (e.g. one modality enabling non‐additive transformations of the other). If the learned $Z_1$ and $Z_2$ are not purely unique, the weight attribution can be misleading. The authors introduce a Bernoulli variable to randomly assign redundancy to one modality, but the impact of this design is not thoroughly analyzed. In Table 4 and Fig.3 the PID values are plausible (e.g. Text-Video having higher synergy), but without ground truth we cannot be sure these correspond to “true” causal contributions. Indeed, recent work emphasizes that there is no unique, universally agreed PID definition, and different valid definitions give different $U,R,S$ splits[11]. Thus, claiming hard explainability may overstate what the method delivers; the “interpretation” is contingent on one particular PID convention and the Gaussian proxy.
5.	**Limited Scope (Two Modalities):** PIDReg is explicitly derived for two modalities. Although an extension to three modalities is sketched in Appendix F, the main paper and experiments only handle bimodal cases (for the tri-modal MOSI/MOSEI they reduce to pairwise combinations). Modern multimodal problems often involve more than two modalities (e.g. vision, audio, text simultaneously). It is unclear how well the method scales to 3+ inputs. The appendix suggests a “pragmatic simplification” for three sources, but without results it is speculative.
6.	**Baselines and Ablations:** While the paper compares many baselines, it omits some natural ones. For example, a simple concatenation‐MLP or linear fusion baseline is not shown; it would be informative to see how much PIDReg’s fancy regularizers improve over a vanilla fusion. The ablation in D.1 reports that removing $L_{\text{Gauss}}$ hurts performance, but we do not see ablations of $L_{\text{CMI}}$ or the IB vs. just end‐to‐end.

**Questions:**

1.	**Joint Gaussianity:** Can you provide empirical evidence (e.g. multivariate normality tests) that $(Z_1,Z_2,Y)$ is close to Gaussian in your trained model? If not, how should one interpret the PID values?
2.	**Target Transformation:** How exactly do you handle the inverse-normal transform of $Y$ in training vs. evaluation? Are the reported RMSE values computed on the original scale of $Y$ or the transformed scale? Please clarify the procedure and its impact on metrics.
3.	**Kernel Width $\sigma$:** How is the kernel width $\sigma$ for the Gram matrices in Eq.(15) chosen? Is it fixed or adaptive? Did you find PIDReg sensitive to this hyperparameter?
4.	**Extension to >2 Modalities:** Appendix F mentions tri-modal PIDReg. Have you actually applied the method to three inputs (e.g. Audio+Visual+Text)? If so, how do you define “synergy” with three sources, and what are the results? If not, how would one generalize the weight-based fusion to more modalities?
5.	**PID Definition:** Which PID measure are you using for the analytic Gaussian solution (minimum‐MI PID à la Williams & Beer)? Different definitions (Broja, $I_{\min}$, etc.) yield different synergy. Why choose this one, and have you checked consistency of your interpretations under alternative PID definitions?

**Details Of Ethics Concerns:**

The paper appears to contain no obvious ethical or societal issues.

---

> ### Author Response · Authors · 2025-11-29
> **Two factual misunderstanding and reply to your Weaknesses 1-3**
>
> First, we sincerely appreciate that reviewer feel our idea is novel and ambitious, and our evaluation is comprehensive, and clear reproducibility (we also provide the code).
>
> However, we would like to gently clarify **two factual misunderstandings** in the review (which is completely understandable):
>
> **Gaussian regularization on the joint $p(Z_1,Z_2,Y)$** Our Shapiro-Wilk test regularization is **not** applied to marginal distributions. It is explicitly imposed on the joint distribution $P(Z_1, Z_2, Y)$.
> The only marginal Gaussian regularization applied to $P(Z_1)$ and $P(Z_2)$ is via the CS divergence in Eq.(8). **That is, we ensure Gaussianity by regularizing both the marginals and the joint distribution.**
>
> **Results of Three Modalities.** We **do include experiments with three modalities on MOSI/MOSEI**. These results are reported in Appendix F.5 (Tables 11 and 12). Incorporating the third modality leads to a clear performance improvement, consistent with the high synergy values detected by our PID analysis.
>
> ---
>
> ### **Weakness 1: Gaussian Regularization on the Joint Distribution**
>
> Please first refer to our clarification above regarding factual point (1).
>
> We also provide empirical justification using synthetic datasets where ground truth is available. When the latent variables are highly nonlinear (**Appendix G.4.2**) or even discrete (**Appendix G.4.3**), PIDReg still faithfully reflects the true data properties, whereas PID-BATCH in [1] fails to capture certain subtle trends (see Figure 14 in Appendix G.7). This advantage and faithfulness also appear in real-world data (Appendix D.2 and Appendix G.7).
>
> The PID formulation is inherently underdetermined, any practical method must impose additional constraints. The Gaussian PID assumption simply means that we estimate PID values using only 1st and 2nd order information.
>
> ---
>
> ### **Weakness 2: Target Transformation**
>
> The rank-based inverse-normal transformation of $Y$ is used **solely** to satisfy the Gaussian-PID requirement and is completely **separated** from the computation of $𝓛_{\mathrm{Pred}}$ and all performance evaluations.
> We never train or evaluate the model on the normalized target. All reported results reflect the true predictive accuracy on the original, raw target metric.
>
> **Prediction branch:**
> - Input raw target $Y$ (physical units).
> - Compute dataset-level mean $\mu$ and standard deviation $\sigma$ and define $Y_{\\mathrm{scaled}} = (Y - \\mu) / \\sigma$
> - Train the model only on $(Z_1, Z_2, Y_{\\mathrm{scaled}})$ to optimize $\mathcal{L}_{\\mathrm{Pred}}$.
> - At evaluation, de-normalize prediction and compute all metrics in physical units using $(Y_{\\mathrm{pred}}, Y)$.
>
> **PID / Gaussianization branch:**
> - From raw $Y$ compute $Y_{\\mathrm{gauss}} = \\text{RankInverseNormalTransform}(Y)$.
> - Use $(Z_1, Z_2, Y_{\\mathrm{gauss}})$ to estimate joint covariance, and compute $\mathcal{L}_{\\mathrm{Gauss}}$ and PID.
>
> When computing predictive metrics, the target $Y$ is scaled using a standard Z-score normalization $Y_{\\mathrm{scaled}}$ which is the conventional practice in regression. Therefore, all reported errors are in the true physical units (e.g., axial position, years).
>
> ---
>
> ### **Weakness 3: Hyperparameter Settings and Sensitivity Analysis**
>
> Our final objective in Eq.(17) contains three regularization terms, but each term has a clear physical meaning.
> We observed that, within each mini-batch, the raw values of $𝓛_{\mathrm{pred}}$, $𝓛_{\mathrm{Gauss}}$, $𝓛_{\mathrm{CMI}}$, and $𝓛_{\mathrm{CS}}$ naturally fall within a similar scale and are of MSE-comparable magnitude. This observation motivates using a shared $\lambda$ for all three regularization terms, which also reduces the number of hyperparameters.
>
> We provide a detailed $\lambda$-sensitivity analysis in **Appendix G.5** using the superconductivity dataset, which further justifies its robustness (see also Table below).
>
> | $\lambda$ | RMSE | Corr | U$_1$ | U$_2$ | R | S |
> |:---:|:---:|:---:|:---:|:---:|:---:|:---:|
> | 0.01 | 10.405 | 0.952 | 0.369 | 0.001 | 0.847 | 0.378 |
> | 0.05 | 10.492 | 0.951 | 0.372 | 0.000 | 0.856 | 0.380 |
> | 0.1  | 10.370 | 0.952 | 0.375 | 0.001 | 0.878 | 0.394 |
> | 0.5  | 10.593 | 0.950 | 0.375 | 0.001 | 0.854 | 0.379 |
> | 1.0  | 10.588 | 0.951 | 0.375 | 0.001 | 0.853 | 0.380 |
> | Relative Change | 2.15% | 0.21% | 1.63% | 0 | 3.66% | 4.23% |
>
>
> **Gradients and sensitivity of W:** Regarding the gradients of $W$ in $𝓛_{\\mathrm{Gauss}}$, we would like to clarify that the gradient is automattically computed and the loss is fully end-to-end differentiable.
>
> **Kernel Width:** The kernel width $\sigma$ in Eq.(15) is determined entirely by the common data-adaptive median rule heuristic.
>
> **Run Time:** The computational cost of PIDReg is of the same order of magnitude as the baselines and remains moderate, imposing no practical limitations on real-world use. At the same time, it provides interpretability that the other baseline models lack.

---

> ### Author Response · Authors · 2025-11-29
> **Reply to your Weaknesses 4 and 5**
>
> ---
>
> ### **Weakness 4: The Interpretability Statement, Synergy Modeling Mechanism, and Bernoulli sampling**
>
> We would like to clarify the our interpretability is not overstated. Our work is the first to embed PID decomposition directly into a multimodal learning framework, providing built-in (rather than post-hoc) interpretability.
> While we instantiate synergy modeling using the Hadamard product, this does not imply that synergy must be modeled in this way; it is simply a practical and effective choice.
>
> Moreover, when the ground truth is known, the faithfulness of our interpretability (and its clear advantage over post-hoc approaches) is demonstrated by new empirical results in **Appendix G.4** and **Appendix G.7**,
> where we modify the latent distributions to be highly non-Gaussian or even discrete. In all such cases, our method substantially outperforms PID-BATCH [Liang et al., 2023], the first work to use PID for post-hoc multimodal interpretability.
>
> When the ground truth is unknown, the faithfulness of our PID explanations is also strongly supported by ablation studies on real-world datasets (**Appendix D.2** and **Appendix G.7**).
> We observe that when PIDReg identifies high synergy between two modalities, the fused model consistently outperforms the best single-modality models.
> Likewise, when PIDReg estimates that a modality carries nearly zero unique information, omitting that modality leads to no noticeable degradation in performance.
>
> **Synergy Modeling:** Using the Hadamard product to model synergy is simple. As discussed in Section 3.1, it has been successfully applied in domains, such as drug synergy prediction, visual question answering, and multimodal sentiment analysis.
>
> For your suggestion, we compare the Hadamard product with five alternative synergy mechanisms, including tensor product, bilinear fusion, and attention-based fusion (details in **Appendix G.6**). According to Table 18 or the following table, Hadamard fusion achieves the most stable and competitive performance on both the full CMU-MOSEI dataset and the challenging sarcasm subset, where synergistic reasoning is particularly critical. Bilinear fusion performs well on the full dataset but degrades on the sarcasm subset, whereas tensor product fusion shows strong synergy detection but weaker overall performance.
>
> Finally, unlike these alternative mechanisms, the Hadamard product is parameter-free.
>
> |  | $A_7$ | $A_2$ | $F1$ | $MAE$ | $Corr$ |
> |:---:|:---:|:---:|:---:|:---:|:---:|
> |  |  |  | **Text+Vision** |  |  |
> | Hadamard | 47.0 | 80.6 | 80.2 | 0.642 | 0.661 |
> | Tensor Products | 45.7 | 72.2 | 72.4 | 0.682 | 0.632 |
> | 2D Convolutions | 28.6 | 66.1 | 52.6 | 0.894 | 0.616 |
> | Bilinear | 47.6 | 67.3 | 73.1 | 0.636 | 0.671 |
> | Concat+MLP | 46.5 | 72.2 | 74.5 | 0.669 | 0.656 |
> | Concat+Attention | 46.7 | 69.7 | 73.4 | 0.645 | 0.647 |
> |  |  |  | **Conflict Subset** |  |  |
> | Hadamard | 45.3 | 62.0 | 59.9 | 0.680 | 0.459 |
> | Tensor Products | 46.2 | 60.8 | 80.3 | 0.672 | 0.450 |
> | 2D Convolutions | 46.9 | 63.0 | 55.0 | 0.711 | 0.466 |
> | Bilinear | 30.4 | 62.9 | 29.7 | 0.894 | 0.389 |
> | Concat+MLP | 45.4 | 56.8 | 60.6 | 0.705 | 0.439 |
> | Concat+Attention | 45.3 | 64.2 | 60.4 | 0.755 | 0.431 |
>
> **Bernoulli Mechanism:** The Bernoulli mechanism ensures a symmetric treatment of redundancy across the two modalities, allowing the redundant information to be attributed to either modality without bias.
>
> To justify this claim, we perform an ablation study on the Bernoulli mechanism as follows: Using the Superconductivity dataset as an example, we remove the Bernoulli component and force all redundancy to be assigned to either Modal 1 or Modal 2.
> Each scenario is repeated five times and averaged. The results are summarized in the table below:
>
> | | $RMSE$ | $Corr$ | $U_1$ | $U_2$ | $R$ | $S$ |
> |---|---|---|---|---|---|---|
> | Bernoulli | 10.391 | 0.952 | 0.385 | 0.002 | 0.887 | 0.389 |
> | Add to Modal 1 | 10.410 | 0.952 | 0.369 | 0.000 | 0.895 | 0.389 |
> | Add to Model 2 | 10.385 | 0.952 | 0.361 | 0.004 | 1.101 | 0.470 |
>
> The results show that, after 200 training epochs, the PID decomposition values remain nearly identical, indicating that the redundancy term is indeed insensitive to which modality it is attributed to.
>
>
> ---
>
> ### **Weakness 5: Clarification on Two Modalities**
>
> **We have empirical results for three modalities. This is a factual misunderstanding.**
>
> Two-modality applications cover the vast majority of multimodal learning scenarios and serve as the atomic units for validating interaction mechanisms.  Moreover, the baselines used in our comparison, DER,  MEIB, etc. are explicitly designed as bimodal frameworks in their original papers.
>
> Appendix F is **not** a sketch; it provides a complete and principled mathematical extension from the bimodal PIDReg framework to the tri-modal case.
> Moreover, Appendix F.5 presents the full tri-modal (Text–Vision–Audio) end-to-end **experimental results** (see Tables 11 and 12).

---

> ### Author Response · Authors · 2025-11-29
> **Reply to your Weakness 6 and remaining questions**
>
> ### **Weakness 6: Vanilla Baselines and Misconception Regarding Ablations**
>
> Thanks for the suggestion. To further strengthen our experimental design, we additionally introduce two Vanilla baselines, Concat-MLP and Simple-Fusion, while keeping the feature extractors (Encoder/Projector) identical to PIDReg and holding all training hyperparameters (batch size, epochs, etc.) strictly fixed. The only change lies in the fusion layer:
>
> - **Concat-MLP:** Remove PID computation, weighted aggregation, and synergy term. We extract $Z_1$ and $Z_2$, form $Z_{\\mathrm{fused}} = \\text{Concat}(Z_1, Z_2)$, and feed $Z_{\\mathrm{fused}}$ into the MLP predictor.
> - **Simple-Fusion:** To isolate the effect of dynamic PID-driven weighting, we retain the same fusion form $Z = w_1 Z_1 + w_2 Z_2 + w_3 (Z_1 \\odot Z_2)$, but disable PID computation and force $w_1 = w_2 = w_3 = 1/3$.
>
> Using the Superconductivity dataset as an example, the results:
>
> | | $RMSE$ | $Corr$ |
> |---|---|---|
> | PIDReg | 10.37 | 0.952 |
> | Concat-MLP | 11.44 | 0.943 |
> | Simple-Fusion | 11.24 | 0.944 |
>
> These results show that the information-theoretic guidance provided by PIDReg offers a deeper and more principled learning signal than simple feature concatenation or naïve fusion. Moreover, PIDReg provides intrinsic interpretability that generic fusion mechanisms fundamentally lack.
>
> **We must point out that the claim that we did not present ablations for $\mathcal{L}_{\\mathrm{CMI}}$ or the IB component is also a factual misunderstanding**. We explicitly provide detailed ablation studies for both in the appendix:
>
> - **Ablation of $\mathcal{L}_{\\mathrm{CMI}}$** (Appendix D.1.2, Table 5): Removing $\mathcal{L}_{\\mathrm{CMI}}$ leads to severe cross-modal information leakage, confirming its necessity for enforcing information purity.
> - **Ablation of the IB mechanism** (Appendix D.3, Table 8): Eliminating the IB component (Eq. 1) significantly degrades performance, demonstrating that IB is not redundant; it is essential for achieving SOTA predictive accuracy as well as for the theoretical completeness of PIDReg.
>
> ---
>
> ### **Question 1: Joint Gaussianity**
>
> We thank the reviewer for raising this question. **Appendix D.1.1** (Fig. 6) was specifically designed to provide empirical evidence that the joint distribution $P(Z_1, Z_2, Y)$ is close to Gaussian.
> When both $L_{CS}$ and $L_{Gauss}$ are removed, the latent joint distribution becomes highly anisotropic.
> In contrast, when we enable $𝓛_{\mathrm{CS}}$ (marginal Gaussianization) together with $𝓛_{\mathrm{Gauss}}$ (joint Gaussian constraint), the latent samples are consistently pulled back toward the origin, forming a tight cluster that exhibits the isotropic structure characteristic of a multivariate Gaussian.
>
> ---
>
> ### **Question 2: Target Transformation**
>
> Please refer to **Weakness 2**.
>
> ---
>
> ### **Question 3: Kernel Width**
>
> Please refer to **Weakness 3**. Our kernel width is data-adaptive, following the median-rule heuristics, rather than manually set.
>
> ---
>
> ### **Question 4: Extension to >2 Modalities**
>
> Please refer to **Weakness 5**. **We do have results in three modalities in Appendix F.5 (Tables 11 and 12). This is a misunderstanding.**
>
> ---
>
> ### **Question 5: PID Definition**
> Thanks for the very good question. We follow the definition of BROJA [1], which is one of the most popular definition and satisify much more desirable properties like additivity and continuity than other definitions (please refer to Table 1 in [2])
> In this paper, we also extend BROJA definition to three or more variables, and successfully tested the results in real-world data (see Appendix F.5).
>
> We did not use $I_{min}$. This is because, for a multivariate target $Y$, $I_{min}$ suffers from known theoretical issues and produces biased estimates [3].
> On the other hand, we also do not use the original definition of Williams and Beer, because their notion of "specific information" is only well-defined in the discrete setting.
>
> We have indeed compared our PID estimates with other PID approaches, such as PID-BATCH [4], which is a post-hoc explainability method. As shown in **Appendix G.7**, our PIDReg produces largely consistent estimates for the relative relationships between $U_1$, $U_2$, $S$, and $R$.
> In cases where our results differ from those of PID-BATCH, we additionally conduct ablation studies to justify the correctness of our decomposition (Tables 19 and 20).
>
> Finally, we want to emphasize that, in all synthetic data in which ground truth is known, our PID estimation precisely captures almost all correct relationships (see results in **Appendix G.4**).

---

> ### Author Response · Authors · 2025-11-29
> **references**
>
> [1] Bertschinger, Nils, et al. "Quantifying unique information." Entropy 16.4 (2014): 2161-2183.
>
> [2] Lyu, Aobo, Andrew Clark, and Netanel Raviv. "Multivariate Partial Information Decomposition: Constructions, Inconsistencies, and Alternative Measures." arXiv preprint arXiv:2508.05530 (2025).
>
> [3] Venkatesh, Praveen, and Gabriel Schamberg. "Partial information decomposition via deficiency for multivariate gaussians." 2022 IEEE International Symposium on Information Theory (ISIT), 2022.
>
> [4] Liang, Paul Pu, et al. "Quantifying & modeling multimodal interactions: An information decomposition framework." Advances in Neural Information Processing Systems 36 (2023): 27351-27393.

---

### Official Review · Reviewer_dZ8M · 2025-11-01

**Soundness:** 3
**Presentation:** 3
**Contribution:** 2
**Rating:** 4
**Confidence:** 3

**Summary:**

This paper introduces PIDReg, a framework for explainable multimodal regression that uses Partial Information Decomposition (PID) to quantify unique, redundant, and synergistic information from different modalities. To make the intractable PID problem solvable within a deep learning context, the authors enforce a multivariate Gaussian distribution on the joint latent space of the modality representations and the target variable. This strong inductive bias enables an analytical PID solution, which guides a feature fusion module in an end-to-end trained model. The authors claim their method achieves state-of-the-art predictive performance and provides faithful modality-level interpretations across several datasets.

**Strengths:**

1. The paper's goal of creating an intrinsically interpretable multimodal model is appreciated. Framing modality contributions through the principled, information-theoretic lens of PID is a novel and creative approach that could provide a much richer vocabulary for model explanation.

2. The evaluation is comprehensive, spanning six real-world datasets from diverse domains, which demonstrates the framework's versatility. The neuroimaging case study, where the model's interpretations align with existing clinical evidence, is particularly compelling.

**Weaknesses:**

1. The framework's entire interpretability claim rests on a fragile assumption that the joint latent space can be molded into a Gaussian without destroying the true underlying information dynamics. It seems that the method does not discover the information structure. Instead, it imposes a Gaussian one and then analyzes it. Are the reported PID values a true reflection of the data's properties, or are they merely artifacts of this powerful, and likely mismatched, prior? The lack of guarantees of identifiability (of true latent variables/structures) is a critical downside, in my opinion.

2. Along the same line, synergy is modeled exclusively via the Hadamard product of latent vectors. This is a highly specific, simplistic, and arbitrary choice for modeling what are likely complex, non-linear interactions. The paper offers no theoretical justification for this choice nor does it ablate any alternatives (e.g., bilinear pooling, attention).

3. The paper claims state-of-the-art predictive performance but primarily compares against other Information Bottleneck (IB) based methods. This is a niche subfield. The most powerful and common multimodal architectures today rely on mechanisms like cross-attention.

**Questions:**

Please see the weaknesses.

---

> ### Author Response · Authors · 2025-11-29
> **Reply to Your Weaknesses 1 and 2**
>
> ### **Weakness 1: Gaussianity, artifacts, and identifiability**
>
> Thank you for raising this important point. The PID formula is inherently underdetermined (three equations, four unknown variables), and any practical method must impose extra constraints. Gaussian PID, to our knowledge, is the only tractable formulation that yields a unique decomposition in high-dimensional, continuous settings, and our Gaussian regularizer simply encourages the latent space to satisfy the assumptions required for this analytic solution.
>
> That is, our PIDReg essentially relies only on first- and second-order statistics to estimate the PID values. Since many real-world dependencies are well captured by the first two moments, this trade-off preserves the majority of meaningful structure. We acknowledge, however, that extending the framework to incorporate higher-order information is an interesting future direction.
>
> **Additional Empirical evidence that PIDReg reflects the true underlying structure**
>
> To empirically demonstrate that PIDReg faithfully reflects the underlying data properties, **Appendix G.4.2** (highly non-Gaussian latents) and **Appendix G.4.3** (discrete latents) evaluate PIDReg on generative models defined in Eqs. (22)–(23). Across all configurations, we consistently observe two key phenomena:
>
> * Inactive components are correctly identified, e.g. if $w_{u1} = 0$ then  $U_1 \approx 0$.
>
> * Relative strengths of the components are correctly captured. In particular, when the true synergy outweighs redundancy, the estimated $S$ is correspondingly larger than the estimated $R$. e.g.,
> if $w_s^{(2)} > w_s^{(1)}$ and $w_r^{(2)} < w_r^{(1)}$, then
> $S_{\mathrm{est}}^{(2)} > S_{\mathrm{est}}^{(1)}$ and $R_{\mathrm{est}}^{(2)} < R_{\mathrm{est}}^{(1)}$.
>
> The fact that our estimated PID components faithfully reflect data properties is also supported by real-world experiments. As shown in **Appendix D.2** and **Appendix G.7**, when the estimated synergy $S$ between two modalities approaches zero, the fused model does not outperform a strong single-modality baseline. Moreover, when the estimated unique information $U$ of a modality is nearly zero, incorporating that modality into the fusion process yields no measurable performance improvement.
>
> **Gaussian regularization in latent space does not introduce obvious artifacts**
>
> To test whether Gaussian enforcement in latent space introduces artifacts, we removed all Gaussianity regularizers and computed PID using only first- and second-order empirical statistics.
> As shown in **Appendix G.4.2** (Table 16) or the following table, the PID estimates remain stable: both $U_1$ and $U_2$ remain nearly zero, and synergy $S$ consistently outweighs redundancy $R$ by a clear margin.
>
> |  | $U_1$ | $U_2$ | $R$ | $S$ |
> |:---:|:---:|:---:|:---:|:---:|
> | Baseline | 0.04 | 0.02 | 0.31 | 0.63 |
> | No $𝓛_{CS}$ | 0.06 | 0.03 | 0.26 | 0.56 |
> | No $𝓛_{Gauss}$ | 0.05 | 0.03 | 0.25 | 0.54 |
> | No $𝓛_{CS}, 𝓛_{Gauss}$ | 0.03 | 0.03 | 0.26 | 0.49 |
>
> ---
>
> ### **Weakness 2: Design Motivation and Efficiency of Hadamard Product**
>
> Using the Hadamard product to model synergy is simple, but it is not an arbitrary design choice. As discussed in Section 3.1, it has been successfully applied in domains, such as drug synergy prediction, VQA, and sentiment analysis.
>
> We compare the Hadamard product with five alternative synergy mechanisms, including tensor product, bilinear fusion, and attention-based fusion. As shown in **Appendix G.6**, the Hadamard product demonstrates a clear and consistent performance advantage.
>
> According to the following table, Hadamard fusion achieves the most stable and competitive performance on both the full CMU-MOSEI dataset and the challenging sarcasm subset, where Text+Vision aids sarcasm detection and synergistic reasoning is particularly critical. Bilinear fusion performs well on the full dataset but degrades on the sarcasm subset, whereas tensor product fusion shows strong synergy detection but weaker overall performance.
>
> Finally, unlike these alternative mechanisms, the Hadamard product is parameter-free, which makes it computationally efficient and less prone to overfitting.
>
> |  | $A_7$ | $A_2$ | $F1$ | $MAE$ | $Corr$ |
> |:---:|:---:|:---:|:---:|:---:|:---:|
> |  |  |  | **Text+Vision** |  |  |
> | Hadamard | 47.0 | 80.6 | 80.2 | 0.642 | 0.661 |
> | Tensor Products | 45.7 | 72.2 | 72.4 | 0.682 | 0.632 |
> | 2D Convolutions | 28.6 | 66.1 | 52.6 | 0.894 | 0.616 |
> | Bilinear | 47.6 | 67.3 | 73.1 | 0.636 | 0.671 |
> | Concat+MLP | 46.5 | 72.2 | 74.5 | 0.669 | 0.656 |
> | Concat+Attention | 46.7 | 69.7 | 73.4 | 0.645 | 0.647 |
> |  |  |  | **Conflict Subset** |  |  |
> | Hadamard | 45.3 | 62.0 | 59.9 | 0.680 | 0.459 |
> | Tensor Products | 46.2 | 60.8 | 80.3 | 0.672 | 0.450 |
> | 2D Convolutions | 46.9 | 63.0 | 55.0 | 0.711 | 0.466 |
> | Bilinear | 30.4 | 62.9 | 29.7 | 0.894 | 0.389 |
> | Concat+MLP | 45.4 | 56.8 | 60.6 | 0.705 | 0.439 |
> | Concat+Attention | 45.3 | 64.2 | 60.4 | 0.755 | 0.431 |

---

> ### Author Response · Authors · 2025-11-29
> **Reply to Your Weakness 3**
>
> ### **Weakness 3: Comparison with Cross-Attention Mechanism**
>
> We **not only** compared PIDReg against state-of-the-art information-bottleneck methods such as MIB and MEIB, **but also** included strong baselines from contrastive multimodal learning (e.g., CoMM, **ICLR 2025**) and uncertainty-aware multimodal regression models.
>
> We fully agree that cross-attention is one of the most powerful and widely used multimodal architectures. To rigorously address the concern regarding Cross-Attention (CA), following [1], we implemented a variant PIDReg-CA to conduct a controlled comparison.
>
> To isolate the impact of the fusion mechanism, we constructed PIDReg-CA by keeping the modality-specific encoders ($h_{\phi_m}$) identical to PIDReg but replacing the PID-guided fusion with a standard Transformer-based interaction module. Latent vectors of three modalities, $Z_T, Z_A, Z_V \in \mathbb{R}^d$, are treated as a sequence of tokens $S \in \mathbb{R}^{3 \times d}$. These are fed into a Transformer Encoder layer to model dense pairwise interactions via Multi-Head Self-Attention (MHSA):
>
> $$\tilde{S} = \text{MHSA}(\text{Concat}(Z_T, Z_A, Z_V)) + \text{Concat}(Z_T, Z_A, Z_V).$$
>
> The fused tokens are flattened and passed to the same MLP predictor used in PIDReg.
> **We would like to emphasize that such cross-attention mechanisms sacrifice interpretability and eliminate the PID decomposition entirely, which runs counter to the core motivation of our paper.**
>
> Using CMU-MOSEI as the test dataset, we compare our PIDReg against its variant (PIDReg-CA) and external SOTA architectures with more advanced CA mechanisms [2,3]. The results are shown as:
>
> | Method             | $A_7$ | $A_2$ |  F1   |  MAE  |  Corr |
> | ------------------ | :---: | :---: | :---: | :---: | :---: |
> | **PIDReg (Tri-Modal)** | 48.7  | 81.7  | 81.5  | 0.620 | 0.679 |
> | **PIDReg-CA** | 48.9 | 81.7 | 81.6  | 0.618 | 0.681 |
> | MulT [2]           | 48.4 | 80.9 | 80.9 | 0.673 | 0.677 |
> | CENet [3]          | 52.3 | 82.1 | 82.4 | 0.588 | 0.738 |
>
> Comparing PIDReg with PIDReg-CA, the cross-attention mechanism yields only negligible performance gains while completely sacrificing interpretability. More advanced cross-attention mechanisms may or may not yield additional accuracy gains, **but they still lack the built-in interpretability that PIDReg provides**.
>
> [1] Wang, Di, et al. "TETFN: A text enhanced transformer fusion network for multimodal sentiment analysis." Pattern Recognition 136 (2023): 109259.
>
> [2] Tsai, Yao-Hung Hubert, et al. "Multimodal transformer for unaligned multimodal language sequences." Proceedings of the conference. Association for computational linguistics. Meeting. Vol. 2019. 2019.
>
> [3] Wang, Di, et al. "Cross-modal enhancement network for multimodal sentiment analysis." IEEE Transactions on Multimedia 25 (2022): 4909-4921.

---

### Meta-Review · Area_Chair_tNE5 · 2026-01-01

**Summary:**

The reviewers’ main concerns focus on the identifiability and faithfulness of the learned representations, the strength and realism of the Gaussian assumption, and the adequacy of the experimental evaluation. In particular, reviewers question whether the modality-specific representations z1 and z2 can be reliably and uniquely recovered, or whether they may instead be mixtures of shared and unique information. There are also concerns that enforcing Gaussian structure constitutes a strong inductive bias that may limit the interpretability and generality of the method. Finally, reviewers note that the experimental evaluation is not yet sufficient to fully support the claims or to demonstrate robustness and practical relevance

**Reviewer Concerns:**

The main concerns raised by the reviewers fall into three categories:

1. The lack of guarantees that the learned representations, especially z1 and z2, which are intended to capture modality-unique information, are accurately or faithfully recovered (identifiability concerns raised by Reviewer dZ8M and also questioned by Reviewer KZFk, who noted that the learned z1 and z2 may not be purely unique);

2. The Gaussian assumption, raised by Reviewer KZFk; and

3. The sufficiency and completeness of the experimental evaluation, raised by all reviewers.

---

The authors’ additional assumptions and clarifications are helpful. However, the key questions regarding identifiability and the Gaussian assumption remain unclear.

1) Identifiability of modality-specific representations

In general, accurately extracting modality-specific information z1 (and z2) solely by using different encoders for different modalities is not guaranteed without additional assumptions. That is, there is no theoretical guarantee that the learned z1 and z2 are purely modality-specific; in practice, they may be mixtures of modality-specific and modality-shared information. As a result, the interpretation of the corresponding weights w1 and w2 (and thus w3) remains potentially problematic.

2) Gaussian assumption

Regarding the Gaussian assumption, the authors respond that “our Shapiro–Wilk test regularization is not applied to marginal distributions. It is explicitly imposed on the joint distribution p(z1,z2,Y). The only marginal Gaussian regularization applied to p(z1) and p(z2)… That is, we ensure Gaussianity by regularizing both the marginals and the joint distribution.” However, once the joint distribution p(z1,z2,Y) is Gaussian, the marginals are necessarily Gaussian as well. This raises further questions about the role, necessity, and implications of the Gaussian constraints and how they affect the learned representations and the resulting decomposition.

**Reviewer Scores:**

Reviewer dZ8M: The concern about identifiability of the learned modality-specific representations remains unresolved.

Reviewer KZFk: The concerns regarding both the Gaussian assumption and the faithfulness of the modality-unique representations remain outstanding.

Overall: The main concerns raised by the reviewers remain outstanding; therefore, no positive score changes are expected.

---

### Decision · Program_Chairs · 2026-01-26

Reject